# ANALYSIS OF AN IDEALIZED STOCHASTIC POLYAK METHOD AND ITS APPLICATION TO BLACK-BOX MODEL DISTILLATION

## ABSTRACT

We provide a general convergence theorem of an idealized stochastic Polyak step size called SPS*. Besides convexity, we only assume a local expected gradient bound, that includes locally smooth and locally Lipschitz losses as special cases. We refer to SPS* as idealized because it requires access to the loss for every training batch evaluated at a solution. It is also ideal, in that it achieves the optimal lower bound for globally Lipschitz function, and is the first Polyak step size to have an $\mathcal{O}(1/\sqrt{t})$ anytime convergence in the smooth setting. We show how to combine SPS* with momentum to achieve the same favorable rates for the last iterate. We conclude with several experiments to validate our theory, and a more practical setting showing how we can distill a teacher GPT-2 model into a smaller student model without any hyperparameter tuning.

## 1 INTRODUCTION

Consider the problem

$$x_* \in \operatorname*{argmin}_{x \in \mathbb{R}^d} f(x), \quad f(x) := \mathbb{E}_{\xi \sim \mathcal{P}} \left[ f_\xi(x) \right], \tag{1}$$

where $\mathcal{P}$ is a distribution over the data, the loss functions $f_\xi$ are convex. We assume that the minimizer $x_* \in \mathbb{R}^d$ exists, and that $\mathbb{E}_\xi \left[ \inf f_\xi \right] > -\infty$ (e.g. the losses are nonnegative).

One of the main costs in developing new machine learning models is training them, that is, finding an approximate solution to (1). The training of GPT-4 is estimated to have cost over \$40M (Cottier et al., 2024). The elevated cost of training bigger models, and the success of Adam (Kingma & Ba, 2015), has sparked an intense research effort into developing new stochastic optimization methods. Yet the performance difference among many newly developed methods is minimal when the step size is tuned (Schmidt et al., 2021). Finding a good step size often involves multiple re-runs on a subset of the data, which adds considerably to this cost.

Here we advance the theory of an adaptive stochastic Polyak step size. The Polyak step size uses both the current loss and gradient norm to compute a step size at each iteration.

We show that if we had access to $f_\xi(x_*)$, the value of the loss at the solution for each batch $\xi$ of data, a variant of the stochastic Polyak step we call SPS* achieves the best known rates across several subclasses of convex functions. Specifically, we show that SPS* achieves either the optimal rate when known, or the best known rate, for convex functions, including Lipschitz, smooth, and strongly convex. Furthermore we only require that these assumptions hold in a ball around the solution. This mirrors the same result in the deterministic setting for the Polyak step size (Hazan & Kakade, 2019).

We also prove convergence in the finite-sum, convex and continuous setting, without any additional assumption, for which we are unaware of any other stochastic method that provably converges.

We then show how to combine this Polyak step size with momentum, in such a way that the last-iterate converges at the optimal (competitive) rate in the Lipschitz (smooth) setting. For this we use *iterate averaging*, which is one of the many equivalent ways of writing momentum (Sebbouh et al., 2021).

These fast and adaptive convergence results speak to the strength of the SPS* method. However, they also show that having access to $f_\xi(x_*)$ for every $\xi$ is a strong assumption, which we can

not expect to hold in general. But we do consider two settings where $f_\xi(x_*)$ is known or can be approximated. The first setting is that of interpolation, where typically $f_\xi(x_*) = 0$ or is relatively easy to compute (Loizou et al., 2021). The second setting is one we call *blackbox model distillation*. In this setting, we can query a teacher (a larger pretrained model) with any input, but we do not have access to the teachers architecture or weights. Our objective is to train the student (a smaller model) on one of the tasks that the teacher is accomplished. The teacher's loss on each input serves as an approximation of $f_\xi(x_*)$ for the student. This enables us to use SPS* with momentum to set the step size for the student, and train it efficiently without having to tune any hyper-parameters.

## 1.1 STOCHASTIC POLYAK STEP SIZE

Here we analyse the following variant of the SPS (Stochastic Polyak step size) method

$$x_{t+1} \; = \; x_t - \gamma_t^{\text{SPS*}} g_t, \qquad \gamma_t^{\text{SPS*}} := \frac{(f_t(x_t) - f_t(x_*))_+}{\|g_t\|^2} \qquad (2)$$

where $\xi_t \sim \mathcal{P}$ is sampled i.i.d at each iteration, and $g_t$ denotes either a gradient (smooth setting) or a subgradient (non-smooth setting) of $f_t := f_{\xi_t}$ evaluated at $x_t$. Throughout, we use the notation $(z)_+ := \max\{z, 0\}$ for $z \in \mathbb{R}$. We refer to (2) as a the SPS* method. We will prove several anytime convergence rates for SPS*. By *anytime*, we mean a proof that the method converges to any predefined tolerance without prior knowledge of that tolerance.

See Table 1 for a comparison between our rates of convergence, that of other variants of SPS, and the best known anytime rates for SGD in each setting. For the SGD rates within each setting, we included rates that rely on the global problem constants. For instance, to achieve the $GD/\sqrt{t}$ rate in the $G$-Lipschitz setting, we need to set the step size as $\gamma = \frac{D}{G}\frac{1}{\sqrt{t}}$, and we need to project the iterates of SGD back onto the ball of radius $D := \|x_0 - x_*\|$. In contrast, SPS* achieves this rate without without access to $G$ or $D$, but with access to $f_\xi(x_*)$ instead.

The main downside to (2) is that it requires access to $f_t(x_*)$. This is why we refer to SPS* as an idealized variant, both because of its ideal convergence rates, and this idealized setting of assuming access to $f_t(x_*)$. In this sense, the comparisons in Table 1 to alternative Polyak type methods are not entirely fair, because they do not require such access to $f_t(x_*)$. Our message here is not that SPS* is a better method than $\text{SPS}_{\max}$, NGN or DecSPS, but rather that $f_t(x_*)$ is the object that we should try to approximate, or learn on the fly.

We do however consider two settings where access to, or approximating, $f_t(x_*)$ is reasonable. One setting where $f_t(x_*)$ is often known is the interpolation setting, where we assume that there exists a minimizer $x_* \in \mathbb{R}^d$ such that the loss over every data is simultaneously minimized, in other words

$$f_\xi(x_*) \; = \; \inf_{x \in \mathbb{R}^d} f_\xi(x), \quad \forall \xi \in \text{support}(\mathcal{P}). \qquad (3)$$

Thus under interpolation, our model has a perfect fit (as measured by $f_\xi(x)$) for every data point. Typically the loss is a non-negative function and its infimum is zero (Loizou et al., 2021), that is $\inf_{x \in \mathbb{R}^d} f_\xi(x) = 0$. When this is the case, we have access to every $f_\xi(x_*)$, which happens to be zero. Alternatively when $\inf f_\xi(x)$ is close to zero, then using zero as an approximation is reasonable. Finally, even when $\inf f_\xi(x)$ is far from zero, it can sometimes be approximated (Loizou et al., 2021).

The ease of approximating $\inf f_\xi(x)$ is what motivated $\text{SPS}_{\max}$ (Loizou et al., 2021) which uses

$$\gamma_t^{\text{SPS}_{\max}} := \min\left\{ \frac{f_t(x_t) - \inf_x f_t(x)}{\|g_t\|^2}, \gamma_b \right\}, \qquad (4)$$

where $\gamma_b > 0$ is an additional hyperparameter to safe-guard against excessively large step sizes. Loizou et al. (2021) present a comprehensive analysis of $\text{SPS}_{\max}$ in the non-smooth, smooth and strongly convex setting. But in all these cases, $\text{SPS}_{\max}$ is only guaranteed to converge when interpolation holds. Outside of interpolation, $\text{SPS}_{\max}$ converges to a neighborhood of the solution. Here we show that it is not necessary to assume that interpolation holds to establish convergence of a SPS type method. Having access to $f_\xi(x_*)$ is sufficient.

To be clear, assuming access to $f_\xi(x_*)$ is not the same as assuming that interpolation holds. Interpolation (3) imposes constraints on the data and the model, usually requiring the model to be overparameterized (Ma et al., 2018; Liu et al., 2022; Gower et al., 2021). In contrast having access to $f_\xi(x_*)$ imposes no constraints on the model and data. Furthermore, there are settings outside of

Table 1: A summary of anytime convergence rates for variants of stochastic Polyak step size. Notation: $D = \|x_0 - x_*\|$, $\Delta_* = \inf f - \mathbb{E}[\inf f_\xi]$, $\Delta_{pos} = \mathbb{E}[\inf f_\xi]$, $G^2 = \max_x \mathbb{E}_\xi\left[\|\nabla f_\xi(x)\|^2\right]$. We compare to the stochastic Polyak methods DecSPS$^{(3)}$, SPSmax Loizou et al. (2021) and NGN (Orvieto & Xiao, 2024). The proof of convergence for SPS* in the Lipschitz convex and strongly convex setting was first given by Garrigos & Gower (2023) and Pedregosa & Schaipp (2023), respectively. For the results making use of the gradient variance constant $\sigma_*^2 = \mathbb{V}_\xi[\nabla f_\xi(x_*)]$, we replaced it with its upper bound $L\Delta_*$ for a more uniform comparison.

| Algorithm | Convex finite sum | $G$-Lipschitz problems | $L$-Smooth problems | $L$-Smooth $\mu$-Convex | $G$-Lipschitz $\mu$-Convex |
|---|---|---|---|---|---|
| DecSPS$^{(3)}$ | ✗ | ✗ | ✗ | $\frac{LD^2 + \Delta_*}{\sqrt{t}}$ | ✗ |
| SPSmax | ✗ | ✗ | $\frac{LD^2}{t} + \Delta_* L$ | $\left(1 - \frac{\mu}{L}\right)^t D^2 + \frac{\Delta_* L}{\mu}$ | ✗ |
| NGN | ✗ | ✗ | $\frac{L^2 D^2}{\sqrt{t}} + \frac{L(\Delta_* + L\Delta_{pos})\log(t)}{\sqrt{t}}$ | ✓$^{(4)}$ | ✗ |
| SGD*$^{(2)}$ | ✗ | $\frac{GD}{\sqrt{t}}$ | $\frac{LD^2}{\sqrt{t}} + \frac{\Delta_* \log(t)}{\sqrt{t}}$ | $\frac{L\Delta_*}{\mu^2}\frac{1}{t} + \frac{L^2 D^2}{\mu^2 t^2}$ | $\frac{B^2}{\mu^2}\frac{1}{t}$ |
| SPS* | $\frac{GD}{\sqrt{t}}^{(1)}$ Rem. 2.4 | $\frac{GD}{\sqrt{t}}$ Cor. 2.2 | $\frac{LD^2}{t} + \frac{\sqrt{L\Delta_*}D}{\sqrt{t}}$ Cor. 2.3 | $\frac{\Delta_*}{\mu^2}\frac{1}{t}^{(5)}$ Thm. C.6 | $\frac{B^2}{\mu^2}\frac{1}{t}$ Thm. C.6 |
| IAM (new) | $\frac{GD}{\sqrt{t}}^{(1)}$ Rem. 2.4 | $\frac{GD}{\sqrt{t}}$ Thm. 3.2 | $\frac{LD^2 \log(t+1)}{t} + \frac{\sqrt{L\Delta_*}D}{\sqrt{t}}$ Thm. 3.3 | ✗ | ✗ |

$^{(1)}$ The convex finite sum result assumes $\mathbb{E}_\xi[f_\xi] = \frac{1}{n}\sum_{i=1}^n f_i$

$^{(2)}$ SGD* denotes SGD where we can use all the global constants $D, G, L, \Delta_*$ and $\mu$ to set the step size. For the left to right, these results can be found in Garrigos & Gower (2023, Thm. 9.12), Gower et al. (2021, Thm. 4.1), Gower et al. (2019, Thm. 3.1), Lacoste-Julien et al. (2012, Sec. 3.2).

$^{(3)}$ Under the additional assumption that the iterates of DecSPS are bounded, we have from Orvieto et al. (2022) that DecSPS converges at a $\mathcal{O}\left(1/\sqrt{t}\right)$ rate in the $G$-Lipschitz and $L$-smooth setting.

$^{(4)}$ The paper claims an $\mathcal{O}\left(\log(t)/t\right)$ anytime rate is possible, but does not give the explicit proof or constants.

$^{(5)}$ Here we have an anytime rate only for $t \geq \mathcal{O}\left(\frac{L}{\mu}\right)$

interpolation where $f_\xi(x_*)$ can be known or reasonably approximated, such as model distillation which we consider in Section 4.1.

As a secondary objective of our work, we also present IAM (Iterate Averaging Adaptive method), a variant of SPS* with momentum. We prove that in the smooth and Lipschitz setting the *last* iterate of IAM converges as fast as the *average* iterate of SPS*. As the last iterate is usually more relevant in practice, this is the first time that a version of SPS with momentum has some theoretical advantage.

## 1.2 RELATED WORK AND CONTRIBUTIONS

Here we detail some of the related work and contrast it to our contributions. For a more extended related work section, please see Section A.

**The stochastic Polyak step size.** The current research into the stochastic Polyak step size was kick-started by the ALI-G method (Berrada et al., 2020) and SPS$_{\max}$ (Loizou et al., 2021). Both ALI-G and SPS$_{\max}$ offered a practical stochastic variant of the Polyak step size with strong empirical results to support their use. In terms of convergence theory, for smooth and convex functions, Loizou et al. (2021) showed that SPS$_{\max}$ converges to a neighborhood of the solution. To enforce that SPS$_{\max}$ does converge in the smooth setting, Orvieto et al. (2022) proposed the DecSPS method that combines SPS$_{\max}$ with a decreasing step size sequence, and show that if the stochastic loss functions are strongly convex and smooth, then suboptimality converges at a rate of $\mathcal{O}(1/\sqrt{T})$, where $T$ is the number of iterations. This rate is slower than SGD in the same setting, which is $\mathcal{O}(1/T)$.

As for SPS*, Garrigos et al. (2023) showed that it converges with the optimal rate in the Lipschitz non-smooth setting. Convergence in the smooth setting was shown by Garrigos et al. (2023); Gower et al. (2021), but only under interpolation.

A proximal version of SPS was introduced by Schaipp et al. (2023) to handle regularization terms. More recently, a new variant of SPS called NGN was introduced by Orvieto & Xiao (2024) for specifically non-negative functions. NGN uses a combination of Gauss-Newton and truncation to introduce a dampened version of the Polyak step sizes. Though NGN also converges to a neighborhood of the solution for smooth functions, Orvieto & Xiao (2024) prove a $\mathcal{O}(1/\sqrt{T})$ and $\mathcal{O}(1/T)$ complexity for convex and strongly convex functions, respectively. Orvieto & Xiao (2024) also give a $\mathcal{O}(\log(T)/\sqrt{T})$ anytime result in the smooth and convex setting.

*Contributions.* We present a unifying anytime convergence in the smooth and non-smooth setting in Theorem 2.1 for SPS*. Besides convexity, Theorem 2.1 only makes local assumptions and thus applies to a broader class of functions as compared to prior results. We then specialize this result into the locally Lipschitz and locally smooth setting in Corollary 2.2 and Corollary 2.3, respectively. Our proof also leverages a new trick, where we explicitly invert a convex monotone function (Lemma C.1). We show how this trick is used in a sketch of the proof of Theorem 2.1 in Section C.2. Finally, our convergence result in the smooth setting in Corollary 2.3 is the first $\mathcal{O}(1/\sqrt{T})$ anytime result which benefits from interpolation (see details in Section E).

**Stochastic Polyak with momentum.**  In the stochastic setting, some very recent works have considered different ways of blending SPS with momentum (Schaipp et al., 2024; Wang et al., 2023). The first analysis of a variant of SPS with momentum was developed by Wang et al. (2023). Their ALR-SMAG method is the result of choosing a learning rate that minimizes a particular upper bound on $\|x_{t+1} - x_*\|$ for the iterates of momentum or heavy-ball. The current analysis for ALR-SMAG shows that it has a slower convergence as compared to SPS unless $\beta_t = 0$, which corresponds to using no momentum. Another recent approach that combines SPS with momentum is proposed by Oikonomou & Loizou (2024), introducing MomSPSmax and its variants, MomDecSPS and MomAdaSPS. These step sizes guarantee convergence in the stochastic setting without relying on the interpolation condition. Instead, they assume in addition that the iterates remain bounded. Specifically, MomSPSmax achieves an $\mathcal{O}(1/t)$ convergence rate to a neighborhood of the solution, while MomDecSPS and MomAdaSPS converge to the exact solution with a rate of $\mathcal{O}(1/\sqrt{t})$.

*Contributions.* We prove that the last iterate of our momentum variant of SPS (Algorithm 1) converges anytime in (i) the convex and *locally* Lipschitz case (see Theorem 3.2) and (ii) the *locally* smooth case (see Theorem 3.3). Furthermore, in the non-smooth setting, the convergence rate in Theorem 3.2 is *at least as fast* as the corresponding rate for SPS* in Corollary 2.2.

**Adaptive methods.**  Historically, line search procedures, such as Armijo line search (Armijo, 1966), used to be commonly employed to estimate the smoothness around the current point when the exact smoothness constant $L$ was not known. More recent works have shown that it is also possible to estimate the value of $L$ using the previously observed gradients (Malitsky & Mishchenko, 2020; Latafat et al., 2024).

In online learning, when the Lipschitz constant of the objective is known, coin-betting approaches (Orabona & Pál, 2016) can be used to adaptively estimate distances to a solution. When the Lipschitz constant is not known, one can either use restarts (Mhammedi & Koolen, 2020), which require a lot of extra work, or use a technique called hints (Cutkosky, 2019a), but the latter introduces even more hyperparameters.

AdaGrad (Streeter & McMahan, 2012; Duchi et al., 2011) and its variants offer an alternative by estimating the gradient magnitudes instead of estimating smoothness. These methods can be combined with momentum and achieve strong complexity results, but they either require bounded domain (Levy et al., 2018; Kavis et al., 2019) or are only studied in the deterministic setting (Li & Lan, 2023). Furthermore, most variants use step sizes that can only decrease over time, meaning they will not adapt if the problem curvature becomes flatter. This has been partially addressed by a series of new methods that have an increasing estimate of distances to the solution set (Defazio & Mishchenko, 2023; Ivgi et al., 2023; Khaled et al., 2023), but their stochastic guarantees are provided only for large batch sizes (Ivgi et al., 2023) or the interpolation setting. We compare to the most relevant of these works in Table 2.

*Contributions.* Our theoretical results show that the SPS* method is adaptive to the following settings and parameters: smoothness $(L)$, initial distance $(D)$, Lipschitz $(G)$ and strong convexity $(\mu)$. The precise definition of these parameters and constants are given later.

## 2 STOCHASTIC POLYAK STEP SIZE

Before giving our convergence proofs, we first will motivate SPS* as the step size that minimizes an upper bound on the distance to a minimizer. Suppose we are at iteration $t$, have drawn a batch of data $\xi_t$, and let $g_t := g_{\xi_t}(x_t)$ be the stochastic (sub)gradient evaluated at $x_t$. For short-hand we will also use $f_t := f_{\xi_t}$. Consider an iterate of SGD,

$$x_{t+1} = x_t - \gamma_t g_t,$$

where $\gamma_t > 0$ is the step size. The subgradient $g_t \in \partial f_t(x_t)$, by definition satisfies

$$f_t(x) \geq f_t(x_t) + \langle g_t, x - x_t \rangle, \quad \forall x \in \mathbb{R}^d. \tag{5}$$

Now consider the task of choosing $\gamma_t$ that brings $x_{t+1}$ as close as possible to the solution $x_*$. In general, this is impossible since we do not know $x_*$. However, we can minimize the upper bound

$$\|x_{t+1} - x_*\|^2 - \|x_t - x_*\|^2 = -2\gamma_t \langle g_t, x_t - x_* \rangle + \gamma_t^2 \|g_t\|^2 \leq -2\gamma_t(f_t(x_t) - f_t(x_*)) + \gamma_t^2 \|g_t\|^2$$

where we use (5) in the inequality. Minimizing the right-hand side under the constraint $\gamma_t \geq 0$ gives[1]

$$\gamma_t^{\text{SPS*}} = \frac{(f_t(x_t) - f_t(x_*))_+}{\|g_t\|^2}, \tag{6}$$

which together with SGD gives the SPS* method (2).

### 2.1 CONVERGENCE THEORY FOR CONVEX PROBLEMS

We now give in Theorem 2.1 our unifying convergence theorem for SPS*, that aside from convexity only assumes in (7) that the expected norm of the stochastic gradients verifies a certain bound within

$$\mathbb{B}_D(x_*) := \{x \in \mathbb{R}^d : \|x - x_*\| \leq D\}, \quad \text{where} \quad D := \|x_0 - x_*\|.$$

Because our proof makes use of a new technical lemma that may find uses elsewhere, we give a sketch of the proof in Appendix C.2. The full proof is also in Appendix C.3.

**Theorem 2.1** (Convergence of SPS*). Consider problem (1) and let $(x_t)_{t \geq 0}$ be the iterates of SPS* given by (2). Then the iterates are almost surely monotone:

$$\|x_{t+1} - x_*\|^2 \leq \|x_t - x_*\|^2 \quad \text{with probability 1.}$$

If there exists $A, B \geq 0$ with $A + B \neq 0$ and such that

$$\mathbb{E}_\xi \left[ \|g_\xi(x)\|^2 \right] \leq A(f(x) - f(x_*)) + B \tag{7}$$

for all $x \in \mathbb{B}_D(x_*)$, then the averaged iterates of SPS* $\bar{x}_T := \frac{1}{T} \sum_{t=0}^{T-1} x_t$ verify:

$$\mathbb{E} \left[ f(\bar{x}_T) - \inf f \right] \leq \frac{D^2 A}{T} + \sqrt{\frac{D^2 B}{T}}.$$

Next we specialize Theorem 2.1 to the non-smooth and smooth settings, where the local bound (7) is always true. The proof of the next two corollaries can be found in Appendix C.4 and C.5.

**Corollary 2.2** (Non-smooth setting). Consider problem (1), and assume that the losses $f_\xi$ are locally Lipschitz in expectation (see Definition B.8). In particular there exists $G \geq 0$ such that

$$\mathbb{E}_\xi \left[ \|g_\xi(x)\|^2 \right] \leq G^2, \tag{8}$$

---

[1]Note that in (6) we divide by the squared norm of the stochastic gradient, which could be equal to zero. This is only possible if $(f_t(x_t) - f_t(x_*))_+ = 0$, so we define $\gamma_t^{\text{SPS*}} := 0$ if $g_t = 0$.

for all $x \in \mathbb{B}_D(x_*)$. Then the averaged iterates of SPS* $\bar{x}_T := \frac{1}{T}\sum_{t=0}^{T-1} x_t$ verify:

$$\mathbb{E}\left[f(\bar{x}_T) - \inf f\right] \leq \frac{GD}{\sqrt{T}}.$$

**Corollary 2.3** (Smooth setting). Consider problem (1), and assume that the losses $f_\xi$ are uniformly locally smooth (see Definition B.16). In particular there exists $L \geq 0$ s.t.

$$\mathbb{E}_\xi\left[\|\nabla f_\xi(x)\|^2\right] \leq 2L\left(f(x) - \mathbb{E}_\xi\left[\inf f_\xi\right]\right), \tag{9}$$

for all $x \in \mathbb{B}_D(x_*)$. Then the averaged iterates of SPS* $\bar{x}_T := \frac{1}{T}\sum_{t=0}^{T-1} x_t$ verify, with $\Delta_* := \inf f - \mathbb{E}_\xi\left[\inf f_\xi\right]$:

$$\mathbb{E}\left[f(\bar{x}_T) - \inf f\right] \leq \frac{2LD^2}{T} + \frac{\sqrt{2L\Delta_*}D}{\sqrt{T}}.$$

For the non-smooth setting, it is typically assumed that the loss functions are *globally* Lipschitz, uniformly with respect to $\xi$, which in turn gives a global bound on the stochastic subgradients. Here instead we require very little: the convexity of our losses entails that $f_\xi$ is $G_\xi$-Lipschitz on $\mathbb{B}_D(x^*)$ (see Proposition B.3), so we only need to assume that the expectation $\mathbb{E}_\xi\left[G_\xi\right]$ is finite. An advantage of this local Lipschitz assumption is that it is *always true for finite sums* (see Proposition B.9). Another advantage of our local assumption is that it is compatible with strong convexity. Indeed there is no function which is both globally Lipschitz and strongly convex, see e.g. Garrigos & Gower (2023, Lem. 9.13). We provide an additional result analogous to Theorem 2.1 in the strongly convex setting in Appendix C.6.

Despite this additional generality, we achieve a $\mathcal{O}(1/\sqrt{T})$ convergence rate, which is the optimal lower bound for convex Lipschitz functions (Drori & Teboulle, 2016). This rate is currently only attainable by combining adaptive methods such as AdaGrad (Duchi et al., 2011) with knowledge of $|x_0 - x_|$ to set the learning rate or projection radius (Orabona, 2019). In contrast, our oracle requires knowledge of $f_\xi(x_*)$. A weaker version of Corollary 2.2 was established in Garrigos et al. (2023, Thm. 2.3), assuming globally Lipschitz losses $f_\xi$.

As for the smooth setting, it is typically assumed in the literature that the loss functions $f_\xi$ are *globally* smooth, uniformly with respect to $\xi$, see Gower et al. (2019; 2020; 2021). Our result instead requires much less: that the losses $f_\xi$ are *locally* smooth, and that their local smoothness constants are uniformly bounded with respect to $\xi$. We defer to Proposition B.18 in the appendix for a formal proof that such local smoothness implies (9). In particular, it is remarkable that our assumption is *always true for finite sums* of $C^2$ losses (see Proposition B.17).

Our smooth result in Corollary 2.3 is, as far as we know, the first $\mathcal{O}(1/\sqrt{T})$ anytime convergence rate for a stochastic variant of the Polyak step size, assuming only smoothness and convexity. Note that SGD has a $\mathcal{O}\left(\log(T)/\sqrt{T}\right)$ anytime rate in this setting, see Appendix E.3.

Another interesting aspect of the convergence rate in Corollary 2.2 is that it benefits from interpolation: when $\Delta_* = 0$ the convergence rate in Corollary 2.2 becomes $\mathcal{O}(1/T)$, which is the expected accelerated rate of SGD under interpolation (Vaswani et al., 2019). We are unaware of prior work that establishes an anytime rate of convergence that is adaptive to interpolation. As a comparison, we illustrate in Theorem E.1 in the appendix how the *complexity* of SGD can benefit from interpolation, to the price of knowing the value of the interpolation constant $\Delta_*$. We further contrast our rate to the best known anytime rate for SGD and SPS$_{\max}$ in Appendix E.3.

**Remark 2.4** (Finite sum). For finite-sum minimization $f = \frac{1}{m}\sum_{i=1}^m f_i$, our assumptions are drastically simplified. The local Lipschitz assumption in Corollary 2.2 is automatically true ; and the local smoothness assumption in Corollary 2.3 is true if the $f_i$ are $C^2$.

We have shown that SPS* has the optimal rate of convergence in the non-smooth setting, and a fast adaptive anytime rate in the smooth setting. This motivates us to think of SPS* as an idealized variant of the stochastic Polyak step size. In Appendix E.5 we show how several practical variants of the stochastic Polyak step size, that do not access $f_\xi(x_*)$, can be viewed as approximations of SPS*.

# 3 MOMENTUM AND THE ITERATE MOVING AVERAGE METHOD

SPS* is missing one important and practical ingredient: momentum. Furthermore, Theorem 2.1 holds for the average iterate, whereas the last iterate is often used in practice. Momentum is often presented as replacing the gradient with an exponential moving average of gradients as follows:

$$m_t = \beta_t m_{t-1} + g_t$$
$$x_{t+1} = x_t - \gamma_t m_t. \qquad (10)$$

where $\gamma_t > 0$ and $\beta_t \in [0, 1)$.

---

**Algorithm 1:** IAM:Iterate Averaging Adaptive Method

**Input** : $z_{-1} = x_0 \in \mathbb{R}^d$, $\lambda_t > 0$, $t \in [T]$

1 **for** $t = 0$ **to** $T - 1$ **do**

2 $\quad \eta_t = \dfrac{\left(f_t(x_t) - f_t(x_*) + \langle g_t, z_{t-1} - x_t \rangle\right)_+}{\|g_t\|^2}$

3 $\quad z_t = z_{t-1} - \eta_t g_t$

4 $\quad x_{t+1} = \dfrac{\lambda_{t+1}}{1+\lambda_{t+1}} x_t + \dfrac{1}{1+\lambda_{t+1}} z_t$

**Return** : $x_T$

---

To derive our momentum variant of SPS, we will make use of the equivalent reformulation given by Lines 3 and 4 in Algorithm 1 where $\eta_t, \lambda_t \geq 0$ are hyperparameters. Though not obvious, the $x_t$ iterates on Line 4 are equivalent to the $x_t$ iterates of Momentum (10) by choosing a particular mapping between $(\beta_t, \gamma_t)$ and $(\lambda_t, \eta_t)$, see Defazio & Gower (2021, Thm. 1) and Lemma D.1.

Inspired by both Wang et al. (2023) and Schaipp et al. (2024), we now choose the learning rate $\eta_t$ on Line 3 that minimizes an upper bound on $D_t := \|z_t - x_*\|^2$. First, expanding the squares gives

$$D_t = D_{t-1} - 2\eta_t \langle g_t, z_{t-1} - x_* \rangle + \eta_t^2 \|g_t\|^2.$$

We can now uppber bound the above by using convexity:

$$\langle g_t, z_{t-1} - x_* \rangle = \langle g_t, x_t - x_* \rangle + \langle g_t, z_{t-1} - x_t \rangle \geq f_t(x_t) - f_t(x_*) + \langle g_t, z_{t-1} - x_t \rangle.$$

With this bound we have that

$$D_t \leq D_{t-1} + \eta_t^2 \|g_t\|^2 - 2\eta_t \left[ f_t(x_t) - f_t(x_*) + \langle g_t, z_{t-1} - x_t \rangle \right]. \qquad (11)$$

Minimizing the right-hand side over $\eta_t \geq 0$ gives

$$\eta_t = \left[ f_t(x_t) - f_t(x_*) + \langle g_t, z_{t-1} - x_t \rangle \right]_+ \frac{1}{\|g_t\|^2}. \qquad (12)$$

Equation (12) states the learning rate of IAM (*Iterate Averaging Adaptive Method*), see Algorithm 1.

## 3.1 CONVERGENCE THEOREMS

The IAM method is also monotonically decreasing. This type of monotonicity for stochastic methods is very rare, with the only other example that we are aware of being SPS* (cf. Theorem 2.1).

**Lemma 3.1.** Let $f_\xi$ be convex for every $\xi$. The distances of iterates $z_t$ of Algorithm 1 to a solution $x_* \in \mathbb{R}^d$ decreases monotonically, that is, with probability one

$$\|z_t - x_*\|^2 \leq \|z_{t-1} - x_*\|^2 \leq \cdots \leq \|z_0 - x_*\|^2.$$

## 3.2 NON-SMOOTH SETTING

Our first result is for the setting setting where $f_\xi$ is non-smooth and $g_\xi$ denotes a subgradient of $f_\xi$.

**Theorem 3.2** (Non-smooth setting). Consider problem (1), and let $D := \|x_0 - x_*\|$. Assume that the losses $f_\xi$ are locally Lipschitz in expectation. In particular there exists $G \geq 0$ such that (8) holds for all $x \in \mathbb{B}_D(x_*)$. Let $B_f(x, y) := f(x) - f(y) - \langle \nabla f(y), x - y \rangle$. Then the iterates of Algorithm 1 (IAM) started from $x_0$ with $\lambda_t = t$ verify the *last iterate* bound

$$\mathbb{E}\left[ f(x_T) - f(x_*) \right] + \frac{1}{T+1} \sum_{t=1}^{T} t \, \mathbb{E}\left[ B_f(x_{t-1}, x_t) \right] \leq \frac{GD}{\sqrt{T+1}}.$$

This rate of convergence is the same as SPS* (Corollary 2.2), with two notable differences. First this rate holds for the last iterate, as opposed to the average of the iterates. Second, this rate for IAM can be faster than that of SPS* due to the additional Bregman divergence term.

Theorem 3.2 restricts the parameter choice of $\lambda_t = t$, which when translated back (See Appendix D.1 for details) to the momentum method (10) restricts the parameters $(\gamma_t, \beta_t)$ to $\beta_t = \frac{t}{t+1} \frac{\eta_{t-1}}{\eta_t}$ and $\gamma_t = \frac{\eta_t}{t+2}$ for all $t$. To allow for other parameter settings, we provide Thm. D.5 in the Appendix, which allows for any deceasing $(\lambda_t)_t$, but does not establish a last-iterate convergence.

### 3.3 SMOOTH SETTING

Consider the setting where the loss functions $f_\xi$ satisfy a local *smoothness* condition.

**Theorem 3.3** (Smooth setting). Consider problem (1), and let $D := \|x_0 - x_*\|$. Assume that the losses $f_\xi$ are uniformly locally smooth. In particular there exists $L \geq 0$ such that (9) holds for all $x \in \mathbb{B}_D(x_*)$. Then the iterates of Algorithm 1 (IAM) started from $x_0$ with $\lambda_t = t$ verify this *last iterate* bound, with $\Delta_* := \inf f - \mathbb{E}_\xi [\inf f_\xi]$:

$$\mathbb{E}\left[f(x_{T-1}) - f(x_*)\right] \leq \frac{2LD^2(\log(T)+1)}{T} + \frac{\sqrt{2L\Delta_*}D}{\sqrt{T}}.$$

Analogous to the SPS* result in Corollary 2.3, the above shows that IAM benefits from interpolation, since Theorem 3.3 gives a $\tilde{\mathcal{O}}(1/T)$ convergence in the case of interpolation ($\Delta_* = 0$). In contrast to SPS* the rate of convergence of IAM has an additional $\log(T + 1)$ on the non-dominant $\mathcal{O}(\frac{1}{T})$ term.

## 4 EXPERIMENTS

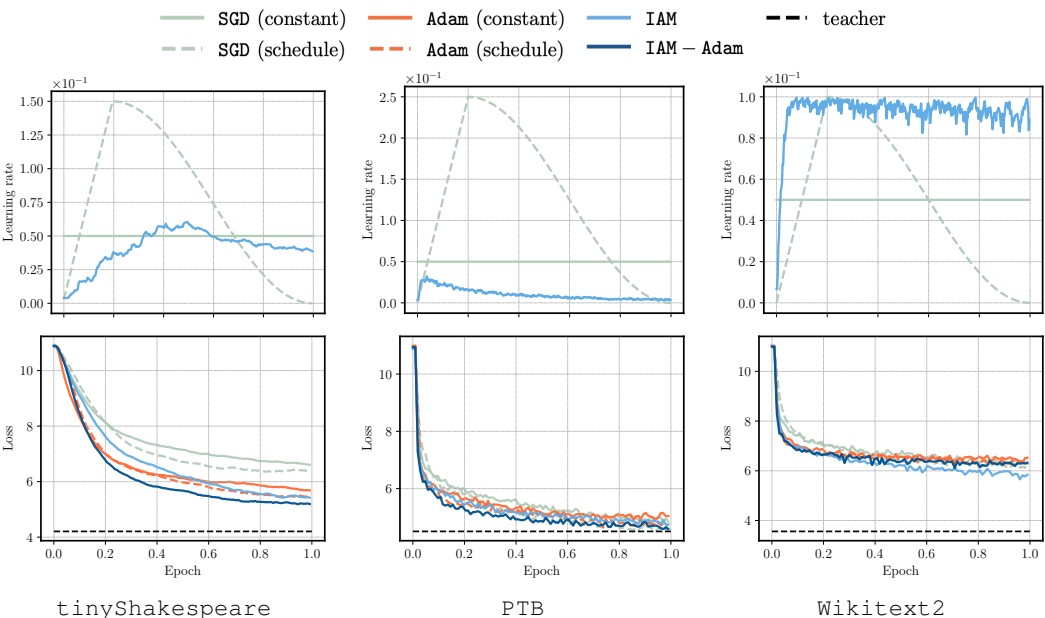

Figure 1: Distilling a teacher GPT2 on three datasets. Adaptive learning rate of IAM and learning rates of SGD (**top**) and cross-entropy training loss (**bottom**). Black line marks the average teacher loss.

Here we present several numerical results. First, we test the extent of our convergence theory for SPS* and IAM. According to Remark 2.4, both SPS* and IAM will converge for differentiable convex finite-sum problems, even when the loss is non-smooth and non-Lipschitz. We test this on Poisson regression in Appendix F.1, where we show that IAM converges to a loss value comparable

to `L-BFGS`, and to `SGD` with the best step size chosen from a grid. In Appendix F.2 we investigate how `IAM` behaves when $f_\xi(x_*)$ is wrongly specified (or guessed inaccurately).

### 4.1 BLACK-BOX MODEL DISTILLATION

Here we consider a variant of knowledge distillation where the goal is to train a small model (called *student*) while having access to a pretrained, large model (called *teacher*). The main idea we propose here is that, when training the student, the loss of teacher model for a given batch $\xi$ can be used as an approximation of $f_\xi(x_*)$.

For a given batch $\xi \sim \mathcal{P}$ from the training set of the student, denote by $f_\xi^s(x)$ the loss function[2] of the student with weights $x$ for batch $\xi$. Denote by $f_\xi^\tau$ the loss of the pretrained teacher model for the same batch. Since the teacher is a significantly larger and more expressive model, we can assume that even after training the student, its loss will not fall below $f_\xi^\tau$. Thus, we use $f_\xi^s(x_*) \approx f_\xi^\tau$ for the `IAM` method (Algorithm 1) to train the student.

Many variations of knowledge distillation have been proposed (Hinton et al., 2015; Beyer et al., 2022; Hsieh et al., 2023). The variant we present here is slightly different to previous works in that it requires only access to the batch loss of the teacher model (and not to the logits). We discuss this relationship in more detail in Appendix F.3.

We use three different datasets, `tinyShakespeare`, `PTB` and `Wikitext2`. As teacher model we use a pretrained `GPT2` model with 774M parameters (Radford et al., 2019; Wolf et al., 2020). The student models are much smaller `GPT2` architectures. All details are deferred to Appendix F.3. Our results are in Figure 1. We compare `IAM` and `IAM-Adam` (`IAM` with an `Adam` preconditioner, see Appendix D.6) to `SGD` and `Adam` with (i) constant learning rate, and (ii) *warmup+cosine-decay* schedule; tuning procedures are detailed in Appendix F.3. We trained for only one pass over the data (epoch), and consequently we used no weight decay.

We then did a complete sweep over all feasible learning rates to see the sensitivity of each method in Figure 2 on the `tinyShakespeare` data. Here we also include the Schedulefree method which was the winning entry for the MLCommons 2024 AlgoPerf Algorithmic Efficiency Challenge Self-Tuning track. In Figure 2 we can see that `IAM-Adam` achieve a better train and validation loss than all the other methods for every *learning* rate. The next best method `adamw-schedulefree` only achieved a similar (but slightly worst) loss for one choice of learning rate.

We find that both versions of `IAM` achieve the best resulting loss on all three problems. Consequently, when we are able to load a suitable pretrained teacher model, we find that `IAM` is able to efficiently train a student model without any hyperparameter tuning.

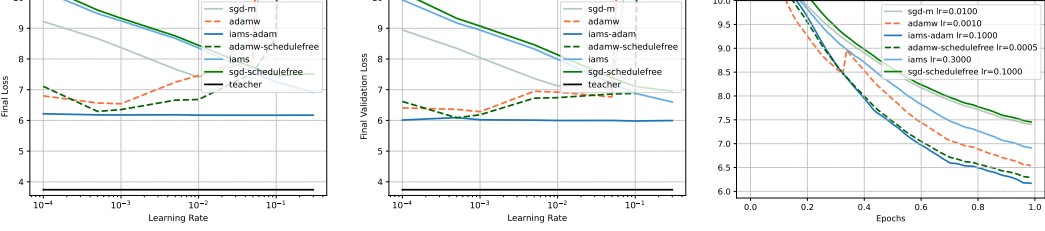

Figure 2: Sensitivity of the learning rate for distilling a teacher `GPT2` on the `tinyshakespeare` data set, Left: Final training loss over all base learning rates, Middle: Final test loss over all base learning rates, Right: The training trajectory for the best choice of learning rate for each method.

## 5 CONCLUSION

We show convergence theory for the stochastic Polyak step size assuming knowledge of the batch losses at a solution. A practical application of this setting is model distillation, where our proposed methods shows competitive performance without any hyperparameter tuning.

---

[2]This is usually the cross-entropy loss for the language modeling tasks we consider.

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

# CONTENTS

## A  EXTENDED BACKGROUND

**Polyak step size.**    The Polyak step size was first introduced by Polyak (1987) in the deterministic setting, where he also proved convergence for non-smooth and convex functions. Hazan & Kakade (2019) revisited the Polyak step size and showed that for the class of gradient descent methods (where we can only choose the step size), it has near-optimal convergence rate in the Lipschitz, smooth, and strongly convex setting. Furthermore, it is optimal without having access to any of the Lipschitz ($G$), smoothness ($L$), or strong convexity ($\mu$) parameters. Recently, the proof of convergence in the smooth setting has been generalized to a broader class of relatively smooth functions (Takezawa et al., 2024) and locally smooth functions (Richtárik et al., 2024). In the smooth and strongly convex setting, Barré et al. (2020) showed how to accelerate gradient descent with the Polyak step size, and without having access to the strong convexity parameter, but estimating it instead.

**The stochastic Polyak step size.**    The current research into the stochastic Polyak step size was kick-started by the `ALI-G` method (Berrada et al., 2020) and $SPS_{max}$ (Loizou et al., 2021). Both `ALI-G` and $SPS_{max}$ offered a practical stochastic variant of the Polyak step size with strong empirical results to support their use. In terms of convergence theory, for smooth and convex functions, Loizou et al. (2021) showed that $SPS_{max}$ converges to a neighborhood of the solution. To enforce that $SPS_{max}$ does converge in the smooth setting, Orvieto et al. (2022) proposed the `DecSPS` method that combines $SPS_{max}$ with a decreasing step size sequence, and show that if the stochastic loss functions are strongly convex and smooth, then suboptimality converges at a rate of $\mathcal{O}(1/\sqrt{T})$, where $T$ is the number of iterations. This rate is slower than `SGD` in the same setting, which is $\mathcal{O}(1/T)$.

As for `SPS`*, Garrigos et al. (2023) showed that it converges with the optimal rate in the Lipschitz non-smooth setting. Convergence in the smooth setting was shown by Garrigos et al. (2023); Gower et al. (2021), but only under interpolation.

A proximal version of `SPS` was introduced by Schaipp et al. (2023) to handle regularization terms. More recently, a new variant of `SPS` called `NGN` was introduced by Orvieto & Xiao (2024) for specifically non-negative functions. `NGN` uses a combination of Gauss-Newton and truncation to introduce a dampened version of the Polyak step sizes. Though `NGN` also converges to a neighborhood of the solution for smooth functions, Orvieto & Xiao (2024) prove a $\mathcal{O}(1/\sqrt{T})$ and $\mathcal{O}(1/T)$ complexity for convex and strongly convex functions, respectively. Orvieto & Xiao (2024) also give a $\mathcal{O}(\log(T)/\sqrt{T})$ anytime result in the smooth and convex setting.

**Momentum.**    Polyak (1964) introduced the momentum method through the heavy-ball viewpoint. In the deterministic setting, Polyak (1964) showed that it converges at an accelerated rate for strongly convex quadratic functions. Only rather recently, a global convergence was established for smooth and non-smooth functions without strong convexity (Ghadimi et al., 2015).

In the stochastic setting, there is little to no theoretical advantage for using momentum for SGD, unless we consider the specialized setting of minimizing a quadratic (Lee et al., 2024; Bollapragada et al., 2024). The main theoretical improvement from using momentum in the stochastic setting for general convex functions is that the last iterate $x_t$ of momentum converges at the same favourable rate as the average iterate of the SGD iterates (Sebbouh et al., 2021; Defazio & Gower, 2021). The analysis in Sebbouh et al. (2021) relies on an equivalent reformulation of momentum known as the *iterate averaging* viewpoint, which we also use in this work. Recent online-to-batch conversion techniques can also achieve the same rate of convergence of the last iterate of SGD without momentum, albeit with slightly worse constants (Cutkosky, 2019b). These online-to-batch techniques rely on monotonic step sizes, and thus are not applicable to Polyak-type step sizes.

Recently Zamani & Glineur (2024) gave a variant of Polyak with heavy-ball momentum that enjoys the optimal rate of convergence in the non-smooth full batch setting for the last iterate.

**Stochastic Polyak with momentum.** In the stochastic setting, some very recent works have considered different ways of blending SPS with momentum (Schaipp et al., 2024; Wang et al., 2023). The first analysis of a variant of SPS with momentum was developed by Wang et al. (2023). Their ALR-SMAG method is the result of choosing a learning rate that minimizes a particular upper bound on $\|x_{t+1} - x_*\|$ for the iterates of momentum or heavy-ball. The current analysis for ALR-SMAG shows that it has a slower convergence as compared to SPS unless $\beta_t = 0$, which corresponds to using no momentum. The same issue holds for the recently introduced MoMo method (Schaipp et al., 2024), which empirically reduces the tuning effort for the learning rate across many tasks, but theoretically has best bounds with no momentum, that is, when the method is equal to SPS. Another recent approach that combines SPS with momentum is proposed by Oikonomou & Loizou (2024), introducing MomSPSmax and its variants, MomDecSPS and MomAdaSPS. These step sizes guarantee convergence in the stochastic setting without relying on the interpolation condition. Instead, they assume in addition that the iterates remain bounded. Specifically, MomSPSmax achieves an $\mathcal{O}(1/t)$ convergence rate to a neighborhood of the solution, while MomDecSPS and MomAdaSPS converge to the exact solution with a rate of $\mathcal{O}(1/\sqrt{t})$.

**Adaptive methods.** Historically, line search procedures, such as Armijo line search (Armijo, 1966), used to be commonly employed to estimate the smoothness around the current point when the exact smoothness constant $L$ was not known. More recent works have shown that it is also possible to estimate the value of $L$ using the previously observed gradients (Malitsky & Mishchenko, 2020; Latafat et al., 2024). Furthermore, in the last decade, more line-search (Nesterov, 2014) and bisection (Carmon & Hinder, 2022) methods have been proposed that adapt simultaneously to smooth and non-smooth objectives. Unfortunately, most of the approaches either don't have strong guarantees in the stochastic case or require large batch sizes.

In online learning, when the Lipschitz constant of the objective is known, coin-betting approaches (Orabona & Pál, 2016) can be used to adaptively estimate distances to a solution. When the Lipschitz constant is not known, one can either use restarts (Mhammedi & Koolen, 2020), which require a lot of extra work, or use a technique called hints (Cutkosky, 2019a), but the latter introduces even more hyperparameters.

AdaGrad (Streeter & McMahan, 2012; Duchi et al., 2011) and its variants offer an alternative by estimating the gradient magnitudes instead of estimating smoothness. These methods can be combined with momentum and achieve strong complexity results, but they either require bounded domain (Levy et al., 2018; Kavis et al., 2019) or are only studied in the deterministic setting (Li & Lan, 2023). Furthermore, most variants use step sizes that can only decrease over time, meaning they will not adapt if the problem curvature becomes flatter. This has been partially addressed by a series of new methods that have an increasing estimate of distances to the solution set (Defazio & Mishchenko, 2023; Ivgi et al., 2023; Khaled et al., 2023), but their stochastic guarantees are provided only for large batch sizes (Ivgi et al., 2023) or the interpolation setting. We compare to the most relevant of these works in Table 2.

## B  BITS OF CONVEX ANALYSIS AND DETAILS ON OUR HYPOTHESES

Here we introduce and define some of the more technical bits of convex analysis we need throughout the paper. In particular we make precise the technical assumptions that we are making on the functions $f_\xi$, which correspond to the assumptions made in the Section 9 of Garrigos & Gower (2023).

### B.1  SUBGRADIENTS

Throughout our paper, we consider for every sampled data $\xi$ a loss function $f_\xi : \mathbb{R}^d \to \mathbb{R}$ taking finite values. We also always assume that $f_\xi$ is convex, which implies that it is continuous on $\mathbb{R}^d$ (see Peypouquet (2015, Prop. 3.5)). Nevertheless, we do not always assume that our loss functions $f_\xi$ are differentiable. For example, $f_\xi(x)$ could be defined with an absolute value, such as $f_\xi(x) = |w_\xi^\top x - y_\xi|$ where $w_\xi$ is a sample feature vector and $y_\xi$ a target value. In general, instead of using gradients we will making use of *subgradients*, which play a similar role.

**Definition B.1.** Let $f : \mathbb{R}^d \to \mathbb{R}$, and $x \in \mathbb{R}^d$. We say that $g \in \mathbb{R}^d$ is a **subgradient** of $f$ at $x \in \mathbb{R}^d$ if
$$\text{for every } y \in \mathbb{R}^d, \quad f(y) - f(x) - \langle g, y - x \rangle \geq 0.$$

Since our loss functions $f_\xi$ are convex and continuous, we are guaranteed that at every $x \in \mathbb{R}^d$, there exists some subgradient that we will note $g_\xi(x)$ (the existence of such subgradient is stated in Peypouquet (2015, Prop. 3.25) and Bauschke & Combettes (2017, Cor. 8.40)). In our proofs we will often need to take the expectation of these subgradients $g_\xi(x)$ with respect to $\xi$. To be able to do this, we must formally assume throughout that the function $\xi \mapsto g_\xi(x)$ is measurable for every $x \in \mathbb{R}^d$. This will for instance allow us to say that the expectation of $g_\xi(x)$ is a subgradient of $f$ at $x$ (see Garrigos & Gower (2023, Lem. 9.5)).

### B.2  LOCAL LIPSCHITZNESS

**Definition B.2.** We say that $f : \mathbb{R}^d \to \mathbb{R}^p$ is locally Lipschitz continuous if, for every $x \in \mathbb{R}^d$, there exists a neighbourhood $\mathbb{B}_\delta(x)$ and $G_x \geq 0$ such that $f$ is $G_x$-Lipschitz continuous on $\mathbb{B}_\delta(x)$.

As a matter of fact, convex functions are locally Lipschitz continuous.

**Proposition B.3.** If $f : \mathbb{R}^d \to \mathbb{R}$ is convex, then it is locally Lipschitz continuous.

*Proof.* See Corollary 8.41 in Bauschke & Combettes (2017). □

By compactness, this definition is equivalent to ask for Lipschitzness over any bounded set.

**Lemma B.4.** A function $f : \mathbb{R}^d \to \mathbb{R}^p$ is locally Lipschitz continuous if and only if for every bounded set $\mathcal{B} \subset \mathbb{R}^d$ there exists $G_\mathcal{B} \geq 0$ such that $f$ is $G_\mathcal{B}$-Lipschitz continuous on $\mathcal{B}$.

*Proof.* One implication is trivial. The other implication is standard. Assuming that $f$ is locally Lipschitz, and given any bounded set $\mathcal{B} \subset \mathbb{R}^d$, we can prove that $f$ is Lipschitz on $\mathcal{B}$. To see this, consider $\mathcal{K}$ the closure of $\mathcal{B}$ which is compact. From the local Lipschitz assumption we are given for every $x \in \mathcal{K}$ a certain $\delta_x$ and $G_x$ such that $f$ is $G_x$-Lipschitz over $\mathbb{B}_{\delta_x}(x)$. It is clear that
$$\mathcal{B} \subset \mathcal{K} \subset \bigcup_{x \in \mathcal{B}} \text{int } \mathbb{B}_{\delta_x}(x).$$

From the compactness of $\mathcal{K}$, there exists a finite number of points $x_1, \ldots, x_n$ such that
$$\mathcal{B} \subset \mathcal{K} \subset \bigcup_{i=1}^{n} \text{int } \mathbb{B}_{\delta_{x_i}}(x_i).$$

The conclusion follows by taking $G_\mathcal{B} = \max_{i=1,\ldots,n} G_{x_i}$. □

Lipschitzness of a convex function is tightly connected to the boundedness of its subgradients, as we see next.

**Definition B.5.** Let $f : \mathbb{R}^d \to \mathbb{R}$ be convex, and $G \geq 0$. We say that $f$ has $G$-bounded subgradients over $\mathcal{B} \subset \mathbb{R}^d$ if, for every $x \in \mathcal{B}$, for every $g \in \partial f(x)$, we have $\|g\| \leq G$.

**Lemma B.6.** Let $f : \mathbb{R}^d \to \mathbb{R}$ be convex. Let $\mathcal{U} \subset \mathbb{R}^d$ be any open set. Then the following is equivalent:

1. $f$ is $G$-Lipschitz over $\mathcal{U}$;

2. $f$ has $G$-bounded subgradients over $\mathcal{U}$.

*Proof.* This is a standard result, we provide a proof for completeness, which is taken from Proposition 16.20 in Bauschke & Combettes (2017). If $f$ is $G$-Lipschitz on $\mathcal{U}$, then for every $x \in \mathcal{U}$ and every $g \in \partial f(x)$ we can define $y = x + \gamma g$ for $\gamma > 0$ small enough so that $y \in \mathcal{U}$ (we exploit here the fact that $\mathcal{U}$ is open). Therefore we can write

$$\|g\|^2 = \langle g, \tfrac{y-x}{\gamma} \rangle = \tfrac{1}{\gamma} \langle g, y - x \rangle \leq \tfrac{1}{\gamma} G \|y - x\| = G \|g\|,$$

and conclude that $f$ has $G$-bounded subgradients on $\mathcal{U}$. If we assume that $G$-bounded subgradients on $\mathcal{U}$, then for every $x, y \in \mathcal{U}$ we can take $g \in \partial f(x)$ and write

$$|f(y) - f(x)| = f(y) - f(x) \leq \langle g, y - x \rangle \leq \|g\| \|y - x\| \leq G \|y - x\|.$$

Note that we assumed $|f(y) - f(x)| = f(y) - f(x)$ here, which is always possible by eventually swapping $x$ and $y$. This proves the claim. $\square$

**Proposition B.7.** Let $f : \mathbb{R}^d \to \mathbb{R}$ be convex and locally Lipschitz continuous, and let $\mathcal{B} \subset \mathbb{R}^d$ be any bounded set. Then there exists $G_{\mathcal{B}} \geq 0$ such that

$$\text{for every } x \in \mathcal{B}, \text{ for every } g \in \partial f(x), \|g\| \leq G_{\mathcal{B}}.$$

*Proof.* Given $\mathcal{B} \subset \mathbb{R}^d$ bounded, consider any $\mathcal{B}'$ which is open and contains $\mathcal{B}$, for instance $\mathcal{B}' = \text{int}(\mathcal{B} + \mathbb{B}(0, 1))$. Then it suffices to apply Lemma B.4 to obtain that $f$ is $G_{\mathcal{B}'}$-Lipschitz over $\mathcal{B}'$. We conclude with Lemma B.6 and the fact that $\mathcal{B} \subset \mathcal{B}'$. $\square$

### B.3 Local Lipschitzness in Expectation

**Definition B.8.** We say that the family $(f_\xi)$ is locally Lipschitz in expectation if for every bounded set $\mathcal{B} \subset \mathbb{R}^d$ there exists constants $G_{\mathcal{B}}(\xi) \geq 0$ such that $f_\xi$ is $G_{\mathcal{B}}(\xi)$-Lipschitz over $\mathcal{B}$, and moreover $\mathbb{E}_\xi\left[G_{\mathcal{B}}(\xi)^2\right] < +\infty$.

We recall that convex finite functions are always Lipschitz on bounded sets. So if the $f_\xi$ are convex, we already know that the constants $G_{\mathcal{B}}(\xi)$ exist, and this definition only require the expectation $\mathbb{E}_\xi\left[G_{\mathcal{B}}(\xi)^2\right]$ to be finite. This is always true for a finite family.

**Proposition B.9.** Let $f_1, \ldots, f_m$ be a finite family of convex functions from $\mathbb{R}^d$ to $\mathbb{R}$. Then this family is locally Lipschitz in expectation.

*Proof.* Each $f_i$ is locally Lipschitz continuous according to Proposition B.3. Therefore they are Lipschitz over any bounded set, according to Lemma B.4. Whatever Lipschitz constants $G_i$ we take, the expectation $\mathbb{E}_i\left[G_i^2\right]$ will be bounded by $\max\limits_{i=1,\ldots,m} G_i^2$ which is finite. $\square$

We conclude this section with the technical result at the core of Corollary 2.2, and which involves the measurable selection function $g_\xi : x \in \mathbb{R}^d \mapsto g_\xi(x) \in \partial f_\xi(x)$.

**Proposition B.10.** Suppose that the family of functions $(f_\xi)$ is locally Lipschitz in expectation, and convex. Let $\mathcal{B} \subset \mathbb{R}^d$ be bounded. Then there exists $G_\mathcal{B} \geq 0$ such that

$$\text{for every } x \in \mathcal{B}, \ \mathbb{E}_\xi \left[ \|g_\xi(x)\|^2 \right] \leq G_\mathcal{B}.$$

*Proof.* Given $\mathcal{B} \subset \mathbb{R}^d$ bounded, consider any $\mathcal{B}'$ which is open and contains $\mathcal{B}$, for instance $\mathcal{B}' = \text{int} (\mathcal{B} + \mathbb{B}(0,1))$. From the definition B.8, we know constants $G_\xi$ such that $f_\xi$ is $G_\xi$-Lipschitz on $\mathcal{B}'$, with $G_\mathcal{B} := \mathbb{E}_\xi [G_\xi] < +\infty$. From Lemma B.6 we know that $f_\xi$ has $G_\xi$-bounded subgradients over $\mathcal{B}'$, so $\|g_\xi(x)\| \leq G_\xi$. Taking the square and then expectation leads to the desired bound. □

### B.4 LOCAL SMOOTHNESS

We now give some technical details about locally smooth functions, which is the assumption made in Corollary 2.2.

**Definition B.11.** We say that $f : \mathbb{R}^d \to \mathbb{R}$ is locally smooth if it is differentiable and if $\nabla f : \mathbb{R}^d \to \mathbb{R}^d$ is locally Lipschitz continuous.

Note that this definition is equivalent to require $\nabla f$ to be Lipschitz continuous over any bounded subset of $\mathbb{R}^d$. A simple example of locally smooth functions are $C^2$ functions:

**Proposition B.12.** If $f : \mathbb{R}^d \to \mathbb{R}$ is of class $C^2$, then it is locally smooth.

*Proof.* The function $f$ being $C^2$ entails that it is differentiable. Moreover, for every open convex bounded set $\mathcal{U} \subset \mathbb{R}^d$, the mean value inequality says that for every $x, y \in \mathcal{U}$,

$$\|\nabla f(y) - \nabla f(x)\| \leq \sup_{z \in \mathcal{U}} \|\nabla^2 f(z)\| \|y - x\|.$$

Because $\nabla^2 f$ is supposed continuous, and because $\mathcal{U}$ is bounded, we know that $\sup_{z \in \mathcal{U}} \|\nabla^2 f(z)\| < +\infty$, which proves that $\nabla f$ is locally Lipschitz continuous. □

**Lemma B.13** (Local descent lemma). If $f$ is locally smooth, then for every bounded set $\mathcal{B} \subset \mathbb{R}^d$ there exists $L_\mathcal{B} \geq 0$ such that

$$\text{for all } x, y \in \mathcal{B}, \quad f(y) - f(x) - \langle \nabla f(x), y - x \rangle \leq \frac{L_\mathcal{B}}{2} \|y - x\|^2. \tag{13}$$

*Proof.* This is just a local version of the classic proof of the descent lemma, see e.g. Lemma 1.30 from Peypouquet (2015). Without loss of generality, we can assume that $\mathcal{B}$ is convex and compact (simply replace $\mathcal{B}$ with its closed convex hull). By compactness, we know that $\nabla f$ is Lipschitz continous on $\mathcal{B}$, for some constant $L_\mathcal{B} \geq 0$. We can then start the proof and fix $x, y \in \mathcal{B}$. Define the auxiliary function $g(t) = f((1-t)x + ty) - t\langle \nabla f(x), y - x \rangle$ for $t \in [0,1]$. It is differentiable and verifies

$$g(1) - g(0) = \int_0^1 g'(t) \, dt$$

which is equivalent, by definition of $g$, to

$$f(y) - f(x) - \langle \nabla f(x), y - x \rangle = \int_0^1 \langle \nabla f((1-t)x + ty) - \nabla f(x), y - x \rangle \, dt.$$

Now we use the Cauchy-Schwarz inequality, together with the Lipschitzness of $\nabla f$ (note that $z := (1-t)x + ty)$ belongs to $B$ which is convex!), to obtain

$$
\begin{aligned}
f(y) - f(x) &- \langle \nabla f(x), y - x \rangle \\
&\leq \int_0^1 \|\nabla f((1-t)x + ty) - \nabla f(x)\| \|y - x\| \, dt \\
&\leq \int_0^1 L_{\mathcal{B}} \|(1-t)x + ty - x\| \|y - x\| \, dt \\
&= \int_0^1 L_{\mathcal{B}} t \|y - x\|^2 \, dt \\
&= \frac{L_{\mathcal{B}}}{2} \|y - x\|^2.
\end{aligned}
$$

□

Convex locally smooth functions verify locally the following cocoercivity inequality:

**Proposition B.14.** If $f : \mathbb{R}^d \to \mathbb{R}$ is locally smooth and convex, then for every bounded set $\mathcal{B} \subset \mathbb{R}^d$ there exists $L_{\mathcal{B}} > 0$ such that

$$
\text{for all } y, x \in \mathcal{B}, \quad \frac{1}{2L_{\mathcal{B}}} \|\nabla f(y) - \nabla f(x)\|^2 \leq f(y) - f(x) - \langle \nabla f(x), y - x \rangle.
$$

*Proof.* This proof is just an adaptation of a classical result (see e.g. Garrigos & Gower (2023, Lem. 2.29)) by making use of additional local arguments. Here again, without loss of generality, we can assume that $\mathcal{B}$ is compact (see the proof of Lemma B.13). Let $x, y \in \mathcal{B}$ be fixed, and let $L_{\mathcal{B}}$ be the local smoothness constant provided by the local descent lemma B.13. Define $T : \mathbb{R}^d \times \mathbb{R}^d \times \mathbb{R} \to \mathbb{R}^d$ be the map defined by

$$
T(x, y, \gamma) = y - \gamma \nabla f(y) + \gamma \nabla f(x).
$$

Because $\nabla f$ is supposed continuous, we know that $T$ is continuous. Now we define

$$
K := \{ T(x, y, \gamma) \mid x \in \mathcal{B}, y \in \mathcal{B}, \gamma \in [0, \tfrac{1}{L_{\mathcal{B}}}] \} \subset \mathbb{R}^d.
$$

From our definitions it is clear that $K = T(\mathcal{B} \times \mathcal{B} \times [0, \tfrac{1}{L_{\mathcal{B}}}])$. In other words, it is the image of a compact set by a continuous function, which means that $K$ is compact. It is also clear that $K$ contains $\mathcal{B}$, simply observe that $T(x, x, 0) = x$. Now we can use again the local descent lemma B.13 to obtain that $f$ verifies (13) on $K$ with a constant $L_K$. Without loss of generality, we can assume that $L_K \geq L_{\mathcal{B}}$ (simply replace $L_K$ with $\max\{L_K, L_{\mathcal{B}}\}$). Now we can proceed with the classic arguments of the proof.

Let $x, y \in \mathcal{B}$ be fixed, and define $z := y - \frac{1}{L_K} \nabla f(y) + \frac{1}{L_K} \nabla f(x)$. By construction, $y \in \mathcal{B} \subset K$ and $z = T(x, y, \frac{1}{L_K}) \in K$. So now we can use the convexity of $f$ together with the descent lemma inequality on $K$ to write

$$
\begin{aligned}
f(x) - f(y) &= f(x) - f(z) + f(z) - f(y) \\
&\leq -\langle \nabla f(x), z - x \rangle + \langle \nabla f(y), z - y \rangle + \frac{L_K}{2} \|z - y\|^2.
\end{aligned}
$$

Now we use the fact that $z - x = y - x + \frac{1}{L_k}(\nabla f(x) - \nabla f(y))$ and $z - y = \frac{1}{L_k}(\nabla f(x) - \nabla f(y))$:

$$
\begin{aligned}
f(x) - f(y) &\leq -\langle \nabla f(x), y - x \rangle - \frac{1}{L_K} \|\nabla f(x) - \nabla f(y)\|^2 + \frac{1}{2L_K} \|\nabla f(x) - \nabla f(y)\|^2 \\
&= -\langle \nabla f(x), y - x \rangle - \frac{1}{2L_K} \|\nabla f(x) - \nabla f(y)\|^2.
\end{aligned}
$$

The above inequality is equivalent to the claimed one, which ends the proof. □

We conclude this section with a weaker result.

**Proposition B.15.** If $f : \mathbb{R}^d \to \mathbb{R}$ is locally smooth and bounded from below, then for every bounded set $\mathcal{B} \subset \mathbb{R}^d$ there exists $L_{\mathcal{B}} > 0$ such that

$$\text{for all } x \in \mathcal{B}, \quad \frac{1}{2L_{\mathcal{B}}}\|\nabla f(x)\|^2 \leq f(x) - \inf f.$$

*Proof.* Note that this result can be seen as an immediate consequence of Proposition B.14 by taking $y = x$ and $x$ as some minimizer of $f$. But we actually do not need to assume that $\operatorname{argmin} f \neq \emptyset$ for this result to be true. We highlight that the following proof is just an adaptation of a classical result (see e.g. Lemma 2.28 from Garrigos & Gower (2023)) by making use of additional local arguments. Here again, without loss of generality, we can assume that $\mathcal{B}$ is compact (see the proof of Lemma B.13). Let $L_{\mathcal{B}}$ be the local smoothness constant provided by the local descent lemma B.13. Define $T : \mathbb{R}^d \times \mathbb{R} \to \mathbb{R}^d$ be the map defined by

$$T(x, \gamma) = x - \gamma \nabla f(x).$$

Because $\nabla f$ is supposed continuous, we know that $T$ is continuous. Now we define

$$K := \{x - \gamma \nabla f(x) \mid x \in \mathcal{B}, \gamma \in [0, \tfrac{1}{L_{\mathcal{B}}}]\} \subset \mathbb{R}^d.$$

From our definitions it is clear that $K = T(\mathcal{B} \times [0, \tfrac{1}{L_{\mathcal{B}}}])$. In other words, it is the image of a compact set by a continuous function, which means that $K$ is compact. It is also clear that $K$ contains $\mathcal{B}$ (simply take $\gamma = 0$). Now we can use again the local descent lemma B.13 to obtain that $f$ verifies (13) with a constant $L_K$. Without loss of generality, we can assume that $L_K \geq L_{\mathcal{B}}$ (simply replace $L_K$ with $\max\{L_K, L_{\mathcal{B}}\}$). Now we can end the proof. Let $x \in B$ be fixed, and define $y := x - \tfrac{1}{L_K}\nabla f(x)$. By construction, $x \in \mathcal{B} \subset K$ and $y = T(x, \tfrac{1}{L_K}) \in K$. So we can use the descent lemma inequality on $K$ to obtain

$$f(x - \tfrac{1}{L_K}\nabla f(x)) - f(x) - \langle \nabla f(x), -\tfrac{1}{L_K}\nabla f(x) \rangle \leq \frac{L_K}{2}\|\tfrac{1}{L_K}\nabla f(x)\|^2.$$

Rewriting and reorganizing terms, we obtain further

$$f(x - \tfrac{1}{L_K}\nabla f(x)) - f(x) \leq -\frac{1}{2L_K}\|\nabla f(x)\|^2.$$

We obtain the desired result by observing that $f(x - \tfrac{1}{L_K}\nabla f(x)) \geq \inf f$. □

### B.5 Uniform Local Smoothness

**Definition B.16.** We say that the family $(f_\xi)$ is uniformly locally smooth if each function $f_\xi$ is differentiable, and for every bounded set $\mathcal{B} \subset \mathbb{R}^d$, there exists a constant $L_{\mathcal{B}} > 0$ independent of $\xi$ such that each gradient $\nabla f_\xi$ is $L_{\mathcal{B}}$-Lipschitz continuous over $\mathcal{B}$.

It is easy to see that any *finite* family of locally smooth functions is uniformly locally smooth: simply take the maximum of the local smoothness constants. In particular, any finite family of $C^2$ functions is uniformly locally smooth:

**Proposition B.17.** Let $f_1, \ldots, f_m$ be a finite family of convex $C^2$ functions from $\mathbb{R}^d$ to $\mathbb{R}$. Then this family is uniformly locally smooth.

*Proof.* Each $f_i$ is $C^2$ therefore it is locally smooth (see Proposition B.12). So, for every bounded set $\mathcal{B}$, there exists constants $L_i$ such that $f_i$ is $L_i$-smooth over $\mathcal{B}$. The conclusion follows after taking $L_{\mathcal{B}} := \max_{i=1,\ldots,m} L_i$. □

**Proposition B.18.** Suppose that the family of functions $(f_\xi)$ is uniformly locally smooth, and convex. Let $f = \mathbb{E}[f_\xi]$. Then, for every bounded set $\mathcal{B} \subset \mathbb{R}^d$, there exists $L_\mathcal{B} > 0$ such that

$$\text{for all } y, x \in \mathcal{B}, \quad \frac{1}{2L_\mathcal{B}}\mathbb{E}\left[\|\nabla f_\xi(y) - \nabla f_\xi(x)\|^2\right] \leq f(y) - f(x) - \langle \nabla f(x), y - x \rangle. \quad (14)$$

*Proof.* By definition of uniformly locally smooth functions, there exists $L_\mathcal{B} \geq 0$ such that each function $f_\xi$ is $L_\mathcal{B}$-smooth on $\mathcal{B}$, which means that we can use Proposition B.14 to write

$$\text{for all } y, x \in \mathcal{B}, \quad \frac{1}{2L_\mathcal{B}}\|\nabla f_\xi(x) - \nabla f_\xi(y)\|^2 \leq f_\xi(y) - f_\xi(x) - \langle \nabla f_\xi(x), y - x \rangle.$$

The conclusion follows after taking expectation with respect to $\xi$. $\qquad\square$

**Proposition B.19.** Suppose that the family of functions $(f_\xi)$ is uniformly locally smooth and bounded from below. Let $f = \mathbb{E}[f_\xi]$. Then, for every bounded set $\mathcal{B} \subset \mathbb{R}^d$, there exists $L_\mathcal{B} > 0$ such that

$$\text{for all } x \in \mathcal{B}, \quad \frac{1}{2L_\mathcal{B}}\mathbb{E}\left[\|\nabla f_\xi(x)\|^2\right] \leq f(x) - \mathbb{E}\left[\inf f_\xi\right]. \quad (15)$$

*Proof.* By definition of uniformly locally smooth functions, there exists $L_\mathcal{B} \geq 0$ such that each function $f_\xi$ is $L_\mathcal{B}$-smooth on $\mathcal{B}$, which means that we can use Proposition B.15 to write

$$\text{for all } x \in \mathcal{B}, \quad \frac{1}{2L_\mathcal{B}}\|\nabla f_\xi(x)\|^2 \leq f_\xi(x) - \inf f_\xi.$$

The conclusion follows after taking expectation with respect to $\xi$. $\qquad\square$

**Proposition B.20.** Suppose that the family of functions $(f_\xi)$ is uniformly locally smooth, and convex. Let $f = \mathbb{E}[f_\xi]$, and assume that $x_* \in \arg\min f \neq \emptyset$. Let $\mathcal{B} \subset \mathbb{R}^d$ be a bounded set, and let $L_\mathcal{B} > 0$ be the constant appearing in (14). Then

$$\text{for all } x \in \mathcal{B}, \quad \mathbb{E}\left[\|\nabla f_\xi(x)\|^2\right] \leq A(f(x) - \inf f) + B,$$

with $A = 4L_\mathcal{B}$ and $B = 2\sigma_*^2$, where $\sigma_*^2 = \mathbb{E}\left[\|\nabla f_\xi(x_*)\|^2\right]$.

*Proof.* For any $x \in \mathcal{B}$, write

$$\begin{aligned}
\mathbb{E}&\left[\|\nabla f_\xi(x)\|^2\right] \\
&= \mathbb{E}\left[\|\nabla f_\xi(x) - \nabla f_\xi(x_*) + \nabla f_\xi(x_*)\|^2\right] \\
&\leq 2\mathbb{E}\left[\|\nabla f_\xi(x) - \nabla f_\xi(x_*)\|^2\right] + 2\sigma_*^2 \\
&\leq 4L_\mathcal{B}(f(x) - \inf f) + 2\sigma_*^2,
\end{aligned}$$

where in the inequalities we first used $\|a + b\|^2 \leq 2\|a\|^2 + \|b\|^2$ and then (14) from Proposition B.14. $\qquad\square$

**Proposition B.21.** Suppose that the family of functions $(f_\xi)$ is uniformly locally smooth, and bounded from below. Let $f = \mathbb{E}[f_\xi]$, and assume that $x_* \in \arg\min f \neq \emptyset$. Let $\mathcal{B} \subset \mathbb{R}^d$ be a bounded set, and let $L_\mathcal{B} > 0$ be the constant appearing in (15). Then

$$\text{for all } x \in \mathcal{B}, \quad \mathbb{E}\left[\|\nabla f_\xi(x)\|^2\right] \leq A(f(x) - \inf f) + B,$$

with $A = 2L_\mathcal{B}$ and $B = 2L_\mathcal{B}\Delta_*$, where $\Delta_* = \inf f - \mathbb{E}\left[\inf f_\xi\right]$.

*Proof.* For any $x \in \mathcal{B}$, use (15) to write

$$\mathbb{E}\left[\|\nabla f_\xi(x)\|^2\right] \leq 2L_\mathcal{B}(f(x) - \mathbb{E}\left[\inf f_\xi\right]) = 2L_\mathcal{B}(f(x) - \inf f + \Delta_*).$$

$\qquad\square$

## C   PROOFS FOR SPS* : SGD WITH POLYAK STEPSIZES

### C.1   AUXILIARY LEMMAS

**Lemma C.1.** Let $A, B \geq 0$ which are not simultaneously zero. Let $\psi(t) = \frac{t^2}{At+B}$ be defined for $t \geq 0$. Then $\psi$ is convex and increasing over $[0, +\infty)$, and its inverse is $\psi^{-1}(s) = \frac{1}{2}(sA + \sqrt{s^2 A^2 + 4sB})$.

*Proof.* The function $\psi$ is twice differentiable over $[0, +\infty)$, and we can compute

$$\psi'(t) = \frac{At^2 + 2Bt}{(At+B)^2},$$

$$\psi''(t) = \frac{(2At + 2B)(At+B)^2 - 2(At^2 + 2Bt)(At+B)A}{(At+B)^4} = \frac{2B^2}{(At+B)^3}.$$

It is immediate to see that $\psi'$ and $\psi''$ are positive, from which we deduce that $\psi$ is convex and increasing.

Next, consider two cases. If $At + B = 0$, this implies $t = 0$ and $\psi(0) = 0$, thus $\psi^{-1}(0) = 0$. If, however, $At + B \neq 0$, for $s \geq 0$ it holds

$$\psi(t) = s \iff \frac{t^2}{At+B} = s \iff t^2 - Ast - Bs = 0.$$

The last equation has a unique nonnegative solution which is $t = \frac{1}{2}(sA + \sqrt{s^2 A^2 + 4sB})$, from which we deduce the expression for $\psi^{-1}$. $\qquad\square$

We will use the following lemma which is often used to study methods AdaGrad type methods.

**Lemma C.2.** Let $c_0, \ldots, c_k \geq 0$ be some non-negative numbers with $c_0 > 0$, and denote $S_t = \sum_{i=0}^t c_i$, then

$$\sqrt{S_t} \leq \sum_{k=0}^t \frac{c_k}{\sqrt{S_k}}. \tag{16}$$

*Proof.* The proof of the lemma can be found in various sources, for instance in the Appendix A of Levy et al. (2018), but since it is very short, we will provide it here for completeness as well. Observe that for any $\alpha \in [0, 1]$, it holds $\alpha \geq 1 - \sqrt{1 - \alpha}$. Substituting $\alpha = c_k / S_k \in [0, 1]$, we get

$$\frac{c_k}{S_k} \geq 1 - \sqrt{1 - \frac{c_k}{S_k}} \implies \frac{c_k}{\sqrt{S_k}} \geq \sqrt{S_k} - \sqrt{S_k - c_k} = \sqrt{S_k} - \sqrt{S_{k-1}}.$$

Summing the last inequality from $k = 1$ to $k = t$ and using $\sqrt{S_0} = \frac{c_0}{\sqrt{S_0}}$, we get the claim. $\qquad\square$

We also rely on the following result.

**Lemma C.3** (Extended Titu's Lemma). For any random variable $X$ and positive-valued random variable $Y$, it holds

$$\mathbb{E}\left[\frac{(X)_+^2}{Y}\right] \geq \frac{(\mathbb{E}[X])_+^2}{\mathbb{E}[Y]}. \tag{17}$$

In addition, for any numbers $a_0, \ldots, a_k$ and positive numbers $b_0, \ldots, b_k$, we have

$$\sum_{t=0}^k \frac{(a_t)_+^2}{b_t} \geq \frac{\left(\sum_{t=0}^k a_t\right)_+^2}{\sum_{t=0}^k b_t}. \tag{18}$$

*Proof.* The proof follows from applying Jensen's inequality to the function $\varphi(x, y) = (x)_+^2/y$. To prove that $\varphi$ is convex takes some work, and it is given in Lemma A.4 in Garrigos & Gower (2023). We also provide a different proof that $\varphi(x, y)$ is convex in the following Lemma C.4 by viewing $\varphi(x, y)$ as a perspective function. The discrete result (18) follows from applying (17) with uniform distribution over $\{a_0, \ldots, a_k\}$ and $\{b_0, \ldots, b_k\}$. $\qquad\square$

**Lemma C.4.** Consider the function $\varphi : \mathbb{R} \times \mathbb{R} \to \mathbb{R}, (x, y) \mapsto \varphi(x, y)$, where

$$\varphi(x, y) := \begin{cases} \frac{(x)_+^2}{y} & \text{if } y > 0, \\ 0 & \text{if } (y = 0) \wedge (x \leq 0), \\ +\infty & \text{else.} \end{cases} \tag{19}$$

Then, $\varphi$ is closed, proper and convex on $\mathbb{R} \times \mathbb{R}$.

*Proof.* Define the convex function $h(x) := (x)_+^2$. From Combettes (2017, Def. 2.1), it follows that $\varphi(x, y)$ defined as in (19) is the perspective function of $h$, that is, for $y > 0$ we have $\varphi(x, y) = yh(x/y)$; for $y = 0$, we compute $\lim_{\alpha \to \infty} \frac{(\alpha x)_+^2}{\alpha} = 0$ if $x \leq 0$ and $+\infty$ otherwise. The perspective functions of closed, proper, convex functions is convex itself (Combettes, 2017, Prop. 2.3). $\qquad\square$

## C.2 SKETCH PROOF OF THEOREM 2.1 ABOUT RATES OF SPS* UNDER ABSTRACT ASSUMPTIONS

Here we give a sketch of the proof of Theorem 2.1 so that we can better highlight the main ideas behind the proof, and the main novelty.

**Theorem 2.1** (Convergence of SPS*). Consider problem (1) and let $(x_t)_{t \geq 0}$ be the iterates of SPS* given by (2). Then the iterates are almost surely monotone:

$$\|x_{t+1} - x_*\|^2 \leq \|x_t - x_*\|^2 \qquad \text{with probability 1.}$$

If there exists $A, B \geq 0$ with $A + B \neq 0$ and such that

$$\mathbb{E}_\xi \left[ \|g_\xi(x)\|^2 \right] \leq A(f(x) - f(x_*)) + B \tag{7}$$

for all $x \in \mathbb{B}_D(x_*)$, then the averaged iterates of SPS* $\bar{x}_T := \frac{1}{T} \sum_{t=0}^{T-1} x_t$ verify:

$$\mathbb{E}\left[ f(\bar{x}_T) - \inf f \right] \leq \frac{D^2 A}{T} + \sqrt{\frac{D^2 B}{T}}.$$

*Proof Sketch.*

Plugging the SPS* step size (6) we can prove (see (22) and above for the formal argument) that

$$\frac{(f_t(x_t) - f_t(x_*))_+^2}{\|g_t\|^2} \leq \|x_t - x_*\|^2 - \|x_{t+1} - x_*\|^2.$$

Taking expectation conditioned on $x_t$, and using that the map $(z_1, z_2) \mapsto (z_1)_+^2/z_z$ is jointly convex on $\mathbb{R} \times \mathbb{R}_{\geq 0}$ (cf. Lemma C.4) together with Jensen's inequality, we get

$$\frac{(f(x_t) - f(x_*))_+^2}{\mathbb{E}_t\left[\|g_t\|^2\right]} \leq \|x_t - x_*\|^2 - \mathbb{E}_t\left[\|x_{t+1} - x_*\|^2\right].$$

We can then use our main assumption (7) to bound the denominator of the left hand side giving

$$\frac{(f(x_t) - f(x_*))^2}{A(f(x_t) - f(x_*)) + B} \leq \|x_t - x_*\|^2 - \mathbb{E}_t\left[\|x_{t+1} - x_*\|^2\right].$$

Taking expectation again, and averaging both sides over $t = 0, \ldots, T-1$ and telescoping we have that

$$\frac{1}{T} \sum_{t=0}^{T-1} \mathbb{E}\left[ \frac{(f(x_t) - f(x_*))^2}{A(f(x_t) - f(x_*)) + B} \right] \leq \frac{\|x_0 - x_*\|^2}{T} - \frac{\mathbb{E}\left[\|x_T - x_*\|^2\right]}{T} \leq \frac{\|x_0 - x_*\|^2}{T}.$$

The final step of the proof, and the main technical novelty, follows by defining the function $\psi(r) = \frac{r^2}{Ar+B}$ for $r \geq 0$, and noting that the left hand side of the above is equal to $\frac{1}{T} \sum_{t=0}^{T-1} \mathbb{E}\left[\psi(f(x_t) - f(x_*))\right]$. We then apply Lemma C.1 in the appendix that shows that $\psi$ is a convex monotone function. Being convex, we can bring the average over $t$ and the expectation inside $\psi$ giving

$$\psi(\mathbb{E}\left[f(\bar{x}_t) - f(x_*)\right]) \leq \frac{\|x_0 - x_*\|^2}{T}.$$

Finally, Lemma C.1 also proves that $\psi$ has an inverse given by

$$\psi^{-1}(s) = \tfrac{1}{2}(sA + \sqrt{s^2 A^2 + 4sB}).$$

Applying this inverse to both sides and using that $\psi^{-1}$ is monotone, gives the result. $\qquad\square$

Next we give the complete and detailed proof of Theorem 2.1.

### C.3 Proof of Theorem 2.1 about rates of SPS* under abstract assumptions

**Theorem 2.1** (Convergence of SPS*). Consider problem (1) and let $(x_t)_{t\geq 0}$ be the iterates of SPS* given by (2). Then the iterates are almost surely monotone:

$$\|x_{t+1} - x_*\|^2 \leq \|x_t - x_*\|^2 \qquad \text{with probability 1.}$$

If there exists $A, B \geq 0$ with $A + B \neq 0$ and such that

$$\mathbb{E}_\xi\left[\|g_\xi(x)\|^2\right] \leq A(f(x) - f(x_*)) + B \tag{7}$$

for all $x \in \mathbb{B}_D(x_*)$, then the averaged iterates of SPS* $\bar{x}_T := \frac{1}{T} \sum_{t=0}^{T-1} x_t$ verify:

$$\mathbb{E}\left[f(\bar{x}_T) - \inf f\right] \leq \frac{D^2 A}{T} + \sqrt{\frac{D^2 B}{T}}.$$

*Proof.* For short-hand we use $f_t := f_{\xi_t}$ to be the stochastic function sampled at iteration $t$. Expanding the squares, using the definition of the algorithm and using the convexity of $f_\xi$, we have that

$$\begin{aligned}
\|x_{t+1} - x_*\|^2 - \|x_t - x_*\|^2 &= 2\gamma_t^{\text{SPS*}}\langle g_t, x_* - x_t\rangle + (\gamma_t^{\text{SPS*}})^2\|g_t\|^2 \\
&\leq -2\gamma_t^{\text{SPS*}}(f_t(x_t) - f_t(x_*)) + (\gamma_t^{\text{SPS*}})^2\|g_t\|^2.
\end{aligned} \tag{20}$$

Let us now verify that

$$\|x_{t+1} - x_*\|^2 \leq \|x_t - x_*\|^2 \text{ with probability 1.} \tag{21}$$

If $g_t = 0$, then by definition we have that $\gamma_t^{\text{SPS*}} = 0$, thus the right-hand side of the above is zero, and (21) holds. Suppose instead that $g_t \neq 0$. Substituting in $\gamma_t^{\text{SPS*}}$ gives

$$\begin{aligned}
\|x_{t+1} - x_*\|^2 - \|x_t - x_*\|^2 &\leq \\
&-2\frac{(f_t(x_t) - f_t(x_*))_+}{\|g_t\|^2}(f_t(x_t) - f(x_*)) + \frac{(f_t(x_t) - f_t(x_*))_+^2}{\|g_t\|^2} \\
&= -\frac{(f_t(x_t) - f_t(x_*))_+^2}{\|g_t\|^2},
\end{aligned} \tag{22}$$

where in the last equality we use the identity $z(z)_+ = (z)_+^2$. Note that in both cases we obtained a nonpositive right-hand side, from which we deduce that (21) holds, that is, $(x_t)_{t\geq 0}$ is Fejér monotone.

Now, let $a_t := f_t(x_t) - f_t(x_*)$ and $b_t := \|g_t\|^2$, and define the function

$$\phi(a, b) = \begin{cases} \frac{(a)_+^2}{b} & \text{if } a \in \mathbb{R}, b > 0, \\ 0 & \text{if } a \leq 0, b = 0, \end{cases}$$

so that the previous inequality can be rewritten as

$$\phi(a_t, b_t) \leq \|x_t - x_*\|^2 - \|x_{t+1} - x_*\|^2. \tag{23}$$

Note that $\phi(a_t, b_t)$ is well-defined even in the case that $g_t = 0$. Indeed, the convexity of $f_t$ implies in this case that $x_t$ minimizes $f_t$, which means that $a_t \leq 0$ while $b_t = 0$. Our main trick is to use Jensen's inequality with regard to the function $\phi$ which is convex (see Lemma C.4 or the Appendix in Garrigos & Gower (2023) for a proof):

$$\phi(\mathbb{E}[a_t], \mathbb{E}[b_t]) \leq \mathbb{E}[\phi(a_t, b_t)] \leq \mathbb{E}[\|x_t - x_*\|^2] - \mathbb{E}[\|x_{t+1} - x_*\|^2]. \tag{24}$$

We can compute $\mathbb{E}[a_t] = \mathbb{E}[f_{\xi_t}(x_t) - f_{\xi_t}(x_*)] = \mathbb{E}[f(x_t) - \inf f]$ and $\mathbb{E}[b_t] = \mathbb{E}[\|g_t\|^2]$.

For the rest of the proof, we are going to use the fact that there exist two constants $A, B \geq 0$, which are not simultaneously zero, and such that (7) holds, that is

$$\mathbb{E}[\|g_\xi(x)\|^2] \leq A(f(x) - \inf f) + B, \text{ for every } x \in \mathbb{B}(x_*, D). \tag{25}$$

We are now going to inject this inequality (25) into (24). If $\mathbb{E}[\|g_t\|^2] \neq 0$, using the fact that $f(x_t) - \inf f \geq 0$ we obtain

$$\frac{\mathbb{E}[f(x_t) - \inf f]^2}{A\mathbb{E}[f(x_t) - \inf f] + B} \leq \phi(\mathbb{E}[a_t], \mathbb{E}[b_t]) \leq \mathbb{E}[\|x_t - x_*\|^2] - \mathbb{E}[\|x_{t+1} - x_*\|^2]. \tag{26}$$

Recall that we defined $\psi(r) = \frac{r^2}{Ar+B}$ for any $r \geq 0$. Let $r_t := \mathbb{E}[f(x_t) - \inf f]$. With this notation, the inequality (26) can be rewritten as

$$\psi(r_t) \leq \mathbb{E}[\|x_t - x_*\|^2] - \mathbb{E}[\|x_{t+1} - x_*\|^2]. \tag{27}$$

We observe that (27) remains true when $\mathbb{E}[\|g_t\|^2] = 0$. Indeed in this case, from the variance bound we have that

$$0 = \mathbb{E}[\|g_t\|^2] \geq \|\mathbb{E}[g_t]\|^2.$$

Furthermore it follows that $\mathbb{E}_t[g_t]$ is a subgradient of the full loss $f(x_t)$ (see Lemma 9.5 in Garrigos & Gower (2023)). Consequently $x_t$ minimizes $f$, meaning in this case that we would have $r_t = 0$, and so $\psi(r_t) = 0 = \phi(0, 0)$.

For the last part of this proof, we sum over $t = 0, \ldots, T-1$ and divide by $T$ to obtain, after telescoping terms:

$$\frac{1}{T} \sum_{t=0}^{T-1} \mathbb{E}[\psi(r_t)] \leq \frac{1}{T} \mathbb{E}[\|x_0 - x_*\|^2] - \frac{1}{T} \mathbb{E}[\|x_T - x_*\|^2] \leq \frac{D^2}{T}.$$

We now lower-bound the left-hand side term by using Jensen's inequality twice

$$\frac{1}{T} \sum_{t=0}^{T-1} \mathbb{E}[\psi(r_t)] \geq \psi\left(\mathbb{E}\left[\frac{1}{T} \sum_{t=0}^{T-1} r_t\right]\right) = \psi\left(\mathbb{E}\left[\frac{1}{T} \sum_{t=0}^{T-1} (f(x_t) - \inf f)\right]\right) \geq \psi(\mathbb{E}[f(\bar{x}_T) - \inf f]),$$

where in the first inequality we use the convexity of $\psi$, and in the second we use the convexity of $f$ together with the fact that $\psi$ is increasing, and we note the average of the iterates $\bar{x}_T := \frac{1}{T} \sum_{t=0}^{T-1} x_t$. The reader can look at Lemma C.1 for a proof that $\psi$ is convex and monotone. Combining the two previous inequalities, we obtain

$$\psi(\mathbb{E}[f(\bar{x}_T) - \inf f]) \leq \frac{D^2}{T}.$$

Since $\psi$ is increasing on $[0, +\infty)$, it has an inverse which is also increasing. Applying the inverse of $\psi$ on both sides gives

$$\mathbb{E}[f(\bar{x}_T) - \inf f] \leq \psi^{-1}\left(\frac{D^2}{T}\right).$$

From Lemma C.1 we know that $\psi^{-1}(s) = \frac{1}{2}(sA + \sqrt{s^2 A^2 + 4sB})$, and using the sublinearity of the square root we further have

$$\psi^{-1}(s) \leq \frac{1}{2}(sA + \sqrt{s^2 A^2} + \sqrt{4sB}) = sA + \sqrt{sB}. \tag{28}$$

From this we finally obtain the desired bound. $\qquad \square$

### C.4 PROOF OF COROLLARY 2.2 ABOUT RATES OF SPS* IN THE NONSMOOTH SETTING

**Corollary 2.2** (Non-smooth setting)**.** Consider problem (1), and assume that the losses $f_\xi$ are locally Lipschitz in expectation (see Definition B.8). In particular there exists $G \geq 0$ such that

$$\mathbb{E}_\xi \left[ \|g_\xi(x)\|^2 \right] \leq G^2, \tag{8}$$

for all $x \in \mathbb{B}_D(x_*)$. Then the averaged iterates of SPS* $\bar{x}_T := \frac{1}{T} \sum_{t=0}^{T-1} x_t$ verify:

$$\mathbb{E}\left[ f(\bar{x}_T) - \inf f \right] \leq \frac{GD}{\sqrt{T}}.$$

*Proof.* Because the losses are locally Lipschitz in expectation, we know from Proposition B.10 that (8) holds true for some $G < +\infty$. This means that (7) holds with $A = 0$ and $B = G^2$. Thus the result follows by plugging in these constants into the rates of Theorem 2.1. □

### C.5 PROOF OF COROLLARY 2.3 ABOUT RATES OF SPS* IN THE SMOOTH SETTING

**Corollary 2.3** (Smooth setting)**.** Consider problem (1), and assume that the losses $f_\xi$ are uniformly locally smooth (see Definition B.16). In particular there exists $L \geq 0$ s.t.

$$\mathbb{E}_\xi \left[ \|\nabla f_\xi(x)\|^2 \right] \leq 2L\big( f(x) - \mathbb{E}_\xi \left[ \inf f_\xi \right] \big), \tag{9}$$

for all $x \in \mathbb{B}_D(x_*)$. Then the averaged iterates of SPS* $\bar{x}_T := \frac{1}{T} \sum_{t=0}^{T-1} x_t$ verify, with $\Delta_* := \inf f - \mathbb{E}_\xi \left[ \inf f_\xi \right]$:

$$\mathbb{E}\left[ f(\bar{x}_T) - \inf f \right] \leq \frac{2LD^2}{T} + \frac{\sqrt{2L\Delta_*}D}{\sqrt{T}}.$$

*Proof.* Assuming the losses to be uniformly locally smooth ensures that we can use Proposition B.19, and obtain that (9) is true. Then we use Proposition B.21, which guarantees that (7) holds with $A = 2L$ and $B = 2L\Delta_*$, and we conclude by plugging in these constants into the rates of Theorem 2.1. □

We provide below an alternative result which makes use of a different interpolation constant. Instead of relying on $\Delta_* = \inf f - \mathbb{E}_\xi \left[ \inf f_\xi \right]$, we use $\sigma_*^2 = \mathbb{E}_\xi \left[ \|\nabla f_\xi(x_*)\|^2 \right]$. There are a couple of connections between those constants. First they are both interpolation constants, in the sense that they are nonnegative, and equal to zero if and only if interpolation holds (see Section 4.3 in Garrigos & Gower (2023)). Second, the former dominates the latter through the inequality $\sigma_*^2 \leq 2L\Delta_*$ for some $L \geq 0$ (see Proposition B.19 with $x = x^*$). In particular it follows from our assumption $\mathbb{E}_\xi \left[ \inf f_\xi \right] > -\infty$ that $\sigma_*^2$ is finite. Actually one could argue that assuming $\sigma_*^2 < +\infty$ is an even weaker assumption than $\mathbb{E}_\xi \left[ \inf f_\xi \right] > -\infty$.

**Theorem C.5** (Smooth setting, variant with $\sigma_*^2$)**.** Consider problem (1), and assume that the losses $f_\xi$ are uniformly locally smooth (see Definition B.16). In particular there exists $L \geq 0$ s.t.

$$\mathbb{E}_\xi \left[ \|\nabla f_\xi(x) - \nabla f_\xi(x^*)\|^2 \right] \leq 2L\big( f(x) - \inf f \big), \tag{29}$$

for all $x \in \mathbb{B}_D(x_*)$. Then the averaged iterates $\bar{x}_T := \frac{1}{T} \sum_{t=0}^{T-1} x_t$ of SPS* verify, with $\sigma_*^2 := \mathbb{E}_\xi \left[ \|\nabla f_\xi(x_*)\|^2 \right]$:

$$\mathbb{E}\left[ f(\bar{x}_T) - \inf f \right] \leq \frac{4LD^2}{T} + \frac{\sqrt{2}D\sigma_*}{\sqrt{T}}.$$

*Proof.* Assuming the losses to be uniformly locally smooth ensures that we can use Proposition B.18 with $y = x$ and $x = x_*$, and obtain that (29) is true. Then we use Proposition B.20, which guarantees that (7) holds with $A = 4L$ and $B = 2\sigma_*^2$, and we conclude by plugging in these constants into the rates of Theorem 2.1. □

A last comment: from the inequality $\sigma_*^2 \leq 2L\Delta_*$, one could think that it is enough to prove results using $\sigma_*^2$ and then use this inequality to transform it into a result using $\Delta_*$. But the situation is not so simple. Indeed, applying this inequality to the bound of Theorem C.5 gives

$$\mathbb{E}\left[ f(\bar{x}_T) - \inf f \right] \leq \frac{4LD^2}{T} + \frac{\sqrt{4L\Delta_*}D}{\sqrt{T}}.$$

We see that the multiplicative constants are less good (by a factor 2) than the ones from Corollary 2.3. We claim that this is due to the use of two different smoothness inequalities (one for proving Theorem C.5 and one for using the inequality $\sigma_*^2 \leq 2L\Delta_*$), while the direct proof of Corollary 2.3 uses smoothness only once.

### C.6 Additional Result: Convergence Rates of SPS* with Local Strongly Convexity

If we assume our loss functions is (locally) strongly convex, then we can improve the rate of convergence of SPS* from $\mathcal{O}\left(1/\sqrt{t}\right)$ to $\mathcal{O}\left(1/t\right)$.

**Theorem C.6** (Convergence of SPS*). Consider (1) and let the iterates $(x_t)_{t\geq 0}$ be given by (2), and let $D := \|x_0 - x_*\|$. Assume that $f_\xi$ is convex for any $\xi$. Let $f(x)$ be convex and satisfy the $\mu$–quadratic growth bound

$$\frac{\mu}{2}\|x - x_*\|^2 \leq f(x) - \inf f, \quad \text{for every } x \in \mathbb{B}(x_*, D) \tag{30}$$

and the expected smoothness bound

$$\mathbb{E}_\xi\left[\|g_\xi(x)\|^2\right] \leq A(f(x) - \inf f) + B, \quad \text{for every } x \in \mathbb{B}(x_*, D). \tag{31}$$

Let $T_0 := \frac{4A}{\mu}\log\left(\frac{D^2\mu^2}{16B}\right)$. It follows that

$$\mathbb{E}\|x_t - x_*\|^2 \leq \frac{16B}{\mu^2}\frac{1}{t+1-T_0}, \quad \forall t \geq \frac{2A}{\mu}\left(2\log\left(\frac{D^2\mu^2}{16B}\right)+1\right). \tag{32}$$

In the non-smooth setting where $A = 0$ and $B = G^2$ we get

$$\mathbb{E}\left[\|x_t - x_*\|^2\right] \leq \frac{16G^2}{\mu^2}\frac{1}{t+1}, \quad \text{for } t \geq 0. \tag{33}$$

This matches the rate given by Pedregosa & Schaipp (2023) for the finite sum setting upto a factor of 4.

In the smooth setting where $A = 4L$ and $B = \sigma_*^2$ we get

$$\mathbb{E}\left[\|x_t - x_*\|^2\right] \leq \frac{64\sigma_*^2}{\mu^2}\frac{1}{t+1-T_0}, \quad \text{for } t \geq \frac{8L}{\mu}\left(2\log\left(\frac{D^2\mu^2}{16\sigma_*^2}\right)+1\right). \tag{34}$$

*Proof.* Let $\delta_t := \mathbb{E}\left[\|x_t - x_*\|^2\right]$. We start the proof from (27), which we repeat here for convenience:

$$\psi(r_t) \leq \delta_t - \delta_{t+1}, \tag{35}$$

where $r_t = \mathbb{E}\left[f(x_t) - f(x_*)\right]$ and $\psi(r) := \frac{r^2}{Ar+B}$ for $r \geq 0$. Due to the monotonicity of the iterates (recall (21)) we have that $\delta_t - \delta_{t+1} \geq 0$. Applying Lemma C.1 together with (28) gives

$$\mathbb{E}\left[f(x_t) - f(x_*)\right] \leq \psi^{-1}(\delta_t - \delta_{t+1})$$
$$\leq A(\delta_t - \delta_{t+1}) + \sqrt{B(\delta_t - \delta_{t+1})}.$$

Using the quadratic growth bound $\frac{\mu}{2}\|x_t - x_*\|^2 \leq f(x_t) - f(x_*)$ gives

$$\frac{\mu}{2}\delta_t \leq A\left(\delta_t - \delta_{t+1}\right) + \sqrt{B\left(\delta_t - \delta_{t+1}\right)}. \tag{36}$$

Our proofs will consider two cases by comparing the two terms on the right hand side of (36). To this end, note that

$$A\left(\delta_t - \delta_{t+1}\right) \leq \sqrt{B\left(\delta_t - \delta_{t+1}\right)} \quad \Longleftrightarrow \quad \delta_t - \delta_{t+1} \leq \frac{B}{A^2}. \tag{37}$$

The remainder of the proof is divided into two parts. First we show that for $t_0 := \lceil \frac{4A}{\mu} \log \left( \frac{D^2 \mu^2}{16B} \right) \rceil$, we have that $\delta_t \leq \frac{16B}{\mu^2}$. For the second part we prove by induction that for $t \geq t_0$ $\delta_{t+1} \leq \frac{16B}{\mu^2} \frac{1}{t+1}$, where the first part will serve as the base case of the induction.

*Base case:* First we prove that for all $t \geq \frac{4A}{\mu} \log \left( \frac{D^2 \mu^2}{16B} \right)$ we have that $\delta_t \leq \frac{16B}{\mu^2}$. We divide this proof also into two cases based on the comparison (37). If $\delta_t - \delta_{t+1} \leq \frac{B}{A^2}$ for any $t < \frac{4A}{\mu} \log \left( \frac{D^2 \mu^2}{16B} \right)$ then by (36) and (37) we have that

$$\frac{\mu}{2} \delta_t \leq A \left( \delta_t - \delta_{t+1} \right) + \sqrt{B \left( \delta_t - \delta_{t+1} \right)} \ \leq 2\sqrt{B \left( \delta_t - \delta_{t+1} \right)} \leq 2\sqrt{B \delta_t} \quad \Longrightarrow$$

$$\delta_t \leq \frac{16B}{\mu^2}, \tag{38}$$

which would prove our result.

Alternatively, suppose that $\delta_t - \delta_{t+1} \geq \frac{B}{A^2}$ for every $t \leq \frac{4A}{\mu} \log \left( \frac{D^2 \mu^2}{16B} \right)$. By (36) and (37) we have that

$$\frac{\mu}{2} \delta_t \leq A \left( \delta_t - \delta_{t+1} \right) + \sqrt{B \left( \delta_t - \delta_{t+1} \right)} \ \leq 2A \left( \delta_t - \delta_{t+1} \right). \tag{39}$$

Re-arranging the above gives

$$\delta_{t+1} \leq \left( 1 - \frac{\mu}{4A} \right) \delta_t. \tag{40}$$

Unrolling this for every $t \leq \frac{4A}{\mu} \log \left( \frac{D^2 \mu^2}{16B} \right)$ gives

$$\delta_t \leq \left( 1 - \frac{\mu}{4A} \right)^t \delta_0. \tag{41}$$

It now follows by taking logarithm and using standard techniques (for example Lemma A.2 in Garrigos & Gower (2023)) that

$$t \geq \frac{4A}{\mu} \log \left( \frac{D^2 \mu^2}{16B} \right) \quad \Longrightarrow \quad \delta_t \leq \left( 1 - \frac{\mu}{4A} \right)^t \delta_0 \leq \frac{16B}{\mu^2}.$$

*Induction step:* Now, for ease of notation, let us re-name our iterates so that $\delta_0$ is the first iterate for which $\delta_0 \leq \frac{16B}{\mu^2}$.

If $\delta_t - \delta_{t+1} \leq \frac{B}{A^2}$ then by (36) and (37) we have that

$$\frac{\mu}{2} \delta_t \leq A \left( \delta_t - \delta_{t+1} \right) + \sqrt{B \left( \delta_t - \delta_{t+1} \right)} \ \leq 2\sqrt{B \left( \delta_t - \delta_{t+1} \right)} \quad \Leftrightarrow$$

$$\frac{\mu^2}{4} \delta_t^2 \leq 4B \left( \delta_t - \delta_{t+1} \right) \quad \Leftrightarrow$$

$$\delta_{t+1} \leq (1 - \frac{\mu^2}{16B} \delta_t) \delta_t. \tag{42}$$

Let $a_t = \frac{\mu^2}{16B} \delta_t$. Multiplying both sides of (42) by $\frac{\mu^2}{16B}$ and using the induction hypothesis

$$a_t = \frac{\mu^2}{16B} \delta_t \leq \frac{\mu^2}{16B} \frac{16B}{\mu^2} \frac{1}{t+1} = \frac{1}{t+1}$$

gives

$$a_{t+1} \leq (1 - a_t) a_t \leq \max_{x \in [0, \frac{1}{t+1}]} (1-x)x = \left( 1 - \frac{1}{t+1} \right) \frac{1}{t+1} \leq \frac{1}{t+2}.$$

Alternatively if $\delta_t - \delta_{t+1} \geq \frac{B}{A^2}$ then by (36) and (37) we have that

$$\frac{\mu}{2} \delta_t \leq A \left( \delta_t - \delta_{t+1} \right) + \sqrt{B \left( \delta_t - \delta_{t+1} \right)} \ \leq 2A \left( \delta_t - \delta_{t+1} \right). \tag{43}$$

Re-arranging the above gives

$$\delta_{t+1} \le \left(1 - \frac{\mu}{4A}\right)\delta_t.$$

Using the induction hypothesis and $t \ge \frac{2A}{\mu}$ we have that

$$\delta_{t+1} \le \left(1 - \frac{\mu}{4A}\right)\delta_t \le \left(1 - \frac{\mu}{4A}\right)\frac{16B}{\mu^2}\frac{1}{t+1} \le \frac{16B}{\mu^2}\frac{1}{t+2}$$

where the last inequality follows from

$$\left(1 - \frac{\mu}{4A}\right)\frac{1}{t+1} \le \frac{1}{t+2} \quad \Leftrightarrow \quad t \ge \frac{2A}{\mu} - 2 \quad \Leftarrow \quad t \ge \frac{2A}{\mu}.$$

$\square$

# D  PROOFS FOR IAM : MOMENTUM WITH POLYAK STEPSIZES

## D.1  MOMENTUM VS. ITERATE MOVING AVERAGE (IAM)

Here we detail the relationship between momentum and iterate averaging, which hinges on the following lemma.

**Lemma D.1.** (Garrigos & Gower (2023, Lemma 7.3) and Defazio & Gower (2021, Theorem 1))
The iterates $(x_t)_{t\ge0}$ generated by (10) and the *iterate-moving-average* (IAM) are equivalent to if $z_{-1} = x_0$, $m_{-1} = 0$ and the $(\gamma_t, \beta_t)$ parameters of momentum and the IAM parameters $(\eta_t, \lambda_t)$ satisfy

$$\beta_t = \frac{\lambda_t}{1+\lambda_t}\frac{\eta_{t-1}}{\eta_t}, \quad \text{and} \quad \gamma_t = \frac{\eta_t}{1+\lambda_{t+1}}, \quad \forall t \ge 0. \tag{44}$$

As an example of using the above lemma, a constant learning rate $\eta_t \equiv \eta$ and $\lambda_t = t$ in the IAM method (3–4) corresponds to a decreasing learning rate $\gamma_t = \frac{\eta}{1+t}$ and an increasing momentum $\beta_t = \frac{t}{1+t}$ in the momentum method (10).

*Proof.* The proof is by induction. Our induction hypothesis is that $x_t$ iterates in (4) and (10) are equivalent upto step $t$ and that the $z_t$ iterates in (3) and $m_t$ in (10) satisfy

$$z_t = x_t - (1+\lambda_{t+1})\gamma_t m_t. \tag{45}$$

For the base case $t = 0$ we have from (3) that

$$z_0 = z_{-1} - \eta_0 g_0$$
$$= x_0 - (1+\lambda_1)\gamma_0 g_0,$$

where in the second equality we used $z_{-1} = x_0$ and (44). Since $m_{-1} = 0$, we have from (10) that $m_0 = g_0$, which proves (45) for the base case. As for the $x_t$ iterates in (4) and (10) being equivalent for $t = 0$ from (4) and (45) we have that

$$x_1 = \frac{\lambda_1}{1+\lambda_1}x_0 + \frac{1}{1+\lambda_1}z_0$$
$$= \frac{\lambda_1}{1+\lambda_1}x_0 - \frac{1}{1+\lambda_1}(x_0 - (1+\lambda_1)\gamma_0 m_0)$$
$$= x_0 - \gamma_0 m_0,$$

which is equivalent to the first step of (10).

Suppose now that $x_t$ iterates in (4) and (10) are equivalent and (45) holds upto time $t$. From (3) at step $t + 1$ we have that

$$
\begin{aligned}
z_{t+1} &= z_t - \eta_{t+1} g_{t+1} \\
&= x_t - (1 + \lambda_{t+1})\gamma_t m_t - \eta_{t+1} g_{t+1}. && \text{Using (45)} \\
&= x_t - (1 + \lambda_{t+1})\gamma_t m_t - (1 + \lambda_{t+2})\gamma_{t+1} g_{t+1} && \text{Using (44)} \\
&= x_t - \gamma_t m_t + (1 + \lambda_{t+2})\gamma_{t+1} \left( \frac{\lambda_{t+1}}{1 + \lambda_{t+2}} \frac{\gamma_t}{\gamma_{t+1}} m_t + g_{t+1} \right) \\
&= x_{t+1} - (1 + \lambda_{t+2})\gamma_{t+1} \left( \beta_{t+1} m_t + g_{t+1} \right) && \text{Using (4) and (44)} \\
&= x_{t+1} - (1 + \lambda_{t+2})\gamma_{t+1} m_{t+1} && \text{Using (10)} ,
\end{aligned}
$$

which shows that (45) holds at time $t + 1$. Finally $t + 1$ step. From (4) and (3) we have that

$$
\begin{aligned}
x_{t+1} &= \frac{\lambda_{t+1}}{1 + \lambda_{t+1}} x_t + \frac{1}{1 + \lambda_{t+1}} z_t \\
&= \frac{\lambda_{t+1}}{1 + \lambda_{t+1}} x_t + \frac{1}{1 + \lambda_{t+1}} (x_t - (1 + \lambda_{t+1})\gamma_t m_t) \\
&= x_t - \gamma_t m_t,
\end{aligned}
$$

which is equivalent to (10), and thus concludes the proof. $\qquad\square$

### D.2  PRELIMINARY BOUNDS FOR IAM IN BOTH NONSMOOTH AND SMOOTH SETTINGS

Our proofs all start from the following Lemma.

**Lemma D.2.** Consider the iterates of Algorithm 1 with $\lambda_t > 0$. Assume that $g_t \neq 0$ for all $t \geq 0$. Let $g(x)$ denote the subgradient of $f(x)$. Denote by $\mathcal{F}_t$ the filtration generated by $\xi_0, \ldots, \xi_{t-1}$. If $f_\xi$ is convex for every $\xi$, then:

(i) (Almost sure boundedness). With probability one, we have $\|z_t - x_*\| \leq \|x_0 - x_*\|$ and $\|x_t - x_*\| \leq \|x_0 - x_*\|$ for all $t \geq 0$.

(ii) (Single recurrence) It holds for any $t \geq 0$

$$
\mathbb{E}\left[ \|z_t - x_*\|^2 \mid \mathcal{F}_t \right] \leq \|z_{t-1} - x_*\|^2 - \frac{\left( f(x_t) - f(x_*) + \langle g(x_t), z_{t-1} - x_t \rangle \right)_+^2}{\mathbb{E}\left[ \|g_t\|^2 \mid \mathcal{F}_t \right]}.
\tag{46}
$$

(iii) (Summed recurrence) It holds for any $k \geq 0$

$$
\mathbb{E}\left[ \|z_k - x_*\|^2 \right] \leq \|z_0 - x_*\|^2 - \frac{\left( \sum_{t=0}^{k} \mathbb{E}\left[ f(x_t) - f(x_*) + \langle g(x_t), z_{t-1} - x_t \rangle \right] \right)_+^2}{\sum_{t=0}^{k} \mathbb{E}\left[ \|g_t\|^2 \right]}.
\tag{47}
$$

*Proof.* Substituting (12) back into the bound (11) gives

$$
\|z_t - x_*\|^2 \leq \|z_{t-1} - x_*\|^2 - \frac{\left( f_{\xi_t}(x_t) - f_{\xi_t}(x_*) + \langle g_t, z_{t-1} - x_t \rangle \right)_+^2}{\|g_t\|^2}.
$$

This shows that $\|z_t - x_*\| \leq \|z_0 - x_*\| = \|x_0 - x_*\|$ almost surely for all $t \geq 0$. Since $x_{t+1}$ is a convex combination of $x_t$ and $z_t$ (see line 4 in Algorithm 1) this also shows by a straightforward induction that $\|x_t - x_*\| \leq \|x_0 - x_*\|$ almost surely for all $t \geq 0$.

To prove (ii), we apply conditional expectation on the above inequality and using Lemma C.3, (17) we obtain

$$
\mathbb{E}\left[ \|z_t - x_*\|^2 \mid \mathcal{F}_t \right] \leq \|z_{t-1} - x_*\|^2 - \frac{\left( f(x_t) - f(x_*) + \langle \mathbb{E}\left[ g_t \mid \mathcal{F}_t \right], z_{t-1} - x_t \rangle \right)_+^2}{\mathbb{E}\left[ \|g_t\|^2 \mid \mathcal{F}_t \right]}.
$$

Using that the expectation with respect to this filtration is independent of $x_t$, we have that the stochastic subgradient $\mathbb{E}\left[g_t \mid \mathcal{F}_t\right]$ is a subgradient of $f(x_t)$, see Lemma 9.5 in Garrigos & Gower (2023) for details[3]. Thus we can write $g(x_t) = \mathbb{E}\left[g_t \mid \mathcal{F}_t\right]$.

Now, define $a_t := f(x_t) - f(x_*) + \langle g(x_t), z_{t-1} - x_t \rangle$ and $b_t = \mathbb{E}\left[\|g_t\|^2 \mid \mathcal{F}_t\right]$. Using (46) subsequently for $t = 0, \dots, k$ and using the tower property, we obtain

$$\mathbb{E}\left[\|z_k - x_*\|^2\right] \leq \|z_0 - x_*\|^2 - \mathbb{E}\left[\sum_{t=0}^{k} \frac{(a_t)_+^2}{b_t}\right].$$

Now using Lemma C.3, (18) yields

$$\sum_{t=0}^{k} \frac{(a_t)_+^2}{b_t} \geq \frac{\left(\sum_{t=0}^{k} a_t\right)_+^2}{\sum_{t=0}^{k} b_t},$$

which implies, using (17), that

$$\mathbb{E}\left[\sum_{t=0}^{k} \frac{(a_t)_+^2}{b_t}\right] \geq \mathbb{E}\left[\frac{\left(\sum_{t=0}^{k} a_t\right)_+^2}{\sum_{t=0}^{k} b_t}\right] \geq \frac{\left(\sum_{t=0}^{k} \mathbb{E}\left[a_t\right]\right)_+^2}{\sum_{t=0}^{k} \mathbb{E}\left[b_t\right]}.$$

Altogether, we obtain (iii), that is

$$\mathbb{E}\left[\|z_k - x_*\|^2\right] \leq \|z_0 - x_*\|^2 - \frac{\left(\sum_{t=0}^{k} \mathbb{E}\left[f(x_t) - f(x_*) + \langle g(x_t), z_{t-1} - x_t \rangle\right]\right)_+^2}{\sum_{t=0}^{k} \mathbb{E}\left[\|g_t\|^2\right]}.$$

$\square$

For our forthcoming proofs we will also make use of a *Bregman viewpoint* of the `IAM` step size.

**Lemma D.3** (Bregman View). For any $x_t, x_{t-1}, x_* \in \mathbb{R}^d$ and $\lambda_t \geq 0$ it holds

$$\begin{aligned} & f(x_t) - f(x_*) + \langle g(x_t), z_{t-1} - x_t \rangle \\ & = (1 + \lambda_t)(f_{\xi_t}(x_t) - f_{\xi_t}(x_*)) - \lambda_t(f_{\xi_t}(x_{t-1}) - f_{\xi_t}(x_*)) + \lambda_t B_{f_{\xi_t}}(x_{t-1}, x_t), \end{aligned} \quad (48)$$

where $B_{f_\xi}(x, y)$ is the Bregman divergence

$$B_{f_\xi}(x, y) := f_\xi(x) - f_\xi(y) - \langle g_\xi(y), x - y \rangle.$$

*Proof.* By re-arranging (4) at time $t - 1$ we have that

$$z_{t-1} - x_t = -\lambda_t(x_{t-1} - x_t). \quad (49)$$

Consequently

$$f(x_t) - f(x_*) + \langle g(x_t), z_{t-1} - x_t \rangle = f(x_t) - f(x_*) - \lambda_t \langle g(x_t), x_{t-1} - x_t \rangle.$$

The proof follows by adding and subtracting $\lambda_t f_{\xi_t}(x_{t-1})$ as follows

$$\begin{aligned} & f_{\xi_t}(x_t) - f_{\xi_t}(x_*) - \lambda_t \langle g_t, x_{t-1} - x_t \rangle \\ & = (1 + \lambda_t)f_{\xi_t}(x_t) - f_{\xi_t}(x_*) - \lambda_t f_{\xi_t}(x_{t-1}) + \lambda_t \left(f_{\xi_t}(x_{t-1}) - f_{\xi_t}(x_t) - \langle g_t, x_{t-1} - x_t \rangle\right) \\ & = (1 + \lambda_t)(f_{\xi_t}(x_t) - f_{\xi_t}(x_*)) - \lambda_t(f_{\xi_t}(x_{t-1}) - f_{\xi_t}(x_*)) + \lambda_t B_{f_{\xi_t}}(x_{t-1}, x_t). \end{aligned}$$

$\square$

---

[3]Very formally, here we need to assume the subgradients $g_\xi(x)$ are measurable in $\xi$ so that this expectation is well defined.

**Lemma D.4.** Consider the iterates of Algorithm 1 with $\lambda_t = t$ and assume that $f_\xi$ is convex for every $\xi$, with subgradients $g_\xi$. Let $g(x)$ be subgradients of $f(x)$. It holds

$$\sum_{t=0}^{k} f(x_t) - f(x_*) + \langle g(x_t), z_{t-1} - x_t \rangle = (k+1)[f(x_k) - f(x_*)] + \sum_{t=1}^{k} \lambda_t B_f(x_{t-1}, x_t),$$

where $B_f$ is defined as in Lemma D.3. In particular, it holds $B_f(x_{t-1}, x_t) \geq 0$.

*Proof.* Note that for this proof, we need an additional, and artificial iterate $x_{-1} = x_0$. Summing over $t = 0, \ldots, k$ in (48) we have that

$$\sum_{t=0}^{k} (f(x_t) - f(x_*) + \langle g(x_t), z_{t-1} - x_t \rangle)$$

$$\stackrel{(48)}{=} \sum_{t=0}^{k} (1 + \lambda_t)(f(x_t) - f(x_*)) - \lambda_t(f(x_{t-1}) - f(x_*)) + \sum_{t=0}^{k} \lambda_t B_f(x_{t-1}, x_t)$$

$$= \sum_{t=0}^{k} \lambda_{t+1}(f(x_t) - f(x_*)) - \lambda_t(f(x_{t-1}) - f(x_*)) + \sum_{t=0}^{k} \lambda_t B_f(x_{t-1}, x_t)$$

$$= (k+1)[f(x_k) - f(x_*)] + \sum_{t=1}^{k} \lambda_t B_f(x_{t-1}, x_t),$$

where the second step used $1 + \lambda_t = 1 + t = \lambda_{t+1}$, and the last step we used telescoping and the fact that $\lambda_0 = 0$. $\square$

### D.3 PROOF OF THEOREM 3.2 ABOUT RATES FOR IAM IN THE NONSMOOTH SETTING

**Theorem 3.2** (Non-smooth setting). Consider problem (1), and let $D := \|x_0 - x_*\|$. Assume that the losses $f_\xi$ are locally Lipschitz in expectation. In particular there exists $G \geq 0$ such that (8) holds for all $x \in \mathbb{B}_D(x_*)$. Let $B_f(x, y) := f(x) - f(y) - \langle \nabla f(y), x - y \rangle$. Then the iterates of Algorithm 1 (IAM) started from $x_0$ with $\lambda_t = t$ verify the *last iterate* bound

$$\mathbb{E}\left[f(x_T) - f(x_*)\right] + \frac{1}{T+1} \sum_{t=1}^{T} t\, \mathbb{E}\left[B_f(x_{t-1}, x_t)\right] \leq \frac{GD}{\sqrt{T+1}}.$$

*Proof.* We start by applying Lemma D.2, which states that $x_t \in D$ and $z_t \in D$ almost surely for all $t \geq 0$. Further, Lemma D.2, (ii) implies that

$$\mathbb{E}\left[\|z_t - x_*\|^2 \mid \mathcal{F}_t\right] \leq \|z_{t-1} - x_*\|^2 - \frac{\left(f(x_t) - f(x_*) + \langle g(x_t), z_{t-1} - x_t \rangle\right)_+^2}{\mathbb{E}\left[\|g_t\|^2 \mid \mathcal{F}_t\right]}. \tag{50}$$

For the denominator of (50), we can therefore estimate

$$\mathbb{E}\left[\|g_t\|^2 \mid \mathcal{F}_t\right] \leq G^2.$$

Applying expectation, and summing from $t = 0, \ldots, k$ (recall that $z_{-1} = x_0$), we get

$$\sum_{t=0}^{k} \mathbb{E}\left[\left(f(x_t) - f(x_*) + \langle g(x_t), z_{t-1} - x_t \rangle\right)_+^2\right] \leq G^2\left[\|x_0 - x_*\|^2 - \mathbb{E}\left[\|z_k - x_*\|^2\right]\right]. \tag{51}$$

Now, applying (18) with $b_t = 1$ we get for any $a_0, \ldots, a_k$ that $\sum_{t=0}^{k} (a_t)^2_+ \geq \frac{1}{k+1} \left( \sum_{t=0}^{k} a_t \right)^2_+$. Therefore, we conclude

$$\sum_{t=0}^{k} \left( f(x_t) - f(x_*) + \langle g(x_t), z_{t-1} - x_t \rangle \right)^2_+ \geq \frac{1}{k+1} \left( \sum_{t=0}^{k} f(x_t) - f(x_*) + \langle g(x_t), z_{t-1} - x_t \rangle \right)^2_+$$

$$\geq \frac{1}{k+1} \left( (k+1)[f(x_k) - f(x_*)] + \sum_{t=1}^{k} \lambda_t B_f(x_{t-1}, x_t) \right)^2_+$$

$$= \left( \sqrt{k+1}[f(x_k) - f(x_*)] + \sum_{t=1}^{k} \frac{\lambda_t}{\sqrt{k+1}} B_f(x_{t-1}, x_t) \right)^2,$$

where we used Lemma D.4 in the second step, and non-negativity of all terms in the third step. Define $\bar{B}_k := \sum_{t=1}^{k} \lambda_t B_f(x_{t-1}, x_t) \geq 0$. Plugging this into (51), we get

$$\mathbb{E} \left[ \left( \sqrt{k+1}[f(x_k) - f(x_*)] + \frac{1}{\sqrt{k+1}} \bar{B}_k \right)^2 \right] \leq G^2 \left[ \|x_0 - x_*\|^2 - \mathbb{E} \left[ \|z_k - x_*\|^2 \right] \right].$$

Now, using Jensen's inequality $\mathbb{E}[X]^2 \leq \mathbb{E}[X^2]$, taking the square-root, and dividing by $\sqrt{k+1}$, we finally obtain

$$\mathbb{E}[f(x_k) - f(x_*)] + \frac{1}{k+1} \mathbb{E}[\bar{B}_k] \leq \frac{G\|x_0 - x_*\|}{\sqrt{k+1}}.$$

$\square$

### D.4 PROOF OF THEOREM 3.3 ABOUT RATES FOR IAM IN THE SMOOTH SETTING

**Theorem 3.3** (Smooth setting). Consider problem (1), and let $D := \|x_0 - x_*\|$. Assume that the losses $f_\xi$ are uniformly locally smooth. In particular there exists $L \geq 0$ such that (9) holds for all $x \in \mathbb{B}_D(x_*)$. Then the iterates of Algorithm 1 (IAM) started from $x_0$ with $\lambda_t = t$ verify this *last iterate* bound, with $\Delta_* := \inf f - \mathbb{E}_\xi[\inf f_\xi]$:

$$\mathbb{E}[f(x_{T-1}) - f(x_*)] \leq \frac{2LD^2(\log(T)+1)}{T} + \frac{\sqrt{2L\Delta_*}D}{\sqrt{T}}.$$

*Proof.* We start the proof by applying Lemma D.2, (iii), which yields

$$\mathbb{E}\left[\|z_k - x_*\|^2\right] \leq \|z_0 - x_*\|^2 - \frac{\left( \sum_{t=0}^{k} \mathbb{E}[f(x_t) - f(x_*) + \langle \nabla f(x_t), z_{t-1} - x_t \rangle] \right)^2_+}{\sum_{t=0}^{k} \mathbb{E}[\|g_t\|^2]}.$$

For the numerator of the last term, use Lemma D.4 and the fact that $(\cdot)^2_+$ is monotonic to obtain

$$\left( \sum_{t=0}^{k} \mathbb{E}[f(x_t) - f(x_*) + \langle \nabla f(x_t), z_{t-1} - x_t \rangle] \right)^2_+ \geq \left( \mathbb{E}[(k+1)(f(x_k) - f(x_*))] \right)^2_+$$

$$= (k+1)^2 \mathbb{E}[f(x_k) - f(x_*)]^2.$$

For the denominator, use (9) to write that $\mathbb{E}[\|g_t\|^2] \leq 2L\mathbb{E}[(f(x_t) - f(x_*) + \Delta_*)]$. Thus, using $z_0 = x_0$, we get

$$\mathbb{E}\left[\|z_k - x_*\|^2\right] \leq \|x_0 - x_*\|^2 - \frac{(k+1)^2 \mathbb{E}[(f(x_k) - f(x_*))]^2}{2L \sum_{t=0}^{k} \mathbb{E}[(f(x_t) - f(x_*) + \Delta_*)]}.$$

Let $c_t = \mathbb{E}[f(x_t) - f(x_*)] + \Delta_*$ and $S_k = \sum_{t=0}^{k} c_t$, then we can rewrite the above as

$$\frac{\mathbb{E}[(f(x_k) - f(x_*))]^2}{S_k} \leq \frac{2L}{(k+1)^2} (\|x_0 - x_*\|^2 - \mathbb{E}[\|z_k - x_*\|^2]) \leq \frac{2L\|x_0 - x_*\|^2}{(k+1)^2}.$$

Taking the square-root yields

$$\frac{\mathbb{E}\left[f(x_k) - f(x_*)\right]}{\sqrt{S_k}} \leq \frac{\sqrt{2L}\|x_0 - x_*\|}{k+1}. \tag{52}$$

Finally, notice that $\mathbb{E}\left[f(x_k) - f(x_*)\right] = c_k - \Delta_*$, so we arrive at the inequality

$$\frac{c_t}{\sqrt{S_t}} \leq \frac{\sqrt{2L}\|x_0 - x_*\|}{t+1} + \frac{\Delta_*}{\sqrt{S_t}}.$$

Summing this from $t = 0$ to $k$ and then applying $\sum_{t=0}^k \frac{1}{t+1} \leq \log(k+1) + 1$ and $S_t \geq (t+1)\Delta_*$ gives

$$\sqrt{S_k} \overset{(16)}{\leq} \sum_{t=0}^k \frac{c_t}{\sqrt{S_t}} \leq \sum_{t=0}^k \frac{\sqrt{2L}\|x_0 - x_*\|}{t+1} + \sum_{t=0}^k \frac{\Delta_*}{\sqrt{S_t}}$$

$$\leq \sqrt{2L}\|x_0 - x_*\|(\log(k+1) + 1) + \sum_{t=0}^k \frac{\sqrt{\Delta_*}}{\sqrt{t+1}}.$$

Furthermore, it holds $\sum_{t=0}^k \frac{1}{\sqrt{t+1}} \leq 2\sqrt{k+1}$, so we finally get

$$\sqrt{S_k} \leq \sqrt{2L}\|x_0 - x_*\|(\log(k+1) + 1) + \sqrt{\Delta_*}\sqrt{k+1}.$$

Using the above inequalities in (52) gives

$$\mathbb{E}\left[f(x_k) - f(x_*)\right] \leq \frac{\sqrt{2L}\|x_0 - x_*\|\sqrt{S_k}}{k+1} \leq \frac{2L\|x_0 - x_*\|^2(\log(k+1) + 1)}{k+1} + \frac{\sqrt{2L\Delta_*}\|x_0 - x_*\|}{\sqrt{k+1}}.$$

$\square$

### D.5 ADDITIONAL RESULT: CONVERGENCE FOR IAM IN THE NONSMOOTH SETTING WITH DECREASING $\lambda_t$

**Theorem D.5.** Consider the setting of Theorem 3.2, except that $\lambda_0 = 0$ and $(\lambda_t)_{t=1}^k$ is any decreasing sequence of nonnegative reals. It follows that

$$\mathbb{E}[f(\overline{x}_k) - f(x_*)] + \frac{1}{k+1}\sum_{t=0}^k \lambda_t \mathbb{E}[B_f(x_{t-1}, x_t)] \leq \frac{G\|x_0 - x_*\|}{\sqrt{k+1}} + \frac{\lambda_1}{k+1}\mathbb{E}[f(x_0) - f(x_*)].$$

The advantage of this result is that it holds for any constant $\lambda_t = \lambda$. Translating this to the momentum method (10), this allows for other parameter setting of $(\gamma_t, \beta_t)$. In particular the setting $\lambda_t = \lambda = 0$ which corresponds to no momentum. In this setting we retrieve the exact same rate as the SPS* method in Corollary 2.2. However, as mentioned earlier the price we need to pay for this, is that this result holds for the Cesaro average and not of the last iterate.

*Proof.* Starting from Lemma D.2:

$$\|z_t - x_*\|^2 \leq \|z_{t-1} - x_*\|^2 - \frac{\left(f(x_t) - f(x_*) + \langle g_t, z_{t-1} - x_t \rangle\right)_+^2}{\|g_t\|^2}. \tag{53}$$

Taking expectation, and using our extended Titu's Lemma C.3 and Bregman viewpoint Lemma D.3 to get

$$\mathbb{E}\|z_t - x_*\|^2 \leq \mathbb{E}\|z_{t-1} - x_*\|^2 - \frac{\mathbb{E}[f(x_t) - f(x_*) + \langle g(x_t), z_{t-1} - x_t \rangle]_+^2}{\mathbb{E}\|g_t\|^2}$$

$$\leq \mathbb{E}\|z_{t-1} - x_*\|^2 - \frac{\mathbb{E}[(1 + \lambda_t)[f(x_t) - f(x_*)] - \lambda_t[f(x_{t-1}) - f(x_*)] + \lambda_t B_f(x_{t-1}, x_t)]_+^2}{\mathbb{E}\|g_t\|^2}$$

$$\leq \mathbb{E}\|z_{t-1} - x_*\|^2 - \frac{\mathbb{E}[(1 + \lambda_t)[f(x_t) - f(x_*)] - \lambda_t[f(x_{t-1}) - f(x_*)] + \lambda_t B_f(x_{t-1}, x_t)]_+^2}{G^2}.$$

Multiplying through by $G^2$ gives

$$\mathbb{E}[(1 + \lambda_t)[f(x_t) - f(x_*)] - \lambda_t[f(x_{t-1}) - f(x_*)] + \lambda_t B_f(x_{t-1}, x_t)]_+^2 \leq G^2 \mathbb{E}\|z_{t-1} - x_*\|^2 - G^2 \mathbb{E}\|z_t - x_*\|^2.$$

Now let $\Delta_t = (1 + \lambda_t)[f(x_t) - f(x_*)] - \lambda_t[f(x_{t-1}) - f(x_*)] + \lambda_t B_f(x_{t-1}, x_t)$. Averaging both sides of the above over $t = 0, \ldots, k$, telescoping terms, and using Jensen's inequality with respect to the convex function $x \mapsto (x_+)^2$ gives

$$\frac{G^2 \|x_0 - x_*\|^2}{k+1} \geq \frac{G^2}{k+1} \left( \mathbb{E}\|x_0 - x_*\|^2 - \mathbb{E}\|z_{k+1} - x_*\|^2 \right)$$

$$\geq \frac{1}{k+1} \sum_{t=0}^{k} \mathbb{E}[\Delta_t]_+^2$$

$$\geq \left( \frac{1}{k+1} \sum_{t=0}^{k} \mathbb{E}[\Delta_t] \right)_+^2.$$

Taking the square root gives

$$\left( \frac{1}{k+1} \sum_{t=0}^{k} \mathbb{E}[\Delta_t] \right)_+ \leq \frac{G\|x_0 - x_*\|}{\sqrt{k+1}}. \tag{54}$$

Now since $(\lambda_t)$ is decreasing and using Jensen's inequality with respect to $x \mapsto f(x)$ we have that

$$\sum_{t=0}^{k} \mathbb{E}[\Delta_t] = \sum_{t=0}^{k} (1 + \lambda_t) \mathbb{E}[f(x_t) - f(x_*)] - \lambda_t \mathbb{E}[f(x_{t-1}) - f(x_*)] + \lambda_t \mathbb{E}[B_f(x_{t-1}, x_t)]$$

$$= \sum_{t=0}^{k} \lambda_t \mathbb{E}[B_f(x_{t-1}, x_t)] + \sum_{t=0}^{k} \mathbb{E}[f(x_t) - f(x_*)] + \sum_{t=0}^{k} \lambda_t \mathbb{E}[f(x_t) - f(x_*)] - \sum_{t=0}^{k} \lambda_t \mathbb{E}[f(x_{t-1}) - f(x_*)]$$

$$= \sum_{t=0}^{k} \lambda_t \mathbb{E}[B_f(x_{t-1}, x_t)] + \sum_{t=0}^{k} \mathbb{E}[f(x_t) - f(x_*)] + \sum_{t=1}^{k} \lambda_t \mathbb{E}[f(x_t) - f(x_*)] - \sum_{t=1}^{k} \lambda_t \mathbb{E}[f(x_{t-1}) - f(x_*)]$$

$$= \sum_{t=0}^{k} \lambda_t \mathbb{E}[B_f(x_{t-1}, x_t)] + \sum_{t=0}^{k} \mathbb{E}[f(x_t) - f(x_*)] + \sum_{t=1}^{k-1} (\lambda_t - \lambda_{t+1}) \mathbb{E}[f(x_t) - f(x_*)]$$

$$+ \lambda_k \mathbb{E}[f(x_k) - f(x_*)] - \lambda_1 \mathbb{E}[f(x_0) - f(x_*)]$$

$$\geq \sum_{t=0}^{k} \lambda_t \mathbb{E}[B_f(x_{t-1}, x_t)] + \sum_{t=0}^{k} \mathbb{E}[f(x_t) - f(x_*)] - \lambda_1 \mathbb{E}[f(x_0) - f(x_*)]$$

$$\geq \sum_{t=0}^{k} \lambda_t \mathbb{E}[B_f(x_{t-1}, x_t)] + (k+1) \mathbb{E}[f(\overline{x}_k) - f(x_*)] - \lambda_1 \mathbb{E}[f(x_0) - f(x_*)],$$

where we used that $\lambda_0 = 0$ and $\lambda_t - \lambda_{t+1} \geq 0$ since $(\lambda_t)$ is a decreasing sequence. Dividing through by $(k+1)$ gives

$$\frac{1}{k+1} \sum_{t=0}^{k} \mathbb{E}[\Delta_t] \geq \frac{1}{k+1} \sum_{t=0}^{k} \lambda_t \mathbb{E}[B_f(x_{t-1}, x_t)] + \mathbb{E}[f(\overline{x}_k) - f(x_*)] - \frac{\lambda_1}{k+1} \mathbb{E}[f(x_0) - f(x_*)].$$

Using the above and (54) gives

$$\left( \frac{1}{k+1} \sum_{t=0}^{k} \lambda_t \mathbb{E}[B_f(x_{t-1}, x_t)] + \mathbb{E}[f(\overline{x}_k) - f(x_*)] - \frac{\lambda_1}{k+1} \mathbb{E}[f(x_0) - f(x_*)] \right) \leq \left( \frac{1}{k+1} \sum_{t=0}^{k} \mathbb{E}[\Delta_t] \right)_+$$

$$\leq \frac{G\|x_0 - x_*\|}{\sqrt{k+1}}.$$

Re-arranging gives the result. $\qquad \square$

### D.6 Additional Result: Deriving an Expression for Adam with Polyak Stepsizes

Following an analogous reasoning used in Section 3, we can derive variants of `IAM` that use precon-
ditioning. This is particularily important for models such as Transformers, where using an `Adam`
preconditioner is required to achieve a reasonable performance.

To arrive at a preconditioned version of `IAM`, let $\boldsymbol{D}_t \in \mathbb{R}^{d \times d}$ be our positive definite symmetric
preconditioner, and let $\|z\|_{\boldsymbol{D}_t}^2 := \langle \boldsymbol{D}_t z, z \rangle$ be the norm induced by this preconditioner. Now consider
the iterative averaging method with this preconditioner:

$$z_t = z_{t-1} - \eta_t \boldsymbol{D}_t^{-1} g_t, \tag{55}$$

$$x_{t+1} = \frac{\lambda_{t+1}}{1 + \lambda_{t+1}} x_t + \frac{1}{1 + \lambda_{t+1}} z_t. \tag{56}$$

Now we upper bound the distance between $z_t$ and a solution $x_*$ under the preconditioned norm via

$$\|z_t - x_*\|_{\boldsymbol{D}_t}^2 = \|z_{t-1} - x_*\|_{\boldsymbol{D}_t}^2 - 2\eta_t \left\langle \boldsymbol{D}_t^{-1} g_t, z_{t-1} - x_* \right\rangle_{\boldsymbol{D}_t} + \eta_t^2 \|g_t\|_{\boldsymbol{D}_t^{-1}}^2$$

$$= \|z_{t-1} - x_*\|_{\boldsymbol{D}_t}^2 - 2\eta_t \langle g_t, z_{t-1} - x_* \rangle + \eta_t^2 \|g_t\|_{\boldsymbol{D}_t^{-1}}^2$$

$$\leq \|z_{t-1} - x_*\|_{\boldsymbol{D}_t}^2 - 2\eta_t \left( f_{\xi_t}(x_t) - f_{\xi_t}(x_*) + \langle g_t, z_{t-1} - x_t \rangle \right) + \eta_t^2 \|g_t\|_{\boldsymbol{D}_t^{-1}}^2,$$

where in the inequality we used that $f_{\xi_t}$ is convex. Minimizing the right-hand side with respect to
$\eta_t$ now gives the step size given in line 3 in Algorithm 2. To arrive at our `IAM-Adam` method, we

---

**Algorithm 2:** `IAM-Adam`

1 **Input:** $z_{-1} = x_0 \in \mathbb{R}^d$, $\lambda_t > 0$
2 **for** $t = 0$ *to* $T - 1$ **do**

3 $\quad \eta_t = \dfrac{\left[ f_{\xi_t}(x_t) - \ell_{\xi_t}^* + \langle g_t, z_{t-1} - x_t \rangle \right]_+}{\|g_t\|_{\boldsymbol{D}_t^{-1}}^2},$

4 $\quad z_t = z_{t-1} - \eta_t \boldsymbol{D}_t^{-1} g_t$

5 $\quad x_{t+1} = \dfrac{\lambda_{t+1}}{1 + \lambda_{t+1}} x_t + \dfrac{1}{1 + \lambda_{t+1}} z_t$

6 **Return:** $x_T$

---

simply set $\boldsymbol{D}_t$ to be the preconditioner used by `Adam`, that is $\boldsymbol{D}_t = \text{diag}(\sqrt{v_t} + \epsilon)$ where

$$v_{t+1} = \beta_2 v_t + (1 - \beta_2) g_t \odot g_t.$$

## E DISCUSSION

### E.1 Anytime Convergence Rates vs. Finite Horizon Complexity Rates

Here we take the time to properly define what we mean by (anytime) convergence rates and (finite
horizon) complexity rates. For the sake of the discussion, we consider a certain quantity of interest
$(q_t)_{t \in \mathbb{N}} \subset (0, +\infty)$ which we want to be small when $t$ grows. Typically $q_t$ will measure an optimality
gap for some algorithm, such as the function value gap $\mathbb{E}[f(x_t) - \inf f]$. What is the algorithm here
does not really matter. What does matter is that this algorithm depends on hyperparameters, which
once fixed generate a sequence which in turn define the quantities of interest $(q_t)_{t \geq 0}$.

1. We say that we have an anytime *convergence rate* for $q_t$ if, for every choice of hyperparame-
   ter, there exists a rate function $r : [0, +\infty) \to [0, +\infty)$ such that $q_t \leq r(t)$ holds true for
   every $t \in \mathbb{N}$, and $r(t) \to 0$ when $t \to +\infty$.

2. We say that we have a finite horizon *complexity rate* for $q_t$ if, for every tolerance $\varepsilon > 0$,
   there exists a choice of hyperparameters and $T \in \mathbb{N}$ such that $q_T \leq \varepsilon$.

A typical example of convergence rate is the function value gap for the Gradient Descent algorithm
with constant stepsize: for every choice of stepsize $\gamma \in (0, \frac{1}{L})$, for every starting point $x_0$, we know

that

$$f(x_t) - \inf f \leq \frac{\|x_0 - x_*\|^2}{2\gamma t}.$$

An other example is SGD with a vanishing stepsize. It should be clear that convergence rates entail complexity rate: given any tolerance $\varepsilon > 0$, because $r(t) \to 0$ we can find a $T$ large enough so that $r(T) \leq \varepsilon$. The reverse is in general not true: $q_t$ can have a complexity rate without having a convergence rate. A typical example is SGD with a constant stepsize: for every choice of stepsize $\gamma \in (0, \frac{1}{4L})$, for every starting point $x_0$, we know that

$$\mathbb{E}\left[f(\bar{x}_t) - \inf f\right] \leq \frac{\|x_0 - x_*\|^2}{2\gamma t} + 2\gamma \sigma_*^2.$$

This means that, for every fixed tolerance $\varepsilon > 0$, we can chose a stepsize $\gamma = O(\varepsilon)$ small enough, and take $T = O(\frac{1}{\varepsilon^2})$ large enough so that

$$\mathbb{E}\left[f(\bar{x}_t) - \inf f\right] \leq \varepsilon.$$

The key difference bewteen convergence rates and complexity rates is that for convergence rates the user does not need to fix the hyperparameters as a function of the desired tolerance.

### E.2 COMPLEXITY OF SGD WITH RESPECT TO INTERPOLATION

**Theorem E.1** (Complexity of SGD). Let $f = \frac{1}{n}\sum_{i=1}^n f_i$ where each $f_i : \mathbb{R}^d \to \mathbb{R}$ is convex and $L$-smooth, and assume that $f$ admits a minimizer, noted $x_*$. Let $x_0 \in \mathbb{R}^d$, and note $D := \|x_0 - x_*\|$ and $\sigma_*^2 = \mathbb{E}\left[\|\nabla f_i(x_*)\|^2\right]$. Let $T \geq 1$ be fixed, and let $\gamma = \frac{\gamma_0}{\sqrt{vT+1}}$ where $\gamma_0 \leq \frac{1}{4L}$, and $v \geq 0$ is a variance upper estimate of $\sigma_*^2$ satisfying $\sigma_*^2 \leq Cv$ for some $C > 0$. Let $(x_t)_{t=0}^T$ be the sequence generated by the SGD algorithm with constant stepsize $\gamma$. Then

$$\mathbb{E}\left[f(\bar{x}_T) - \inf f\right] \leq \frac{D^2}{\gamma_0 T} + \frac{\sqrt{v}}{\sqrt{T}}\left(\frac{D^2}{\gamma_0} + 2\gamma_0 C\right),$$

where $\bar{x}_T = \frac{1}{T}\sum_{t=0}^{T-1} x_t$. In particular the choice $v = \sigma_*^2$ leads to $\gamma = \frac{\gamma_0}{\sqrt{\sigma_*^2 T + 1}}$ and gives the rate

$$\mathbb{E}\left[f(\bar{x}_T) - \inf f\right] \leq \frac{D^2}{\gamma_0 T} + \frac{\sigma_*}{\sqrt{T}}\left(\frac{D^2}{\gamma_0} + 2\gamma_0\right).$$

The above theorem shows that SGD enjoys a $\mathcal{O}\left(LD^2/T + \sigma_* LD^2/\sqrt{T}\right)$ complexity, which is similar to the result of SPS* in Corollary 2.3. But there are some important differences. First, SPS* has an anytime convergence rate valid for every $T$, while SGD has a complexity rate: it is a "finite horizon" rate where the horizon must be known before setting the stepsize. The other difference is that for SGD to achieve this complexity, we need access to the smoothness constant $L$. In contrast, SPS* adapts to both smoothness and non-smoothness. Of course the main counterpart for SPS* to achieve this is that it requires access to $f_\xi(x_*)$. Note that we discuss in Section E.5 on how to implement SPS* in practice.

The last important point is that SGD needs access to the gradient variance constant $\sigma_*^2$, or at least a faithful upper estimate $v$ of $\sigma_*^2$. By *faithful*, we mean here a constant $v$ which is zero whenever interpolation holds, so that the complexity reduces to $\mathcal{O}\left(\frac{1}{T}\right)$ when interpolation holds. An example of such faithful upper estimate is the function noise constant $\Delta_* := \inf f - \mathbb{E}_\xi\left[\inf f_\xi\right]$, since it satisfies $\sigma_*^2 \leq 2L\Delta_*$ and is zero if and only if interpolation holds (see Lemma 4.15 & 4.18 in Garrigos & Gower (2023)). Observe that if we assume being able to access $f_\xi(x_*)$, then we can reasonably compute $\Delta_*$. In that case SGD can benefit from it by setting $\gamma = \frac{\gamma_0}{\sqrt{\Delta_* T + 1}}$, leading to a complexity of the form

$$\mathbb{E}\left[f(\bar{x}_T) - \inf f\right] \leq \frac{D^2}{\gamma_0 T} + \frac{\sqrt{\Delta_*}}{\sqrt{T}}\left(\frac{D^2}{\gamma_0} + 4\gamma_0 L\right),$$

which is (up to constants) as good as the one of SPS*, except for the fact that this is a *finite horizon complexity*.

*Proof of Theorem E.1.* We are using the complexity rate of SGD for a sum of convex smooth functions from Theorem 5 in Garrigos & Gower (2023). The result states that, provided $\gamma \leq \frac{1}{4L}$, we can guarantee

$$\mathbb{E}\left[f(\bar{x}_T) - \inf f\right] \leq \frac{D^2}{\gamma T} + 2\gamma \sigma_*^2.$$

With our choice of stepsize $\gamma = \frac{\gamma_0}{\sqrt{vT+1}}$, one sees that the hypothesis $\gamma \leq \frac{1}{4L}$ is guaranteed as long as $\gamma_0 \leq \frac{1}{4L}$. The desired bound then follows, after cleaning some constants. More precisely, write

$$\frac{1}{\gamma} = \frac{\sqrt{1+vT}}{\gamma_0} \leq \frac{1+v\sqrt{T}}{\gamma_0} \quad \text{and} \quad 2\gamma\sigma_*^2 = \frac{2\gamma_0\sigma_*^2}{\sqrt{1+vT}} \leq \frac{2\gamma_0 C v}{\sqrt{1+vT}} \leq \frac{2\gamma_0 C\sqrt{v}}{\sqrt{T}},$$

and combine all the above inequalities to conclude that

$$\mathbb{E}\left[f(\bar{x}_T) - \inf f\right] \leq \frac{D^2}{\gamma T} + 2\gamma\sigma_*^2 \leq \frac{D^2(1+v\sqrt{T})}{\gamma_0 T} + 2\frac{\gamma_0\sigma_*^2}{\sqrt{T}} = \frac{D^2}{\gamma_0 T} + \frac{D^2\sigma_*^2}{\gamma_0\sqrt{T}} + 2\frac{\gamma_0\sigma_*^2}{\sqrt{T}}.$$

$\square$

### E.3 COMPARISON OF RATES WITH A STOCHASTIC POLYAK STEPSIZE, IN THE SMOOTH CASE

Let us start with SGD. Previously, Theorem 2.3.2 in Garrigos et al. (2023) showed a $\mathcal{O}(1/T)$ anytime convergence rate for SPS* under an interpolation assumption, but with no guarantees without interpolation. Theorem 3.4 in Loizou et al. (2021) provided the following general anytime bound for $\text{SPS}_{\max}$, independently on interpolation holding true or not:

$$\mathbb{E}\left[f(\bar{x}_T) - f(x_*)\right] \leq \max\left(1, 2\gamma_b L\right)\left(\frac{\|x_0 - x_*\|^2}{\gamma_b T} + 2\Delta_*\right),$$

where $\gamma_b$ is the parameter appearing in the definition of $\text{SPS}_{\max}$, recall eq. (4) ; and $\Delta_* = \inf f - \mathbb{E}\inf f_\xi$ is a constant which is zero if and only if interpolation holds (see Section 5.3 in Garrigos & Gower (2023) for a proof). Unfortunately, this bound is not a convergence rate. Even worse, because the variance term $\Delta_*$ cannot be controlled say, by a multiplicative free parameter), the above bound cannot be converted into a complexity result, even if we had access to the value of $L$. More recently, a variant of stochastic Polyak stepsizes called NGN was proposed Orvieto & Xiao (2024), and provably enjoys the following complexity bound (see Theorem 4.5 in Orvieto & Xiao (2024)):

$$\mathbb{E}\left[f(\bar{x}_T) - f(x_*)\right] \leq \mathcal{O}\left(\frac{\|x_0 - x_*\|^2}{\gamma' T} + \gamma'\Delta_* + \gamma'(2\gamma' L - 1)_+\mathbb{E}\left[\inf f_\xi\right]\right),$$

where $\gamma' > 0$ is a free parameter of the NGN method. One can see that this rate can be converted into a $\mathcal{O}(1/\sqrt{T})$ complexity, provided we take a small enough parameter $\gamma'$.

In contrast, our smooth result in Corollary 2.3 is, as far as we know, the first $\mathcal{O}\left(1/\sqrt{T}\right)$ *anytime convergence rate* which is adaptive to interpolation for a stochastic variant of the Polyak stepsize, let alone for vanilla SGD. For SGD, it is not possible to have at the same time an anytime $\mathcal{O}\left(1/\sqrt{T}\right)$ convergence rate together with adaptivity to interpolation, at least up to our knowledge. For instance, taking vanishing stepsizes $\gamma_t \propto \frac{1}{\sqrt{t}}$ guarantees (see Theorem 5.7 in Garrigos & Gower (2023)) an anytime convergence rate

$$\mathbb{E}\left[f(\bar{x}_T) - f(x_*)\right] \leq \mathcal{O}\left(\frac{1 + \sigma_*^2\log(T+1)}{\sqrt{T}}\right)$$

which is an anytime $\mathcal{O}\left(1/\sqrt{T}\right)$ convergence rate, but does not revert to a $\mathcal{O}(1/T)$ rate when interpolation holds. This is an instance of a rate which does not benefit from interpolation. Another choice is to fix a finite horizon $T \geq 1$ and to take $\gamma \propto \frac{1}{\sqrt{T}}$ which guaranteees (see Theorem 5.5 in Garrigos & Gower (2023)) a bound

$$\mathbb{E}\left[f(\bar{x}_T) - f(x_*)\right] \leq \mathcal{O}\left(\frac{1 + \sigma_*^2}{\sqrt{T}}\right)$$

Table 2: A summary of related work and conceptual differences to our approach and the work in AcceleGrad Levy et al. (2018), UniXGrad Kavis et al. (2019), AC-FGM Li & Lan (2023), Prodigy Mishchenko & Defazio (2024), and USFGM Rodomanov et al. (2024).

| Algorithm | Last iterate | Smooth problems | Non-smooth problems | Unbounded domain | Stoch. gradients | Can increase step size |
|---|---|---|---|---|---|---|
| AcceleGrad | ✗ | ✗[(1)] | ✓ | ✗ | ✓ | ✗ |
| UniXGrad | ✗ | ✓ | ✓ | ✗ | ✓ | ✗ |
| AC-FGM | ✗ | ✓ | ✓ | ✓ | ✗ | ✓ |
| Prodigy | ✗ | ✓ | ✓ | ✓ | ✗ | ✓ |
| USFGM | ✓ | ✓ | ✓ | ✗ | ✓ | ✗ |
| SPS* (our result) | ✗ | ✓ | ✓ | ✓ | ✓ | ✓ |
| IAM (ours) | ✓ | ✓ | ✓ | ✓ | ✓ | ✓ |

[(1)] AcceleGrad's smooth analysis is for deterministic problems.

which has the same issues as the above mentioned result, on top of not being an anytime convergence rate. Finally, if one can estimate $\sigma_*^2$, then it is possible to set the stepsize $\gamma$ as in Theorem E.1 to obtain a rate

$$\mathbb{E}\left[f(\bar{x}_T) - f(x_*)\right] \leq \mathcal{O}\left(\frac{1}{T} + \frac{\sigma_*^2}{\sqrt{T}}\right)$$

which is a finite horizon (complexity) rate.

Let us now discuss and compare our result in Theorem 3.3 about rates for Momentum with Polyak stepsizes, an algorithm that we call IAM (see Algorithm 1). It is known that Momentum (as rewritten in (3)-(4)) with a learning rate $\eta \leq \frac{1}{4L}$ enjoys a finite iterate complexity rate (see Theorem 7.4 in Garrigos & Gower (2023)):

$$\mathbb{E}\left[f(x_T) - f(x_*)\right] \leq \frac{\|x_0 - x_*\|^2}{\eta(T+1)} + 2\eta\sigma_*^2.$$

This is finite horizon complexity rate, which is not adaptive with respect to $L$. Instead, we show that using a Polyak stepsize as defined in Algorithm 1 allows for a better anytime convergence rate, which is adaptive to $L$.

### E.4 COMPARISON OF ADAPTIVE METHODS

In Table 2 we make a qualitative comparison between our methods SPS* and IAM and other adaptive methods. Table 2 highlights how SPS*, to the best of our knowledge, is the only stochastic adaptive stepsize that has favorable convergence rates in the smooth and non-smooth problems, without having to modify the method, admits an unbounded domain and can increase the stepsize (not monotonic). The IAM method shares the same benefits, but also has a fast last iterate convergence, as opposed to the average iterate for SPS*.

### E.5 SPS* IN PRACTICE: APPROXIMATING $f_\xi(x_*)$ AND SAFE-GUARDS

In practice, outside of the interpolation regime, it is unlikely that we would have access to $f_\xi(x_*)$. To derive a practical method that is more generally applicable, we would need to estimate $f_\xi(x_*)$. Let us call this estimate $\ell_\xi^*$, and consider the step size

$$\gamma_t^{\text{SPS}} := \frac{(f_\xi(x_t) - \ell_\xi^*)_+}{\|g_t\|^2}. \tag{57}$$

The estimates $\ell_\xi^*$ would have to be *underestimates*, otherwise the resulting method would stop early. Indeed, as $x_t \to x_*$ we have that $f_\xi(x_t) \to f_\xi(x_*)$, and the step size (57) would be zero before reaching convergence.

There are two natural underestimates for $f_\xi(x_*)$. The first is to use $\inf f_\xi$. This is the approach used in `SPSmax` (Loizou et al., 2021). The advantage of using $\inf f_\xi$ is that it often can be computed, indeed if no weight decay is being used (no L2 regularization), then often $\inf f_\xi = 0$. Which brings us to the second approach, which is to simply use 0 as an underestimate, which holds for the ubiquitous case of having a positive loss (Berrada et al., 2020; Orvieto & Xiao, 2024).

An issue with using an underestimate is that the step size (57) can become too large, potentially even being unbounded if $\|g_t\| \to 0$, which could lead to divergence.

To safeguard against taking exceeding large step sizes, we can use clipping Loizou et al. (2021), dampening Orvieto & Xiao (2024), or a combination of both Berrada et al. (2020). By clipping, we mean to take the minimum between the stepsize in (57) and a hyperparameter $\gamma_b > 0$ as is done in Loizou et al. (2021) in the $SPS_{\max}$ method[4]

$$\gamma_t^{\text{SPS}_{\max}} := \min\left\{ \frac{(f_\xi(x_t) - \ell_\xi^*)_+}{\|g_t\|^2}, \gamma_b \right\}. \tag{58}$$

We refer to dampening by adding an additional constant $\epsilon$ to the denominator, as is done in Orvieto & Xiao (2024), Gower et al. (2022) and Berrada et al. (2020):

$$\gamma_t^{\text{SPS}dam} := \frac{(f_\xi(x_t) - \ell_\xi^*)_+}{\|g_t\|^2 + \epsilon}. \tag{59}$$

In particular in Orvieto & Xiao (2024), this dampening parameter depends in the iteration and is proportional to $f_\xi(x_t)$.

Thus we can view several practical variants of `SPS` as approximations of `SPS*`, where $f_\xi(x_*)$ is replaced by an underestimate, and a further safeguard is included to avoid large step sizes. These safeguards can also be motivated through a variational viewpoint based on solving relaxations of the interpolation condition (Gower et al., 2022).

# F  EXPERIMENTS

## F.1  NON-LIPSCHITZ NON-SMOOTH CONVEX PROBLEM

To model discrete events with a Poisson regression, we need to solve

$$\min_{w \in \mathbb{R}^d} \frac{1}{n} \sum_{i=1}^n \left( \ell(w^\top x_i) - y_i \log\left( \ell(w^\top x_i) \right) \right), \tag{60}$$

where $\ell : \mathbb{R} \mapsto \mathbb{R}$ is called the link function. One of the most commonly used link functions is the exponential function $\ell(z) = \exp(z)$. With this link function (60) becomes

$$\min_{w \in \mathbb{R}^d} \frac{1}{n} \sum_{i=1}^n \left( \exp(w^\top x_i) - y_i w^\top x_i \right). \tag{61}$$

We fit two different data sets. The first data set is on diabetes patients sourced from (Efron et al., 2004), which is a medical dataset containing information on 442 patients ($n$), each described by 10 physiological and lifestyle features ($d$). The second data set is a bike sharing records (Fanaee-T & Gama, 2014) in Washington, D.C., over a two-year period (2011-2012). It includes a total of 17,379 data points, and 12 features such as weather conditions, seasonal information, and temporal data. The target variable is the count of total bike rentals on an hourly basis.

As a baseline, we ran `L-BFGS` (Liu & Nocedal, 1989) in full batch mode, and `SGD` with constant learning rate tuned across

$$\gamma \in 0.001 \cdot \{0.01, 0.1, 0.5, 1.0, 2.0, 5.0, 20, 50\}.$$

---

[4]Though $SPS_{\max}$ has an additional constant $c$ in $\frac{(f_\xi(x_t) - \ell_\xi^*)_+}{c\|g_t\|^2}$.

Each method was given the same budget in terms of epochs. To highlight how important the choice of the learning rate is, in Figure 4 we plot the resulting loss ($y$-axis) of the last iterate of each method for different learning rates ($x$-axis). We find that the `IAM` method converges to a loss that is comparable to `LBFGS` and `SGD` with the best possible learning rate. Furthermore, `IAM` is the only method guaranteed to converge on this non-smooth and non-Lipschitz objective.

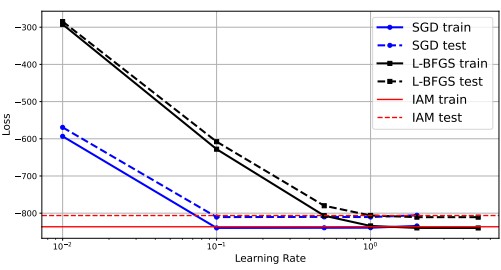
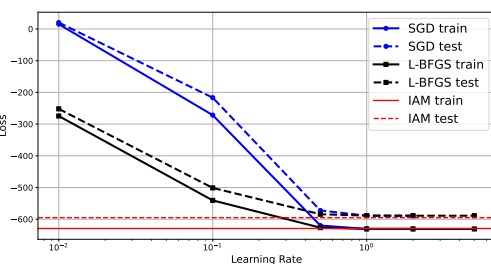

Bike Sharing Data, 7 epochs

Figure 3: Diabetes Data, 15 epochs

Figure 4: Sensitivity to learning rate for each method. Larger learning rates diverged.

### F.2 MISSPECIFICATION OF $f_\xi(x_*)$

In numerous machine learning applications a lower bound of $f_\xi(x_*)$ is known a priori, because loss functions are typically non-negative. We study the following three versions of `IAM`:

- *theoretical* version where we specify correctly $f_{\xi_t}(x_*)$ in every iteration $t$, computed from the oracle values $f_i^*$, $i \in [n]$,
- *averaged* version, where we specify $f_{\xi_t}(x_*)$ with $f(x_*)$ in every iteration,
- *lower-bound* version, where we specify $f_{\xi_t}(x_*)$ with zero.

**Description of experimental setup.**  Consider the following problem setup, which is adopted from Orvieto et al. (2022): solve

$$\min_{x \in \mathbb{R}^d} \frac{1}{n} \sum_{i=1}^n f_i(x), \quad f_i(x) := (x - x_*^i)^T H_i (x - x_*^i) + f_i^*,$$

where $H_i \in \mathbb{R}^{d \times d}$ are symmetric positive definite matrices and $x_*^i \in \mathbb{R}^d$. This is clearly an instance of (1), where $\mathcal{D}$ is the uniform distribution over $[n]$ and $f_\xi(x) = f_i(x)$, and it holds $f(x) = \frac{1}{n} \sum_{i=1}^n f_i(x)$.

We consider two cases, (i) the interpolated case with $x_*^i = \bar{x}$ for all $i \in [n]$, and (ii) $x_*^i = \bar{x} + 0.05\varepsilon_i$, where $\varepsilon_i \in \mathbb{R}^d$ is standard normal. Following Orvieto et al. (2022), we generate $H_i = A_i^T A_i/(3d)$ where the entries of $A_i \in \mathbb{R}^{3d \times d}$ are standard normal. We generate $f_i^*$ from a uniform distribution with mean $0.5$ and standard deviation $\nu$, followed by truncation at zero to make sure all $f_i^*$ are non-negative.

Note that in case (i) $\bar{x}$ is the minimizer of $f$ and of each $f_i$. Further, $f_i(x_*) = \inf_{x \in \mathbb{R}^d} f_i(x) = f_i^*$, and $f(x_*) = \frac{1}{n} \sum_{i=1}^n f_i^* = \inf_{x \in \mathbb{R}^d} f(x)$. In the other case (ii), we compute the solution $x_*$ by solving a linear system, and then compute $f_i(x_*)$. We always compute $f_\xi(x_*)$ by averaging $f_i(x_*)$ over the corresponding mini-batch.

We vary the standard deviation $\nu \in \{0.01, 0.1\}$ and the batch size $b \in \{4, 16\}$.

We run all versions of `IAM` with $\lambda_t = 9$ for all $t \geq 0$, as suggested by our convergence Theorems 3.3 and 3.2. As a baseline, we compare to `SGD-M` with constant learning rate and momentum $\beta = 0.9$. We set the learning rate to the theoretical value $\frac{1}{4L_{\max}}$ (cf. Sebbouh et al. (2021)), where $L_{\max} := \max_{i=1,\ldots,n} L_i$ and $L_i := 2\lambda_{\max}(H_i)$ denotes the smoothness constant of $f_i$ (here $\lambda_{\max}$ denotes the largest eigenvalue).[5] We further compare to `MoMo` that has access to $f_\xi(x_*)$, cf. Schaipp et al. (2024, Eq. 17).

---

[5]Note that in `Pytorch` this requires setting `dampening=0.9`.

**Discussion.** In the interpolated case, see Figure 5, the theoretical version of `IAM` matches the rate of `SGD-M` *without any tuning*. However, if $f_\xi(x_*)$ is mis-specified, the convergence stales. This effect is more pronounced if the noise is large, or the batch size is small. In the non-interpolated case, see Figure 6, we observe that the theoretical version of `IAM` obtains a smaller final loss than `SGD-M`. This matches our theoretical result in the smooth setting, where we showed that we get convergence even if $\sigma_*^2 > 0$.

Compared to `MoMo`, we observe roughly the same convergence behaviour, with `IAM` typically having a slightly bigger slope. As a side note, we observe that `MoMo` also converges without interpolation, even though this case is not covered by the theory of Schaipp et al. (2024).

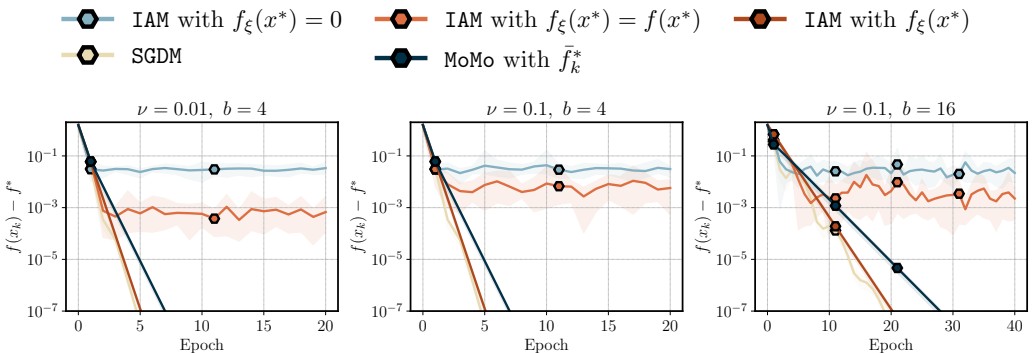

Figure 5: **Interpolation true:** `IAM` with the correct $f_{\xi_t}(x_*)$ converges as fast as `SGD-M` with the theoretical step size $\frac{1}{4L_{\max}}$. When $\nu$ is small **(left)**, the initial progress of `IAM` with the average $f(x_*)$ is equally good, before it stales. For $\nu$ large, the convergence stales earlier **(midlle)**. Increasing the batch size **(right)** slightly increases the gap between `IAM` with $f_{\xi_t}(x_*) = 0$ and $f_{\xi_t}(x_*) = f(x_*)$.

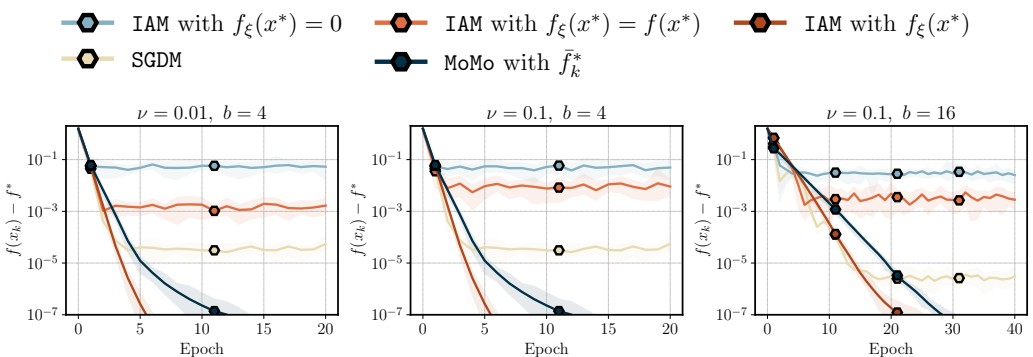

Figure 6: **Interpolation false:** see caption of Figure 5.

### F.3 SUPPLEMENTARY MATERIAL ON DISTILLATION EXPERIMENT

Here we provide the complete details of our distillation experiments in Section 4.1, together with some additional plots.

**Datasets and models.** The datasets we consider are below. We used the `GPT2Tokenizer` from the Transformers library.

- `tinyShakespeare` (Karpathy, 2015): 40 000 lines from Shakespeare plays. The dataset has 303 688 tokens.
  Source: https://huggingface.co/datasets/karpathy/tiny_shakespeare

- `PTB` (Penn Treebank) (Marcus et al., 1993): The dataset contains $1\,094\,404$ tokens.
  Source: https://huggingface.co/datasets/ptb-text-only/ptb_text_only

- `Wikitext2` (Merity et al., 2016): This is a subset of a 100 million token large collection of featured articles from Wikipedia. The dataset contains $2\,389\,828$ tokens.
  Source: https://huggingface.co/datasets/Salesforce/wikitext

We use the module `GPT2LMHeadModel` from the HuggingFace `transformers` library (Wolf et al., 2020) to define our `GPT2` models [link]. The teacher model we use for `tinyShakespeare` and `Wikitext2` is the `gpt2-large` configuration within this module, which has 774 million parameters, and was pretrained by a team from OpenAI (Radford et al., 2019). We use the same tokenizer from the teacher model for the student model. For the `PTB` dataset, we use a different teacher model, as we found that `gpt2-large` had a poor fit, with the loss being above 5.0 on this dataset. So instead, we used the `GPT-J-6B` model (Wang, 2021), which has 6 billion parameters [link].

For the student model, we specify the configuration in Table 3.

Table 3: Parameters of the Student GPT2 model

| Dataset | Embedding size | Number of layers | Number of attention heads |
|---|---|---|---|
| tinyShakespeare | 768 | 2 | 4 |
| PTB | 768 | 2 | 4 |
| Wikitext2 | 1200 | 12 | 12 |

**Hyperparameter tuning.** For our methods `IAM` (Algorithm 1) and `IAM-Adam` (Algorithm 2) we set $\lambda_t = 9$, which corresponds to using momentum $\beta = 0.9$ if the learning rates were constant, see Lemma D.1. Note that there is no hyperparameter tuning at all for `IAM` and `IAM-Adam`.

For the baseline methods `SGD` and `Adam`, we do the following tuning: we run bot `SGD` and `Adam` with constant learning rate and with a *warmup+cosine decay* schedule (Loshchilov & Hutter, 2017). This schedule does a linear warmup over the first 20% of iterations to a peak learning-rate $\gamma$, then performs a cosine decay to 0 over the remaining steps. For `Adam`, we set the learning rate to its default value of $10^{-3}$ for the constant schedule, and we set $\gamma = 1.5 \times 10^{-3}$ for the *warmup+cosine decay* schedule.

For `SGD` the learning rate needs to be tuned to get a reasonable performance: for the constant schedule, we chose the best-performing learning rate from the set

$$\gamma_{\text{constant}} \in \{0.0001, 0.001, 0.01, 0.05, 0.1, 0.2\}.$$

When using a scheduler with `SGD`, we take the best-performing value $\gamma_{\text{constant}}$ and then independently tune the peak learning rate within the set

$$\gamma_{\text{constant}} \cdot \{1.2, 1.5, 2, 3, 5\}.$$

For `SGD` we use a momentum parameter of 0.9. In the `Pytorch` implementation of `SGD`, we also set the `dampening` parameter to 0.9 to ensure comparability of the tuned learning rate to the one of `IAM`.

**Relationship to existing distillation techniques.** In this paragraph, we aim to give a short overview over various distillation techniques which often vary in terms of their general setup and loss function. However, as model distillation is not the main focus of this paper, we point to the references below for additional background. In their seminal work, Hinton et al. (2015) propose to minimize the KL divergence between the teacher and student output probabilities. Follow-up works use a loss function that combines KL divergence and the standard loss for the student task (e.g., cross-entropy loss for classification, squared loss for regression) (Romero et al., 2015). On the other hand, Hsieh et al. (2023) propose to use the teacher output as surrogate labels in case of unavailable labeled training data for the students. We also refer to Beyer et al. (2022) for an overview of training techniques that improve the distillation performance.

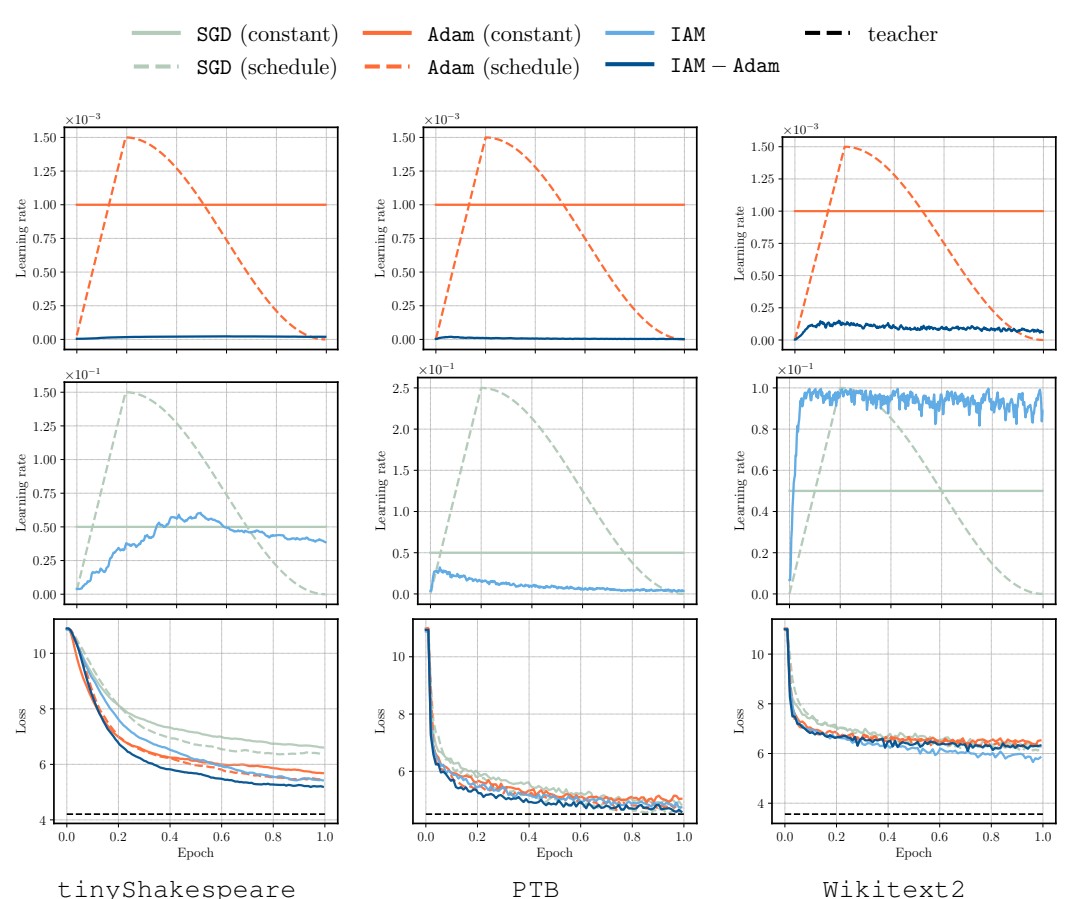

Figure 7: Full display of Figure 1. Adaptive learning rate of `IAM-Adam` compared to `Adam` **(top)**, of `IAM` compared to `SGD` **(middle)**, and the cross-entropy training loss **(bottom)**. Black line marks the average teacher loss.

The distillation setup that we propose in this paper is slightly different: we use only the final batch loss of the teacher model. The reason for this is that the `IAM` methods we investigate rely on an accurate guess of the optimal batch loss $f_\xi(x_*)$. In the distillation setting, we can leverage the pretrained teacher model in order to approximate the optimal batch loss values. The notion of distillation we use here might of independent interest, as it only needs access to the final batch loss value, but not the output probabilities of the model (the *logits*) nor its weights.

**Additional plots.** In Figure 7 we give the full plot of our distillation experiments, including the evolution of the learning rates for `IAM` and `IAM-Adam`.

In Figure 8 we give the distillation of several different small `GPT2` models for the `tinyShakespeare` data set.

Finally, we also include a experiment on distilling a large data set, the `FineWeb-1b` data set in Figure 9. Here we found, contrary to our other experiment, that `IAM` (and `IAM-Adam`) were not the most robust of all methods, but did achieve the best validation loss for the best learning rate.

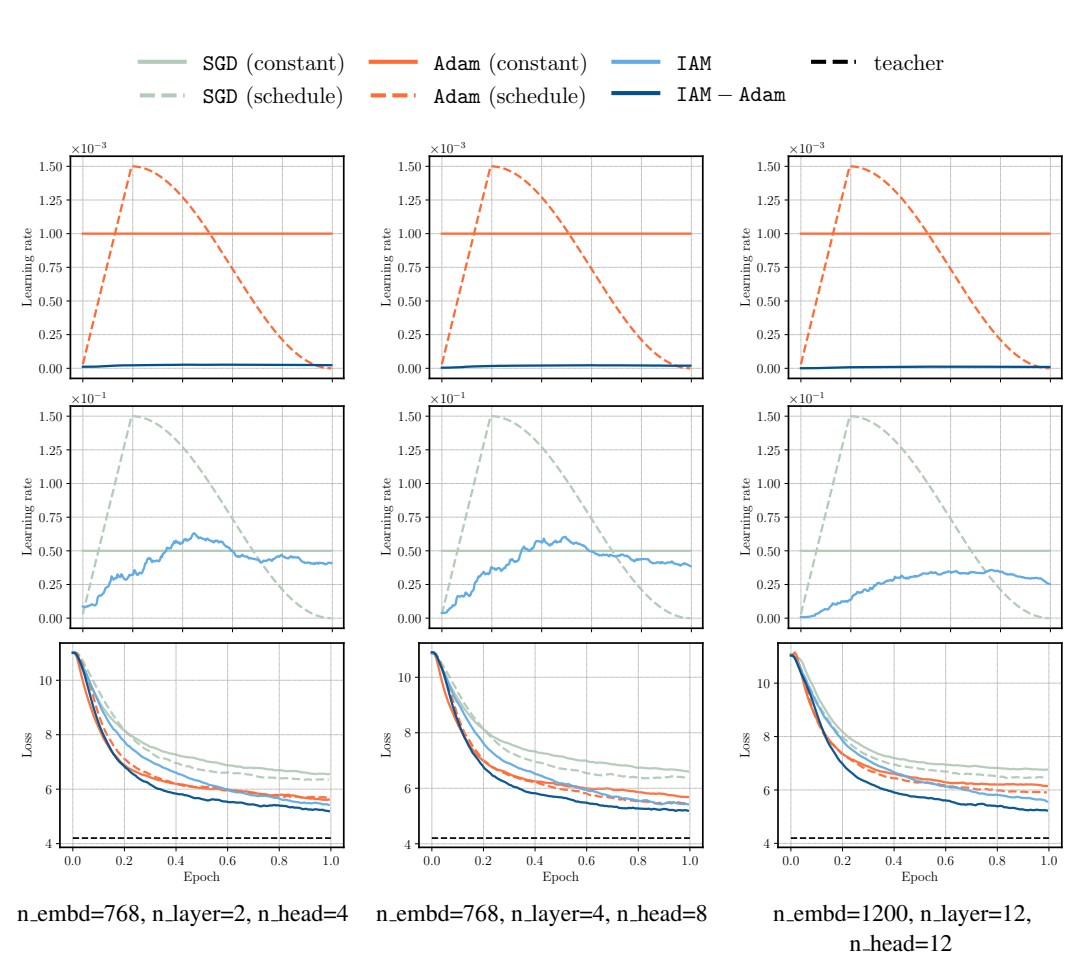

Figure 8: Distilling `gpt2-medium` into successively larger student models for the `tinyShakespeare` dataset.

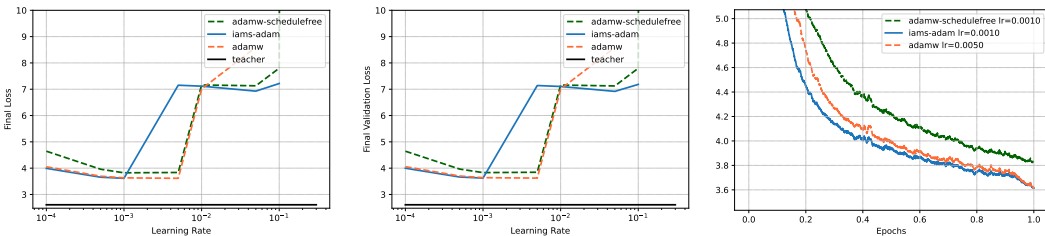

Figure 9: Distilling a teacher `GPT2` on the `Fineweb-1B`.

