# OpenReview forum: "Analysis of an Idealized Stochastic Polyak Method and its Application to Black-Box Model Distillation"
_ICLR.cc/2026/Conference — Submitted to ICLR 2026_

### Official Review · Reviewer_LzVz · 2025-10-29

**Soundness:** 3
**Presentation:** 3
**Contribution:** 2
**Rating:** 4
**Confidence:** 3

**Summary:**

The paper presents a general convergence theorem for an idealized stochastic Polyak step size method, called SPS*. The analysis is for convex function with a local expected gradient bound, it covers both locally smooth and locally Lipschitz losses. SPS* is idealized since it requires access to the loss at the optimal solution. The paper establishes O(1/sqrt(t)) convergence rate for globally Lipschitz functions and Polyak step size. The paper also combines SPS* with momentum, also establishing  O(1/sqrt(t)) convergence rate. There are some experiments, showing comparable performance to SGD and ADAM, but less sensitivity to parameter tuning.

**Strengths:**

There is clear novelty in the introduction of the idealized stochastic Polyak step size (SPS*). This particular formulation, together with the general convergence theorem under minimal assumptions, appears to be new and fills a gap in existing research on adaptive step sizes. The authors extend the concept of Polyak step sizes in a way that achieves optimal convergence rates and integrates naturally with momentum, which enhances its theoretical appeal.

Another strength of the paper is in its focus on reducing sensitivity to hyperparameter tuning, a long-standing issue in stochastic optimization. By proposing a theoretically grounded approach that adapts the step size automatically, the work contributes to developing more robust algorithms that can perform well without extensive manual tuning.

**Weaknesses:**

One weakness of the paper is that SPS* is presented in an idealized form requiring access to the loss evaluated at the optimal solution for each training batch. This makes it impractical in real-world scenarios, where such information is unavailable, and limits the immediate applicability of the proposed method. Although the authors acknowledge this and suggest directions for more practical adaptations, the gap between the idealized theory and an implementable algorithm remains significant.

Moreover, the main application discussed where this step-size selection is possible is model distillation. As far as I know, model distillation is always associated with deep learning where the loss functions are non-convex. All the theoretical results are for convex problems, so the paper, or at least the theory, does not apply to this main application example considered in the paper. So it seems a bit of a speculative contribution, not a good alignment between the mutation and what is actually done.

It could be discussed better what is the theoretical contribution of the paper, or compare better to the state of the art. The paper is not improving the convergence rate compared to the state of the art, it is just establishing for a particular SPS similar convergence rate can be achieved, or that is my understanding. It would be good if the theoretical results could be put in better context with existing literature.

**Questions:**

For Theorem 3.2 it is highlighted that the convergence rate is for the last iterate. Which is in some sense correct, there is some convergence for last iterate, but since on the left side there is the average of tB(t) for all previous iterations, it is difficult to see how much gap will be created by this term, and it seems to have similar effect as the averaging, or even worse, since to me, this might grow to infinity as T grows, since t is in there.

---

> ### Author Response · Authors · 2025-11-21
> **Response to Reviewer LzVz**
>
> Thank you for your time. We hope that you will re-consider your score given our responses.
>
> > **Q1.**  One weakness of the paper is that SPS* is presented in an idealized form requiring access to the loss evaluated at the optimal solution for each training batch. This makes it impractical in real-world scenarios, where such information is unavailable, and limits the immediate applicability of the proposed method. Although the authors acknowledge this and suggest directions for more practical adaptations, the gap between the idealized theory and an implementable algorithm remains significant.
>
> **A1.** We agree that gaps between the theoretically guaranteed result and the practical applications are always frustrating; however, we would argue that such gaps exist for almost any theoretical result that aims for practical applicability. Our paper clearly addresses these limitations. The claim that our method is impractical in real-world scenarios is incorrect. One of the reasons why large vision models have been successful in the last 10 years is because they reached the interpolation regime, which is exactly where our algorithm can be implemented trivially. On top of that, our numerical experiments show – despite this gap – that the method works well in practice and is very useful for avoiding hyperparameter tuning.
> If the reviewer can specify how we could make this argument more convincing than what we already provided, we are happy to work on it. We believe that the limitation of our idealized method does not per se justify a low score, exactly because we clearly address it and show that the method still can be very practical in certain situations.
>
> > **Q2.** Moreover, the main application discussed where this step-size selection is possible is model distillation. As far as I know, model distillation is always associated with deep learning where the loss functions are non-convex. All the theoretical results are for convex problems, so the paper, or at least the theory, does not apply to this main application example considered in the paper. So it seems a bit of a speculative contribution, not a good alignment between the mutation and what is actually done.
>
> **A2.** We first would like to point out that our paper contains experiments for a convex problem in Appendix F.1, where the test problem is highly aligned with the theoretical setup. Our (non-convex) distillation problems additionally show that our IAM method can be applied on non-convex problems while performing very well, despite lacking theoretical guarantees (as is true for most optimization methods on non-convex problems). We kindly disagree that it is a weakness if we test the proposed method outside of the theoretical framework, and in fact believe that this even strengthens our contributions and the arguments in favor of our methods.
>
> > **Q3.**  It could be discussed better what is the theoretical contribution of the paper, or compare better to the state of the art. The paper is not improving the convergence rate compared to the state of the art, it is just establishing for a particular SPS similar convergence rate can be achieved, or that is my understanding. It would be good if the theoretical results could be put in better context with existing literature.
>
> **A3.** We discussed the theoretical contribution extensively, and compiled a comprehensive Table 1 in the paper which provides an overview of how our new results compare to existing results. Our contributions in relation to previous work is also summarized in Section 1.2 and in even more detail in Appendix E. We can also give a highly compressed summary of our theoretical contributions here: we prove optimal rates in the nonsmooth case under lighter assumptions (Lipschitz continuity only required on a bounded set), and a anytime 1/sqrt(t) rate for Polyak-type step size methods for the first time. Again, compared to other methods, our result requires assumptions to only hold on a bounded set, which is much easier to satisfy.
>
> > **Q4.**  For Theorem 3.2 it is highlighted that the convergence rate is for the last iterate. Which is in some sense correct, there is some convergence for last iterate, but since on the left side there is the average of tB(t) for all previous iterations, it is difficult to see how much gap will be created by this term, and it seems to have similar effect as the averaging, or even worse, since to me, this might grow to infinity as T grows, since t is in there.
>
> **A4.** The Bregman divergences are always non-negative, therefore the presence of the Bregman-terms makes this bound actually tighter, and not looser. The bigger are those terms, the *better* is our bound. Due to the non-negativity of the Bregman terms, we could as well drop them and the bound would remain still correct. If the reviewer thinks that those terms can be confusing for the reader, we would be happy to do so.

---

### Official Review · Reviewer_BoT6 · 2025-10-29

**Soundness:** 3
**Presentation:** 3
**Contribution:** 3
**Rating:** 6
**Confidence:** 3

**Summary:**

The paper studies the convergence rates of an idealized Polyak step size for stochastic gradient descent, which uses the instantaneous loss at the global optimum. For Lipschitz convex, smooth convex, and strongly convex loss functions, the idealized algorithm converges at the optimal rates, while being adaptive to problem parameters. The paper also presents an anytime version of plain SGD to obtain anytime convergence. The paper demonstrates the application of the proposed method for the interpolation setting and a distillation setup.

**Strengths:**

1. The paper is clearly written. It provides sufficient review of the literature, and provides detailed proofs for most statements, in a structured manner. I like how the authors first derive a master theorem 2.1 with a general conditions, and then specializes it to different settings. I went through most proofs in Appendices B and C smoothly.

2. As far as related works mentioned in the paper, the rates achieved by the current paper is as good or better in settings listed in Table 1.

**Weaknesses:**

While the convergence rates are good, the disadvantage is the assumption on knowing $f_t (x_*)$ in equation (2). While I do appreciate the empirical results of the method on two tasks, theoretically the assumption limits the applicability of the method to general machine learning losses. Given the assumption, it is less surprising that the method achieves better rates than previous methods, which use more easily available quantities in their stepsizes.

**Questions:**

1. See weakness above.

2. With acceleration, in the $L$-smooth setup, SGD can achieve $O(\frac{LD^2}{t^2})$ rate *on the optimization error* (first term in the rate) while the statistical error (second term) still decays like $O(\frac{1}{\sqrt{t}})$. Do you think it is possible to achieve that with the proposed method?

---

> ### Author Response · Authors · 2025-11-21
> **Response to Reviewer BoT6**
>
> We thank the reviewer for their careful assessment of the paper and are particularly happy to hear about the clear and structured presentation. We address all questions below.
>
> > **Q1.**  While the convergence rates are good, the disadvantage is the assumption on knowing $f\_t(x\_*)$ in equation (2). While I do appreciate the empirical results of the method on two tasks, theoretically the assumption limits the applicability of the method to general machine learning losses. Given the assumption, it is less surprising that the method achieves better rates than previous methods, which use more easily available quantities in their stepsizes.
>
> **A1.** We agree with the reviewer that this assumption is strong, however we want to remark that our current submission addresses this in a very clear and transparent way (in the title, abstract and the rest of the paper). Further, we have shown that there indeed exist practical situations where this assumption is justified, in both interpolation regimes (image tasks) and model distillation.
>
> > **Q2.**  With acceleration, in the $L$-smooth setup, SGD can achieve $O(\frac{LD^2}{t^2})$ rate on the optimization error (first term in the rate) while the statistical error (second term) still decays like $O(\frac{1}{\sqrt{t}})$. Do you think it is possible to achieve that with the proposed method?
>
> **A2.** We did study Nesterov’s momentum method with our new stepsize, but the resulting convergence rate was $O(1/t)$ instead of $O(1/t^2)$, which is why we did not include the corresponding theory in our submission. Surprisingly, this was not only a proof issue as when we tested the method numerically, it did converge with the $O(1/t)$ rate despite using momentum. The reason behind this is that our approach is based on monotonicity of distances, which is a property not satisfied by the accelerated methods, since they are inherently oscillating due to being discretized solutions to a second-order ODE.

---

### Official Review · Reviewer_hVNM · 2025-10-31

**Soundness:** 3
**Presentation:** 3
**Contribution:** 3
**Rating:** 4
**Confidence:** 3

**Summary:**

The paper presents a theoretical analysis of an idealized stochastic Polyak step size SPS*, an adaptive method for stochastic optimization. The method is called idealized because it assumes access to the loss value at the optimal solution for every training batch.

The authors provide a convergence theorem for SPS* and demonstrate that it achieves optimal or best known anytime convergence rates for various classes of convex functions, including Lipschitz and smooth functions. These results only require local assumptions in a ball around the solution.

The paper introduces IAM, a new variant of SPS* that incorporates momentum. They prove that the last iterate of IAM achieves the same fast and adaptive convergence rates that SPS* achieves for the average iterate, which is often more practical.

The primary practical application demonstrated is in knowledge distillation. The authors propose using the loss of a large pre-trained teacher model on a given data batch as an approximation for the optimal loss required by SPS* to train a smaller student model. IAM method effectively trains a student GPT-2 model on several language datasets, achieves performance competitive with fine-tuned Adam and SGD, yet IAM does not require tuning.

**Strengths:**

The paper has a strong theory. The convergence analysis of SPS* under local assumptions is a valuable contribution to the optimization field. The variant of knowledge distillation is novel, practical and interesting. The IAM method works well without tuning against strong tuned baselines on distilling tasks.

**Weaknesses:**

The practical use-case of the method (distilling models) seems niche to me. The optimal loss value for every training batch is unknown for real ML problems which makes the method indeed idealized.

Further, the need for the parameter tuning is replaced with selecting and deploying a suitable teacher model. It feels like a more complex dependency to me, not a hyperparameter-free approach at all. This is a far more significant hyperparameter choice than a learning rate.

Also, the assumption that teachers loss is approximately a students loss at optimum seems like a strong one. If the student model is much smaller than the teacher, it may be incapable of achieving it.

I think the loss landscapes of teacher and student can be very different and teacher can be a poor optimization guide.

**Questions:**

How does the optimizer behave in the regime when the student model is much smaller than the teacher?

How does the method degrade as the student capacity to match the teacher loss diminishes?

What happens if the teacher model is only slightly better than a baseline student?

What happens when their architectures are different?

Will the convergence guarantee still hold under the approximation of a batch optimum, or does it converge to a neighborhood of the solution?

---

> ### Author Response · Authors · 2025-11-21
> **Responses for Reviewer hVNM**
>
> Dear Reviewer,
>
> we thank you for the comments and questions on our submission. We are happy to hear that the theoretical contributions and the application on model distillation are considered to be interesting and novel. Most of the questions seem to revolve around the choice for the teacher model and are addressed below.
>
> **Q1.** The practical use-case of the method (distilling models) seems niche to me. The optimal loss value for every training batch is unknown for real ML problems which makes the method indeed idealized.
>
> **A1.** While agreeing that this assumption is a limitation, we think that the current submission makes this very clear and transparent (in the title, abstract and the rest of the paper). As the reviewer pointed out, we (i) can show novel and interesting theoretical results and (ii) provide novel and practical applications where the assumption of knowing the optimal loss is (approximately) satisfied. Furthermore, there is another practical use-case, that of models that typically interpolate the data, such as many vision models. To highlight this, we will add experiments on standard vision benchmarks to the camera ready to highlight this. We therefore hope that this point is not a sufficient reason to reject the paper.
>
> **Q2.** The need for the parameter tuning is replaced with selecting and deploying a suitable teacher model. This is a far more significant hyperparameter choice than a learning rate.
>
> **A2.** On this point we respectfully disagree. Tuning the learning rate and schedule usually requires many different training runs, and therefore a high computational burden. For selecting a teacher model, it is sufficient to choose any model that should solve the same task with at least the same capacity. This is very easy to do, for example by looking up available models on Huggingface, and using models that are larger in size and trained on more data. Due to the scaling laws, this will reliably ensure that the teacher model has at least the same capacity. We ourselves chose a teacher model without any  “tuning” for the selection of the teacher model, but instead used the most obvious choice (pretrained model of the same architecture with more parameters).
>
> **Q3.** The assumption that teachers loss is approximately a students loss at optimum seems like a strong one. If the student model is much smaller than the teacher, it may be incapable of achieving it.
>
> **A3.** We refer to our response above. While it is true that the student model might not reach the teacher loss even at convergence, our experiments show that even in this case, the teacher loss is a good enough approximation, and can improve over a manually tuned baseline (see Figure 1 Wikitext2).
>
> **Q4.**  I think the loss landscapes of teacher and student can be very different and teacher can be a poor optimization guide.
>
> **A4.** Teacher and student model in general do not live in the same parameter space, and therefore we do not know how their loss landscapes could be compared in a sensible way. In any case, for our methods, we are only interested in the loss functions of the teacher model (not its parameters). Could the reviewer kindly provide additional reasoning why the teacher might be a poor optimization guide? As our experiments show, using the teacher loss is indeed a useful guide for setting the learning rate.

---

> ### Author Response · Authors · 2025-11-21
> **Response part II**
>
> > **Q5.**  How does the optimizer behave in the regime when the student model is much smaller than the teacher?
>
> **A5.** This is a good question. We looked into this question in Figure 8 in the appendix. We fixed the teacher as a gpt2-medium model (355 million parameters), and tested with several student models with
>
> | Embed, Layers, Heads | # Parameters |
> |------------------------|--------------|
> | 768, 2, 4 heads      | 52.8M        |
> | 768, 4, 8 heads      | 67.0M        |
> | 1200, 12, 12 heads   | 267.9M       |
>
>
> For all these smaller sizes student models IAM-Adam and IAM performed very well.  Only
> for the two smallest student models (52.8M and 67.0M)  did AdamW almost match the performance of IAM.
>
> > **Q6.**  How does the method degrade as the student capacity to match the teacher loss diminishes?
>
> **A6.** The methods degraded quite gracefully as the we shrunk the student capacity, see the previous response and Figure 8 in the paper.
>
> > **Q7.** What happens if the teacher model is only slightly better than a baseline student?
>
> **A7.** This setting occurred  in PTB experiment in Figure 1, and in the right most panel in Figure 8. We still observed favourable performance in this setting as well.
>
> > **Q8.** What happens when their architectures are different?
>
> **A8.** This is a reasonable question. Ultimately, it doesn’t matter if the student and teacher have completely different architectures. It only matters that the student cannot achieve a better loss than the teacher. Furthermore, the architecture of the teacher for PTB is a GPT-J model, which is similar but not the same as the GPT2 model. GPT‑J mainly uses parallel residual block instead of a sequential pre‑LN block in GPT2, and  GPT-J uses duplicated LayerNorms (two per block) , whereas GPT2 has one LayerNorm before each sublayer.
>
> > **Q9.** Will the convergence guarantee still hold under the approximation of a batch optimum, or does it converge to a neighborhood of the solution?
>
> **A9.**  For the convergence, we indeed need the value of the batch loss at the global minimizer $x\_*$, which might be different from the minimal value of the batch. Prior work [1,2] has shown that when using the minimal batch loss instead, a neighborhood of convergence is unavoidable. Our work is the first to show that this neighborhood can be avoided *if* we have access to the batch loss at the global minimum.
>
> [1] Orvieto, A. et al, N. Dynamics of SGD with stochastic Polyak stepsizes: Truly adaptive variants and convergence to exact solution. 2022.
>
> [2] Orabona, F., D'Orazio, R.. New Perspectives on the Polyak Stepsize: Surrogate Functions and Negative Results. 2025.

---

### Official Review · Reviewer_oMJJ · 2025-11-01

**Soundness:** 3
**Presentation:** 3
**Contribution:** 3
**Rating:** 6
**Confidence:** 3

**Summary:**

This paper proposes a variant of Polyak stepsize for stochastic convex optimization, named ideal stochastic Polyak stepsize (SPS*), which computes the on-the-fly step size as $\gamma_t = (f_t(x_t) - f_t(x_*))_+ / \\|g_t\\|^2$ with the caveat that the minimum to the stochastic loss $f\_t$, namely $ f\_t(x\_ * ) $, is known. It also incorporates SPS* with momentum and proposes an iterative averaging method named IAM. Moreover, this paper provides concreate theoretical and empirical analysis of the proposed methods. On the theory side, it provides convergence guarantee of both SPS* and IAM under different local assumptions, such as local Lipschitzness, local smoothness, and local strongly convexity. Empirically, this paper provides experiment results on blackbox distillation tasks of language models, which is a task where $f(x\_*)$ is known. Under this task, SGD and Adam with IAM schedule (which does not require any lr tuning) outperforms other schedules whose base learning rate is carefully tuned.

**Strengths:**

- The paper is well-written. Table 1 clearly highlights previous results in related works and how the new results improve the current literature. The paper also motivates the choice of the exact formula of SPS* and IAM. The main results are also clearly presented are easy to follow.
- The theoretical analysis is solid and novel. The convergence analysis covers many settings where the loss is assumed to be locally Lipschitz, smooth, or strongly convex. The convergence rates under smooth assumption are in particular notable because these rates exhibit interpolation between an accelerated rate $O(1/t)$ and the standard rate $O(\sqrt{\Delta_*}/\sqrt{t})$ depending on the (dataset) interpolation constant $\Delta_*$. As $\delta_*\to 0$, these rates automatically adapts to the accelerated rate. Moreover, the convergence analysis of SPS with momentum is novel and is rarely studies in prior works.
- The empirical result on blackbox distillation is encouraging: IAM doesn't require tuning on learning rate, yet it outperforms the cosine decay with warmup schedule that requires a careful tuning on base lr via grid search.

**Weaknesses:**

- As the paper has clearly pointed out, SPS* and IAM requires knowing the stochastic loss minimum $f_t(x_*)$ in each iteration, thus limiting its practical applications other than the interpolation problem and blackbox distillation problem where such minimum is given. However, I personally find this limitation only a minor limitation because SPS* can be a target formula to be learned on-the-fly by more practical schedules.
- While the convergence analysis only requires an assumption of local bound as defined in eq (7), which later translates to local Lipschitzness or local smoothness assumptions, these assumptions on made on the stochastic loss $f_t$. On the other hand, while previous analysis usually makes global assumptions on Lipschitzness or smoothness, such assumptions are on the expected loss $\mathbb{E}[f_t]$. Moreover, I find the term "local" a bit confusing because it actually refers to the entire bounded domain $B(x_*,D)$ where $D=\\|x_0-x_*\\|$ instead some small neighborhood around $x_*$, especially when the iterates are monotonically converging to $x_*$.
- Regarding experiment results, the current baseline only considers constant schedule and cosine decay with linear warmup schedule. However, I think it's worth comparing to other popular schedules for better comparison, such as linear decay and WSD (trapezoid-shape warmup-stable-decay). Especially the latter schedule is observed to be better than cosine annealing for certain tasks.

**Questions:**

- See previous section for most questions.
- Regarding Thm 3.2, what's the order of the Bregman divergence terms? Does it cancel out certain terms and improves the overall rate if it's moved to RHS of the bound?

---

> ### Author Response · Authors · 2025-11-21
> **Responses for Reviewer oMJJ**
>
> Dear reviewer, we thank you for your feedback and comments. We address all questions below.
>
> > **Q1.** As the paper has clearly pointed out, SPS* and IAM requires knowing the stochastic loss minimum in each iteration, thus limiting its practical applications other than the interpolation problem and blackbox distillation problem where such minimum is given. However, I personally find this limitation only a minor limitation because SPS* can be a target formula to be learned on-the-fly by more practical schedules.
>
> **A1.** We fully agree with this point: while the assumption of knowing $f_\xi(x*)$ for all $\xi$ is strong, our paper contributes interesting theoretical results and novel applications. We think that the current submission is very transparent with this limitation.
>
> > **Q2.** While the convergence analysis only requires an assumption of local bound as defined in eq (7), which later translates to local Lipschitzness or local smoothness assumptions, these assumptions on made on the stochastic loss $f_t$. On the other hand, while previous analysis usually makes global assumptions on Lipschitzness or smoothness, such assumptions are on the expected loss $\mathbb{E}[f_t]$. Moreover, I find the term "local" a bit confusing because it actually refers to the entire bounded domain $B(x_*, D)$ where $D = \Vert x_0 - x_* \Vert$ instead some small neighborhood around $x_*$, especially when the iterates are monotonically converging to $x_*$.
>
> **A2.**
> About our local assumption on f_i vs. global assumption on f : this is a fair point. There is indeed a literature where assumptions can be made directly on the expected function f instead of the f_i. We first argue that there is not much difference between making assumptions on f or f_i. In most applications the function f_i is the composition of a simple loss with a parametrized model, and f and f_i generally share the same regularity properties (Lipschitzness or smoothness). There is even a slight advantage in making an assumption on f_i instead of f, which is that the constants into play are usually lower (think about the expectation of the smoothness constants which is smaller than the smoothness of the overall function). Second, we stress that there is a huge difference between making local vs. global assumptions. Most problems have a hard time verifying global bounds. Instead, local bounds are *for free* when the f_i are convex or C^2 and the sample space is finite (see Remark 2.4).
>
> About the term “local” : the reviewer is correct that this word carries an ambiguity between “around a point” and “not global / on bounded sets”. But within our framework, both are equivalent! This is due to compactness arguments (see Proposition B.7 and B.14 for instance). It allows us to say, for finite sum problems, that our hypotheses are immediately satisfied if the functions f_i are convex or of class C^2 (two purely local notions). We are open to suggestions for re-naming this assumption, but so far we could not find a more suitable (and compact) description. We will reconsider this point for the final version of the paper.
>
>
> > **Q3**  Regarding experiment results, the current baseline only considers constant schedule and cosine decay with linear warmup schedule. However, I think it's worth comparing to other popular schedules for better comparison, such as linear decay and WSD (trapezoid-shape warmup-stable-decay). Especially the latter schedule is observed to be better than cosine annealing for certain tasks.
>
> **A3** We agree that comparing additional schedules will provide further insight. We opted for the cosine schedule as it can still be considered the de-facto choice in deep learning. Furthermore, to the best of our knowledge the improvements of WSD or linear-decay over cosine are of a much smaller magnitude than the improvement of cosine over constant (e.g. see https://arxiv.org/abs/2405.18392), and the performance of different schedules also can be problem-dependent. But if the reviewer thinks this is important, we will add comparisons
> to a tuned WSD schedule as well in our revised paper.
>
>
> > **Q4**  Regarding Thm 3.2, what's the order of the Bregman divergence terms? Does it cancel out certain terms and improves the overall rate if it's moved to RHS of the bound?
>
> **A4** We cannot easily quantify how big are those (nonnegative) divergence terms, without making additional assumptions. An example of using these terms can be found in the proof of online-to-batch conversion for strongly convex losses (Theorem 6) in “The Road Less Scheduled” by A. Defazio et al, where they are combined with a strong convexity assumption to get an improvement.

---

### Meta-Review · Area_Chair_qLXN · 2025-12-14

**Summary:**

The reviewers indicated the following concerns:
- The proposed method requires knowing the stochastic loss minimum $f_t(x_*)$ at each iteration, which is unrealistic in many applications
- The local Lipschitzness or local smoothness assumptions are imposed in stochastic functions, which are stronger than assumptions on the expected functions
- The experimental analysis did not consider other popular schedules for better comparison, such as linear decay and WSD
- The parameter tuning is replaced with selecting and deploying a suitable teacher model, which may be more complex
- The assumption that teachers loss is approximately a students loss at optimum is strong. The teacher can be a poor optimization guide
- The proposed method cannot achieve an acceleration in the convergence
- The theoretical analysis is restricted to convex problems, while the application considers model distillation, which is nonconvex
- The theoretical contribution is not clear

**Reviewer Concerns:**

After reading the authors' response, I think the following concerns may not be well addressed
- The proposed method requires knowing the stochastic loss minimum $f_t(x_*)$ at each iteration, which is unrealistic in many applications (the authors simply said that they indicated this is a transparent limitation in the paper, which, however, is not a convincing argument)
- The experimental analysis did not consider other popular schedules for better comparison, such as linear decay and WSD (the authors did not add experimental results in the rebuttal phase, instead they mentioned that the improvement of linear decay over cosine decay is marginal. They also promised to add comparisons with more decays)
- The proposed method cannot achieve an acceleration in the convergence (the authors mentioned that this is a limitation of their algorithm)
- The theoretical analysis is restricted to convex problems, while the application considers model distillation, which is nonconvex (the authors indicated that they also have some experimental results on convex problems. However, as the major application is model distillation due to the requirement of knowing the optimal value, the lack of analysis to nonconvex case is indeed an issue)

The following concerns are well addressed
- The parameter tuning is replaced with selecting and deploying a suitable teacher model, which may be more complex (the authors explained that deploying a suitable teacher model is not difficult and provide some examples)
- The assumption that teachers loss is approximately a students loss at optimum is strong. The teacher can be a poor optimization guide (the authors provided experimental analysis to show that teacher loss is a good enough approximation)
- The theoretical contribution is not clear (the authors clarified their theoretical contribution in the responses)

**Reviewer Scores:**

Reviewer oMJJ is unlikely to change his/her score as his/her concerns on knowing $f_t(x_*)$ per iteration is not addressed. Furthermore, the concern on comparison with more stepsize schedules is not addressed

Reviewer hVNM is unlikely to change his/her score as his/her concerns on knowing $f_t(x_*)$ per iteration is not addressed.

Reviewer BoT6 is unlikely to change his/her score as his/her concerns on knowing $f_t(x_*)$ per iteration is not addressed. Furthermore, the proposed method cannot achieve an acceleration.

Reviewer LzVz is unlikely to change his/her score as his/her concerns on knowing $f_t(x_*)$ per iteration is not addressed. Furthermore, there is a lack of analysis in the nonconvex problems as indicated by the reviewer.

---

### Decision · Program_Chairs · 2026-01-26

Reject